# Sample Complexity of Distributionally Robust Average-Reward Reinforcement Learning

**Zijun Chen**
Department of Computer Science and Engineering
Hong Kong University of Science and Technology
zchendg@connect.ust.hk

**Shengbo Wang**
Daniel J. Epstein Department of
Industrial and Systems Engineering
University of Southern California
shengbow@usc.edu

**Nian Si**[*]
Department of Industrial Engineering
and Decision Analytics
Hong Kong University of Science and Technology
niansi@ust.hk

## Abstract

Motivated by practical applications where stable long-term performance is critical—such as robotics, operations research, and healthcare—we study the problem of distributionally robust (DR) average-reward reinforcement learning. We propose two algorithms that achieve near-optimal sample complexity. The first reduces the problem to a DR discounted Markov decision process (MDP), while the second, Anchored DR Average-Reward MDP, introduces an anchoring state to stabilize the controlled transition kernels within the uncertainty set. Assuming the nominal MDP is uniformly ergodic, we prove that both algorithms attain a sample complexity of $\widetilde{O}\left(|\mathbf{S}||\mathbf{A}|t_{\mathrm{mix}}^2\varepsilon^{-2}\right)$ for estimating the optimal policy as well as the robust average reward under KL and $f_k$-divergence-based uncertainty sets, provided the uncertainty radius is sufficiently small. Here, $\varepsilon$ is the target accuracy, $|\mathbf{S}|$ and $|\mathbf{A}|$ denote the sizes of the state and action spaces, and $t_{\mathrm{mix}}$ is the mixing time of the nominal MDP. This represents the first finite-sample convergence guarantee for DR average-reward reinforcement learning. We further validate the convergence rates of our algorithms through numerical experiments.

## 1 Introduction

Reinforcement learning (RL) [35] is a core machine learning framework in which agents learn to make decisions by interacting with their environments to maximize long-term rewards. RL has been successfully applied across a wide range of domains—from classic applications in robotics and control systems [19, 13] to more recent advances in game playing [21, 6, 8] and large language model (LLM)-driven reasoning tasks [47, 14].

A central assumption in RL is that the training environment (e.g. a simulator) faithfully represents the real-world deployment setting. In practice, however, this assumption rarely holds, leading to fragile policies underperform when exposed to mismatches between training and deployment environments. This remains a major obstacle to translating RL's successes in simulated settings to reliable performance in real-world applications.

To address this challenge, Zhou et al. [56] built upon the distributionally robust Markov decision process (DR-MDP) framework [16, 27, 48] to propose a distributionally robust reinforcement learning

---

[*]Corresponding author

39th Conference on Neural Information Processing Systems (NeurIPS 2025).

(DR-RL) framework. Subsequent work advanced the field, including both model-free [22, 40, 43] and model-based settings [29, 50, 7, 33], as well as approaches for offline learning [32] and generative models [40, 43, 53, 7], along with functional approximations [2, 24].

However, the aforementioned developments predominantly focus on discounted-reward or finite-horizon settings, while the average-reward case remains largely overlooked. This gap is significant because average-reward reinforcement learning is crucial in many practical applications where long-term performance matters more than short-term gains. For example:

- Control systems (e.g., robotics, autonomous vehicles) often require optimizing steady-state performance rather than cumulative discounted rewards.

- Operations research problems (e.g., inventory management, queueing systems) rely on long-run average metrics for stability and efficiency.

- Healthcare or energy management applications may prioritize sustained optimal performance over finite-time rewards.

Average-reward RL is not only important but also more challenging in terms of algorithm design and theoretical analysis. In the standard (non-robust) RL setting, the minimax sample complexity for generative models in discounted-reward cases was resolved as early as 2013 [1]. In contrast, analogous results for average-reward settings were only developed much later, with recent advances in Wang et al. [37], Zurek and Chen [57, 58, 59, 60] under granular structural assumptions.

This paper marks the first systematic analysis of the statistical properties of distributional robust average-reward MDPs (DR-AMDPs) in the tabular setting, addressing a critical gap in the literature.

Specifically, we propose two algorithms that achieve near-optimal (in a minmax sense) sample complexity for learning DR-AMDPs. The first is based on a reduction to the DR discounted-reward MDP (DR-DMDP), where a discount factor $\gamma$ must be carefully chosen to balance the trade-off between finite sample statistical error and the algorithmic bias introduced by the reduction. The second algorithm, anchored DR-AMDP, modifies the entire uncertainty set of transition kernels by introducing an anchoring state with a certain calibration probability.

To demonstrate their statistical efficiency, we consider a tabular setting where the nominal MDP is uniformly ergodic (Definition 3.3) with a uniform mixing time upper bound $t_{\mathrm{mix}}$ for all stationary, Markovian, and deterministic policies. We show that, to learn the optimal robust average reward and policy within $\varepsilon$ accuracy, both algorithms achieve a sample complexity of $\widetilde{O}\left(|\mathbf{S}||\mathbf{A}|t_{\mathrm{mix}}^2\varepsilon^{-2}\right)$ under KL and $f_k$-divergence uncertainty sets, assuming a sufficiently small uncertainty radius $\delta$ (to be defined). Here, $|\mathbf{S}|$ and $|\mathbf{A}|$ denote the cardinality of the state and action spaces, respectively. Compared to standard (non-robust) average-reward RL literature, this rate is optimal in its dependence on $|\mathbf{S}||\mathbf{A}|$ and $\varepsilon$.

Our analysis establishes three key contributions to the theory of DR-RL under the average-reward criterion. First, we address a fundamental modeling challenge: conventional uncertainty sets can contain MDPs that are not unichain, thereby invalidating the standard Bellman equations. To resolve this issue, we derive structural conditions on the uncertainty set that ensure stability for all MDPs within it. Second, we develop and analyze the reduction yielding the first stability-sensitive sample complexity bound for DR-DMDPs of $\widetilde{O}(|\mathbf{S}||\mathbf{A}|t_{\mathrm{mix}}^2(1-\gamma)^{-2}\varepsilon^{-2})$ and hence the aforementioned upper bound for DR-AMDPs. Building on this framework, we introduce the anchored algorithm and show that its output coincides with that of the reduction approach under a suitable choice of the anchoring parameter. Third, both algorithms are designed to function without requiring prior knowledge of model-specific parameters, particularly the mixing time $t_{\mathrm{mix}}$. Collectively, our work offers a unified treatment that connects robustness, stability, algorithm design, and finite-sample guarantees for DR-RL under the average-reward criterion.

The remainder of this paper is organized as follows: Section 2 surveys existing results for both standard and robust RLs. Section 3 introduces the mathematical preliminaries, including key notations and problem formulation. Our main theoretical contributions, including algorithmic development and sample complexity analysis, are presented in Section 4. Finally, Section 5 provides empirical validation of our theoretical findings and Section 6 concludes the paper and discusses future work.

Table 1: Summary of S.O.T.A. sample complexity results in the literature, where $t_{\mathrm{mix}}$ is defined in 3.2.

| | Type | Sample Complexity | Origin |
|---|---|---|---|
| **Standard** | Discounted | $\widetilde{\Theta}(|\mathbf{S}||\mathbf{A}|(1-\gamma)^{-3}\epsilon^{-2})$ | Azar et al. [1], Li et al. [20] |
| | Discounted Mixing | $\widetilde{\Theta}\left(|\mathbf{S}||\mathbf{A}|t_{\mathrm{mix}}(1-\gamma)^{-2}\varepsilon^{-2}\right)$ | Wang et al. [37] |
| | Average Mixing | $\widetilde{\Theta}\left(|\mathbf{S}||\mathbf{A}|t_{\mathrm{mix}}\varepsilon^{-2}\right)$ | Wang et al. [38], Zurek and Chen [57] |
| **DR-RL** | Discounted | $\widetilde{O}(|\mathbf{S}||\mathbf{A}|(1-\gamma)^{-4}\epsilon^{-2})$ | Shi and Chi [32], Wang et al. [43] |
| | Discounted Mixing | $\widetilde{O}\left(|\mathbf{S}||\mathbf{A}|t_{\mathrm{mix}}^2(1-\gamma)^{-2}\varepsilon^{-2}\right)$ | Theorem 4.3 |
| | Average Mixing | $\widetilde{O}(|\mathbf{S}||\mathbf{A}|t_{\mathrm{mix}}^2\varepsilon^{-2})$ | Theorem 4.4 & 4.5 |

## 2   Literature Review

**Sample Complexity of Discounted-Reward Tabular RL:** There is an extensive literature on the sample complexity of tabular reinforcement learning. In standard (non-robust) settings, the minimax sample complexity for discounted-reward problems has been well studied. Early works by Azar et al. [1], Li et al. [20] established the minimax rate of $\widetilde{\Theta}(|\mathbf{S}||\mathbf{A}|(1-\gamma)^{-3}\varepsilon^{-2})$. More recent research has shifted toward instance-dependent bounds that leverage structural properties of the MDP. For example, Wang et al. [38], Zurek and Chen [59] derive tighter instance-dependent bounds of $\widetilde{O}(|\mathbf{S}||\mathbf{A}|t_{\mathrm{mix}}(1-\gamma)^{-2}\varepsilon^{-2})$ and $\widetilde{O}(|\mathbf{S}||\mathbf{A}|\mathrm{H}(1-\gamma)^{-2}\varepsilon^{-2})$ under the assumptions that $P$ is uniformly ergodic or weakly communicating, where H denotes the span of the relative value function.

**Sample Complexity of Average-Reward Tabular RL:** Recently, there has been growing interest in the sample complexity of average-reward reinforcement learning in standard (non-robust) settings. Early work by Jin and Sidford [17] established a bound of $\widetilde{O}(|\mathbf{S}||\mathbf{A}|t_{\mathrm{mix}}^2\varepsilon^{-2})$ using primal-dual stochastic mirror descent. A rate of $\widetilde{O}(|\mathbf{S}||\mathbf{A}|t_{\mathrm{mix}}\varepsilon^{-3})$ was established later via a reduction to the discounted MDP setting [18]. Subsequent analyses achieved tighter bounds: Wang et al. [38] obtained a rate of $\widetilde{O}(|\mathbf{S}||\mathbf{A}|t_{\mathrm{mix}}\varepsilon^{-2})$, while Zurek and Chen [57, 59] derived instance-dependent optimal rates of $\widetilde{O}(|\mathbf{S}||\mathbf{A}|\mathrm{H}\varepsilon^{-2})$ for weakly communicating MDPs. Further refinements [58, 60] achieved similar rates without requiring prior knowledge, using plug-in and span penalization approaches. Beyond the generative model setting, Zhang et al. [55] and Chen [5] provided finite-sample analyses for synchronous and asynchronous $Q$-learning, respectively. Asymptotic properties were also studied in Yu et al. [54], Wan et al. [36] for asynchronous $Q$-learning.

**DR-DMDP and DR-RL:** Our work builds on the theoretical foundations of robust MDPs [11, 16, 27, 48, 49, 31, 42], which primarily develop dynamic programming principles under the discounted-reward setting. Recent advances in distributionally robust reinforcement learning (DR-RL) have investigated the sample complexity of DR-DMDPs under various divergence-based uncertainty sets. For example, Wang et al. [43], Shi and Chi [32] establish a model-free upper bound of $\widetilde{O}(|\mathbf{S}||\mathbf{A}|(1-\gamma)^{-4}\varepsilon^{-2})$ under KL-divergence. Under $\chi^2$-divergence, Shi et al. [33] obtain a similar upper bound, while Clavier et al. [7] show that $l_p$-norm constraints admit a tighter minimax rate of $\widetilde{\Theta}(|\mathbf{S}||\mathbf{A}|(1-\gamma)^{-3}\varepsilon^{-2})$. In this paper, by incorporating the mixing time parameter, we present a "instance-dependence" sample complexity bound of $\widetilde{O}\left(|\mathbf{S}||\mathbf{A}|t_{\mathrm{mix}}^2(1-\gamma)^{-2}\varepsilon^{-2}\right)$, which improves the dependence on the effective horizon from $(1-\gamma)^{-4}$ to $(1-\gamma)^{-2}$. Several other works contribute to the theoretical and algorithmic landscape of DR-RL in various settings, including Panaganti and Kalathil [28], Yang et al. [52], Xu et al. [50], Blanchet et al. [3], Liu et al. [23], Wang et al. [41], Yang et al. [51].

**Distributionally Robust Average-Reward MDPs:** While the sample complexity of learning DR-DMDPs has been extensively studied, the average-reward setting remains relatively underexplored. Wang et al. [46, 44, 45] propose robust relative value iteration and TD/Q-learning algorithms and prove their convergence, but without providing sample complexity guarantees. Grand-Clément et al. [12] show that for $(s, a)$-rectangular uncertainty sets, the optimal policy can be stationary and deterministic; however, this result may not extend to $s$-rectangular uncertainty sets. More recently, Wang and Si [39] study average-reward robust MDPs under $s$-rectangular uncertainty and provide

one-sided weak communication conditions that ensure the existence of solutions to the Bellman optimality equation. Although existing work has investigated the existence and structure of optimal policies in average-reward DR-RL, non-asymptotic sample complexity bounds remain an open question.

We provide a summary of state-of-the-art sample complexity results in the literature in Table 1. In particular, we establish the first sample complexity guarantees for the average-reward DR-RL formulation, which achieves optimal dependence for $\varepsilon$ and $|\mathbf{S}||\mathbf{A}|$.

# 3 Preliminaries

## 3.1 Markov Decision Processes

We briefly review and define some notations for classical tabular MDP models. Let $\Delta(\mathbf{S})$ denotes the probability simplex over $\mathbb{R}^\mathbf{S}$. A finite discounted MDP (DMDP) is defined by the tuple $(\mathbf{S}, \mathbf{A}, r, P, \gamma)$. Here, $\mathbf{S}, \mathbf{A}$ denote the finite state and action spaces respectively; $r : \mathbf{S} \times \mathbf{A} \to [0, 1]$ is the reward function; $P = \{p_{s,a} \in \Delta(\mathbf{S}) : (s, a) \in \mathbf{S} \times \mathbf{A}\}$ is the controlled transition kernel, and $\gamma \in (0, 1)$ is the discount factor. An average-reward MDP (AMDP) model, on the other hand, is specified by $(\mathbf{S}, \mathbf{A}, r, P)$ without the discount factor.

Define the canonical space $\Omega = (\mathbf{S} \times \mathbf{A})^\mathbb{N}$ equipped with $\mathcal{F}$ the $\sigma$-field generated by cylinder sets. The state-action process $\{(S_t, A_t), t \geq 0\}$ is defined by the point evaluation $X_t(\omega) = s_t, A_t(\omega) = a_t$ for all $t \geq 0$ for any $\omega = (s_0, a_0, s_1, a_1, \dots) \in \Omega$. A general history dependent policy $\pi = (\pi_t)_{t \geq 0} \in \Pi_{\text{HD}}$ is a sequence of the agent's decision rule. Here, the decision rule $\pi_t$ at time $t$ is a mapping $\pi_t : (\mathbf{S} \times \mathbf{A})^t \times \mathbf{S} \to \Delta(\mathbf{A})$, signifying the conditional distribution of $A_t$ given the history. It is known in the literature [30, 12] that to achieve optimal decision making in the context of infinite horizon AMDPs, DMDPs, or their robust variants (to be introduced), it suffices to consider the policy class $\Pi$ of stationary, Markov, and deterministic policies; i.e. $\pi \in \Pi$ can be seen as a function $\pi : \mathbf{S} \to \mathbf{A}$. Thus, in the subsequent development, we restrict our discussion to $\Pi$.

As in Wang et al. [38] a policy $\pi \in \Pi$ and an initial distribution $\mu \in \Delta(\mathbf{S})$ uniquely defines a probability measure on $(\Omega, \mathcal{F})$. We will always assume that $\mu$ is the uniform distribution over $\mathbf{S}$. The expectation under this measure is denoted by $E_P^\pi$. To simplify notation, we define $P_\pi(s, s') := \sum_{a \in \mathbf{A}} \pi(a|s) p_{s,a}(s')$ and $r_\pi(s) := \sum_{a \in \mathbf{A}} \pi(a|s) r(s, a)$.

**Discounted-reward MDP (DMDP):** Given a DMDP instance $(\mathbf{S}, \mathbf{A}, r, P, \gamma)$ and $\pi \in \Pi$, the *discounted value function* $V_P^\pi : \mathbf{S} \to \mathbb{R}$ is defined as: $V_P^\pi(s) = E_P^\pi [\sum_{t=0}^\infty \gamma^t r(S_t, A_t)|S_0 = s]$. An optimal policy $\pi^* \in \Pi$ achieves the optimal value $V_P^*(s) := \max_{\pi \in \Pi} V_P^\pi(s)$.

**Average-reward MDP (AMDP):** For AMDP model $(\mathbf{S}, \mathbf{A}, r, P)$ and $\pi \in \Pi$, the *long-run average-reward function* $g_P^\pi : \mathbf{S} \to \mathbb{R}$ is defined as $g_P^\pi(s) := \limsup_{T \to \infty} T^{-1} E_P^\pi [\sum_{t=0}^{T-1} r(S_t, A_t)|S_0 = s]$.

When $P$ is uniformly ergodic (a.k.a.unichain, to be defined later), the $g_P^\pi$ is constant across states [30]. In this context, an optimal policy $\pi^* \in \Pi$ achieves the long-run average-reward $\max_{\pi \in \Pi} g_P^\pi$.

## 3.2 Uniform Ergodicity

Motivated by engineering applications where policies induce systems that are stable in the long run, we consider a stability property of MDPs known as *uniform ergodicity*, a stronger version of the *unichain* property. In this setting, the controlled Markov chain induced by any reasonable policy converges in distribution to a unique steady state in total variation distance $\|\cdot\|_{\text{TV}}$ defined by $\|p - q\|_{\text{TV}} := \sup_{A \subset \mathbf{S}} |p(A) - q(A)|$ for probability vectors $p, q \in \Delta(\mathbf{S})$.

We start with reviewing concepts relevant to uniformly ergodic Markov chains.

**Definition 3.1.** (Uniform Ergodicity) A transition kernel $K \in \mathbb{R}^{\mathbf{S} \times \mathbf{S}}$ is uniformly ergodic if one of the following holds

- There exists a probability measure $\rho$ for which $\|K^n(s, \cdot) - \rho\|_{\text{TV}} \to 0$ for all $s \in \mathbf{S}$.

- $K$ satisfies the $(m, p)$-*Doeblin condition*: For some $m \in \mathbb{N}$ and $p \in (0, 1]$ if there exists a probability measure $\psi$ and a stochastic kernel $R$ s.t. $K^m(s, s') = p\psi(s') + (1 - p)R(s, s')$.

It is well known [25] that $\rho$ must be the unique stationary distribution of $K$ and that two conditions are equivalent. The $\psi$ and $R$ in the Doeblin condition are known as the minorization measure and the residual kernel respectively.

Next, we introduce the mixing and minorization times associated with a uniformly ergodic kernel $K$.

**Definition 3.2.** (Mixing Time and Minorization Time) Define the mixing time of a uniformly ergodic transition kernel $P$ as $t_{\mathrm{mix}}(K) := \inf \left\{ m \geq 1 : \max_{s \in \mathbf{S}} \|K^m(s, \cdot) - \rho(\cdot)\|_{\mathrm{TV}} \leq \frac{1}{4} \right\}$, and the minorization time as $t_{\mathrm{minorize}}(K) := \inf \{ m/p : \min_{s \in \mathbf{S}} K^m(s, \cdot) \geq p\psi(\cdot) \text{ for some } \psi \in \Delta(\mathbf{S}) \}$.

It is shown in Theorem 1 of Wang et al. [37] that for a uniformly ergodic transition kernel $K$, these metrics of stability are equivalent up to constants: $t_{\mathrm{minorize}}(K) \leq 22 t_{\mathrm{mix}}(K) \leq 22 \log(16) t_{\mathrm{minorize}}(K)$.

While the MDP sample complexity literature typically uses the mixing time as a complexity parameter, for our purposes, the Doeblin condition and the associated minorization time offer sharper theoretical insights into how adversarial robustness affects the statistical complexity of RL. Given the equivalence between $t_{\mathrm{mix}}(K)$ and $t_{\mathrm{minorize}}(K)$, and the latter's advantage in revealing these insights, we will use $t_{\mathrm{minorize}}(K)$ time throughout this work.

Having reviewed the uniform ergodicity of a stochastic kernel $K$, we define uniformly ergodic MDPs.

**Definition 3.3** (Uniformly Ergodic MDP). An MDP (or its controlled transition kernel $P$) is said to be uniformly ergodic if for all policies $\pi \in \Pi$, $t_{\mathrm{minorize}}(P_\pi) < \infty$. Then, define $t_{\mathrm{minorize}} := \max_{\pi \in \Pi} t_{\mathrm{minorize}}(P_\pi) < \infty$.

To provide sharper sample complexity results, it is useful to define the following upper bound parameter on $m$.

$$m_\vee := \max_{\pi \in \Pi} \inf \{ m : P_\pi \text{ is } (m, p) - \text{Doeblin and } m/p = t_{\mathrm{minorize}}(P_\pi) \text{ for some } p \}. \quad (3.1)$$

This is well defined: In Appendix A, Lemma A.3, we prove that for any transition kernel $P_\pi$, the equality $m/p = t_{\mathrm{minorize}}(P_\pi)$ is always attained by some $m$ and $p$ s.t. $\min_{s \in \mathbf{S}} P_\pi^m(s, \cdot) \geq p\psi(\cdot)$.

It is easy to see that $m_\vee \leq t_{\mathrm{minorize}}$, and we will demonstrate by the example in Section 5 that it is possible for $m_\vee = 1$ while $t_{\mathrm{minorize}}$ can be arbitrarily large.

### 3.3 Distributionally Robust Discounted-Reward and Average-Reward MDPs

This paper focuses on a robust MDP setting where the stochastic dynamics of the system is influenced by adversarial perturbations on the transition structure. We assume the presence of an adversary that can transition probabilities within KL or $f_k$-divergence uncertainty sets. Specifically, for probability measures $q, p \in \Delta(\mathbf{S})$ where $q$ is absolutely continuous w.r.t. $p$, denoted by $q \ll p$, we define $D_{\mathrm{KL}}(q\|p) := \sum_{s \in \mathbf{S}} \log(q(s)/p(s)) q(s)$ and $D_{f_k}(q\|p) := \sum_{s \in \mathbf{S}} f_k(q(s)/p(s)) p(s)$. Here, the function $f_k$ is defined for $k \in (1, \infty)$ by $f_k(t) = (t^k - kt + k - 1)/(k(k-1))$. When $k = 2$, $D_{f_k}$ is the $\chi^2$-divergence.

We assume that the underlying MDP has an **unknown** *nominal controlled transition kernel*

$$P = \{ p_{s,a} \in \Delta(\mathbf{S}) : (s, a) \in \mathbf{S} \times \mathbf{A} \}. \quad (3.2)$$

For each $(s, a) \in \mathbf{S} \times \mathbf{A}$ we define the uncertainty set under divergence $D = D_{\mathrm{KL}}, D_{f_k}$ and parameter $\delta > 0$ centered at $p_{s,a}$ by $\mathcal{P}_{s,a}(D, \delta) := \{ p : D(p\|p_{s,a}) \leq \delta \}$. This set contains all possible adversarial perturbations of the transition out of $(s, a)$. Note that the parameter $\delta$ controls the size of the $\mathcal{P}_{s,a}(D, \delta)$, quantifying the power of the adversary. The uncertainty set for the entire controlled transition kernel is $\mathcal{P}(D, \delta) := \times_{(s,a) \in \mathbf{S} \times \mathbf{A}} \mathcal{P}_{s,a}(D, \delta)$. An uncertainty set of this product from is called SA-rectangular [42].

We will suppress the dependence of $D$ and $\delta$ when it is clear from the context. Also, for notation simplicity, define the mapping $\Gamma_{\mathcal{P}_{s,a}} : \mathbb{R}^{\mathbf{S}} \to \mathbb{R}$ for $\mathcal{P}_{s,a} \subset \Delta(\mathbf{S})$ by

$$\Gamma_{\mathcal{P}_{s,a}}(V) := \inf_{p \in \mathcal{P}_{s,a}} E_{S \sim p} [V(S)].$$

Optimal distributionally robust Bellman operators $\mathcal{T}_\gamma^*$ and $\mathcal{T}^*$ are central to our algorithmic design.

**Definition 3.4.** The optimal DR Bellman operators $\mathcal{T}_\gamma^*, \mathcal{T}^* : \mathbb{R}^{\mathbf{S}} \to \mathbb{R}^{\mathbf{S}}$ are defined by

$$
\begin{aligned}
\mathcal{T}_\gamma^*(V)(s) &:= \max_{a \in \mathbf{A}} \left\{ r(s,a) + \gamma \Gamma_{\mathcal{P}_{s,a}}(V) \right\} \\
\mathcal{T}^*(v)(s) &:= \max_{a \in \mathbf{A}} \left\{ r(s,a) + \Gamma_{\mathcal{P}_{s,a}}(v) \right\}
\end{aligned}
\tag{3.3}
$$

**DR-DMDP:** A DR-DMDP model is given by the tuple $(\mathbf{S}, \mathbf{A}, \mathcal{P}, r, \gamma)$. For fixed $\pi \in \Pi$, define the DR value function

$$
V_{\mathcal{P}}^\pi(s) := \inf_{\mathbf{P} \in (\mathcal{P})^{\mathbb{N}}} E_{\mathbf{P}}^\pi \left[ \sum_{t=0}^\infty \gamma^t r(S_t, A_t) \middle| S_0 = s \right].
\tag{3.4}
$$

See Iyengar [16] for a rigorous construction of the expectation $E_{\mathbf{P}}^\pi$. Then, the optimal value function is $V_{\mathcal{P}}^*(s) := \max_{\pi \in \Pi} V_{\mathcal{P}}^\pi(s)$. It is well known (c.f. Iyengar [16]) that $V_{\mathcal{P}}^*$ is the unique solution of the DR Bellnbman equation: $V_{\mathcal{P}}^* = \mathcal{T}_\gamma^*(V_{\mathcal{P}}^*)$.

Note that the expectation $E_{\mathbf{P}}^\pi$ is under the adversarial perturbation from a Markovian policy class $(\mathcal{P})^{\mathbb{N}}$. It is possible to consider other information structures for the adversary while retaining the satisfaction of the Bellman equation [42].

**DR-AMDP:** A DR-AMDP model is given by the tuple $(\mathbf{S}, \mathbf{A}, \mathcal{P}, r)$. To simplify our presentation, we restrict our consideration to uniformly ergodic DR-AMDPs.

For each $\pi \in \Pi$ we define the DR long-run average-reward function by

$$
g_{\mathcal{P}}^\pi(s) := \inf_{\mathbf{P} \in (\mathcal{P})^{\mathbb{N}}} \limsup_{T \to \infty} E_{\mathbf{P}}^\pi \left[ \frac{1}{T} \sum_{t=0}^{T-1} r(S_t, A_t) \middle| S_0 = s \right].
\tag{3.5}
$$

Natually, the optimal average reward is $g_{\mathcal{P}}^*(s) := \max_{\pi \in \Pi} g_{\mathcal{P}}^\pi(s)$.

This paper focuses on a setting where the DR-AMDP is uniformly ergodic in the following sense.

**Definition 3.5.** A DR-AMDP (or $\mathcal{P}$) is said to be uniformly ergodic if for all controlled kernels $Q \in \mathcal{P}$, $Q$ is uniformly ergodic as in Definition 3.3.

We note that $\mathcal{P} = \mathcal{P}(D, \delta)$ is compact in the sense that $\mathcal{P}_{s,a}(D, \delta)$ is a compact subset of $\Delta(\mathbf{S})$ for all $s, a$. With uniform ergodicity and compactness, Wang et al. [44] shows that $g_{\mathcal{P}}^*(s)$ is constant for $s \in \mathbf{S}$ which uniquely solves the DR Bellman equations.

**Proposition 3.6** (Theorems 7 an 8 of Wang et al. [44])**.** *If $\mathcal{P}$ is uniformly ergodic with a uniformly bounded minorization time, then $g_{\mathcal{P}}^*(s) \equiv g_{\mathcal{P}}^*$ is constant in $s \in \mathbf{S}$. Moreover, there exists a solution $(g, v)$ of $v(s) = \mathcal{T}^*(v)(s) - g^*(s)$ for all $s \in \mathbf{S}$ and any such solution satisfies $g(s) = g_{\mathcal{P}}^*$ for all $s \in \mathbf{S}$. Moreover, the policy $\pi^*(s) \in \arg\max_{a \in \mathbf{A}} \left\{ r(s,a) + \Gamma_{\mathcal{P}_{s,a}}(v) \right\}$ achieves the optimal average-reward $g_{\mathcal{P}}^*$.*

# 4  DR-AMDP: Algorithms and Sample Complexity Upper Bound

In this section, we introduce two algorithms for DR-AMDPs and establish their sample complexity upper bounds. Before presenting the algorithms and results, we first specify the assumptions on the data-generating process and MDP models, along with insights into their rationale and relevance.

We assume the availability of a simulator, a.k.a. a *generative model*, which allows us to sample independently from the nominal controlled transition kernel $p_{s,a}$, for any $(s,a) \in \mathbf{S} \times \mathbf{A}$. Given sample size $n$, we sample i.i.d. $\left\{ S_{s,a}^{(1)}, \cdots, S_{s,a}^{(n)} \right\}$ from $p_{s,a}$ and construct the empirical transition probability

$$
\widehat{p}_{s,a}(s') := \frac{1}{n} \sum_{i=1}^n \mathbb{1} \left\{ S_{s,a}^{(i)} = s' \right\}.
\tag{4.1}
$$

Unlike Wang et al. [44], which requires a unichain assumption on every element of $\mathcal{P}$, we only assume that the nominal controlled transition kernel $P$ is uniformly ergodic. We will establish that this weaker condition, coupled with a properly constrained adversarial uncertainty set in Assumption 2, will still guarantee the uniform ergodicity for all $Q \in \mathcal{P}$.

**Assumption 1.** The nominal controlled transition kernel $P$ in (3.2) is uniformly ergodic with minorization time $t_{\text{minorize}}$ as in Definition 3.3.

To introduce limits on the adversarial power and facilitate our sample complexity analysis, we introduce the following complexity metric parameter:

**Definition 4.1.** Define the minimum support as:

$$\mathfrak{p}_\wedge := \min_{(s,a,s')\in\mathbf{S}\times\mathbf{A}\times\mathbf{S}} \{p_{s,a}(s') : p_{s,a}(s') > 0\} \tag{4.2}$$

**Assumption 2.** Suppose the parameter $\delta$ satisfies $\delta \leq \frac{1}{8m_\vee^2}\mathfrak{p}_\wedge$ when $\mathcal{P} = \mathcal{P}(D_{\text{KL}}, \delta)$, and $\delta \leq \frac{1}{\max\{8,4k\}m_\vee^2}\mathfrak{p}_\wedge$ when $\mathcal{P} = \mathcal{P}(D_{f_k}, \delta)$.

Here, the constant $1/8$ can potentially be relaxed. As mentioned earlier, this restriction on the adversarial power parameter $\delta$ ensures the minorization times remain uniformly bounded across the uncertainty set by a constant multiple of the nominal controlled kernel's minorization time.

**Proposition 4.2.** *Suppose Assumptions 1 hold, and $\mathcal{P} = \mathcal{P}(D_{\text{KL}}, \delta)$ or $\mathcal{P}(D_{f_k}, \delta)$ satisfying Assumption 2. Then, for all $Q \in \mathcal{P}$ and $\pi \in \Pi$, $t_{\text{minorize}}(Q_\pi) \leq 2t_{\text{minorize}}$, where $t_{\text{minorize}}$ is from Assumption 1.*

The proof is deferred to Appendix B, E. We further note that without Assumption 2, the Hard MDP instance in Section 5 will have a non-mixing worst-case adversarial kernel and state-dependent optimal average reward even when $\delta = \Theta(\mathfrak{p}_\wedge/m_\vee^2)$. This emphasizes the necessity of limiting the adversarial power to obtain a stable worst-case system and state-independent average reward.

We propose two algorithms: Reduction to DR-DMDP and Anchored DR-AMDP. Notably, these are the first to provide finite-sample guarantees for DR-AMDPs and achieve the canonical $n^{-1/2}$ convergence rate in policy and estimation. Furthermore, both algorithms operate without requiring prior knowledge of $t_{\text{minorize}}$. Together, these contributions represent foundational advances in the study of data-driven learning of DR-AMDPs.

## 4.1 Reduction to DR-DMDP

First, we present the algorithmic reduction from DR-AMDP to DR-DMDP. The algorithm design in this section is inspired by prior works [46, 18]. Specifically, we apply value iteration to an auxiliary empirical DR-DMDP model to obtain both the value function and optimal policy. Utilizing a calibrated discount $\gamma = 1 - n^{-1/2}$ where $n$ is the input sample size in Algorithm 1, we achieve an $\varepsilon$-approximation of the target DR-AMDP value and policy with the auxiliary DR-DMDP using $\widetilde{O}(|\mathbf{S}||\mathbf{A}|t_{\text{minorize}}^2\mathfrak{p}_\wedge^{-1}\varepsilon^{-2})$ samples.

---

**Algorithm 1** Distributional Robust DMDP: $\text{DR} - \text{DMDP}(\gamma, n, D)$

---

**Input:** Discount factor $\gamma \in (0, 1)$, sample size $n \geq 1$, $D = D_{\text{KL}}$ or $D_{f_k}$.
For all $(s, a) \in \mathbf{S} \times \mathbf{A}$, compute the $n$-sample empirical transition probability $\widehat{p}_{s,a}$ as in (4.1)
Construct the uncertainty set as $\widehat{\mathcal{P}} = \times_{(s,a)\in\mathbf{S}\times\mathbf{A}} \widehat{\mathcal{P}}_{s,a}$ where $\widehat{\mathcal{P}}_{s,a} = \{p : D(p||\widehat{p}_{s,a}) \leq \delta\}$.
Compute the solution $V_{\widehat{\mathcal{P}}}^*$ as the solution to the empirical DR Bellman equation; i.e. $\forall s \in \mathbf{S}$:

$$V_{\widehat{\mathcal{P}}}^*(s) = \max_{a\in\mathbf{A}} \left\{ r(s, a) + \gamma\Gamma_{\widehat{\mathcal{P}}_{s,a}}(V_{\widehat{\mathcal{P}}}^*) \right\}.$$

Then, extract any optimal policy $\widehat{\pi}^* \in \Pi$ from $\widehat{\pi}^*(s) \in \arg\max_{a\in\mathbf{A}} \left\{ r(s, a) + \gamma\Gamma_{\widehat{\mathcal{P}}_{s,a}}(V_{\widehat{\mathcal{P}}}^*) \right\}$.
**return** $\widehat{\pi}^*, V_{\widehat{\mathcal{P}}}^*$

---

With the help of Proposition 4.2, the Algorithm 1 has the following optimal sample complexity guarantee.

**Theorem 4.3.** *Suppose $\mathcal{P} = \mathcal{P}(D_{\text{KL}}, \delta)$ or $\mathcal{P}(D_{f_k}, \delta)$ and Assumptions 1, 2 are in force. Then, for any $n \geq 32\mathfrak{p}_\wedge^{-1}\log(2|\mathbf{S}|^2|\mathbf{A}|/\beta)$, the policy $\widehat{\pi}^*$ and value function $V_{\widehat{\mathcal{P}}}^*$ returned by Algorithm 1*

*satisfy*

$$0 \le V_{\mathcal{P}}^* - V_{\mathcal{P}}^{\widehat{\pi}^*} \le \frac{c \cdot t_{\mathrm{minorize}}}{(1-\gamma)\sqrt{n\mathfrak{p}_\wedge}} \sqrt{\log(2|\mathbf{S}|^2|\mathbf{A}|/\beta)} \text{ and}$$

$$\|V_{\widehat{\mathcal{P}}}^* - V_{\mathcal{P}}^*\|_\infty \le \frac{c' \cdot t_{\mathrm{minorize}}}{(1-\gamma)\sqrt{n\mathfrak{p}_\wedge}} \sqrt{\log(2|\mathbf{S}|^2|\mathbf{A}|/\beta)}$$

(4.3)

*with probability at least $1 - \beta$, where the constants $c, c' \le 96\sqrt{2}$ for both the KL and $f_k$ cases.*

The proof of Theorem 4.3 is deferred to Appendix D, G. We note that Theorem 4.3 implies that to achieve an $\varepsilon$-optimal policy as well as producing a uniform $\varepsilon$-error estimate of $V_{\mathcal{P}}^*$ with high probability using Algorithm 1, we need $\widetilde{O}(|\mathbf{S}||\mathbf{A}|t_{\mathrm{minorize}}^2(1-\gamma)^{-2}\mathfrak{p}_\wedge^{-1}\varepsilon^{-2})$ samples. Compared to state-of-the-art sample complexity results for DR-DMDPs [32, 43, 33], Theorem 4.3 provides a significant refinement: when the nominal controlled kernel is uniformly ergodic, the effective horizon dependence improves to $(1-\gamma)^{-2}$. Notably, this $(1-\gamma)^{-2}$ scaling is also known to be optimal in the non-robust setting [37], which corresponds to DR-DMDPs when $\delta = 0$. As we will show, this optimal dependence directly enables the canonical $n^{-1/2}$ convergence rate for policy learning and value estimation in the DR-AMDP setting.

---

**Algorithm 2** Reduction to DMDP

---

**Input:** Samples size $n$.
Assign $\gamma = 1 - 1/\sqrt{n}$ and run Algorithm 1 with input $\mathrm{DR} - \mathrm{DMDP}(\gamma, n)$ to obtain $\widehat{\pi}^*, V_{\widehat{\mathcal{P}}}^*$.
**return** $\widehat{\pi}^*, V_{\widehat{\mathcal{P}}}^*/\sqrt{n}$

---

**Theorem 4.4.** *Suppose $\mathcal{P} = \mathcal{P}(D_{\mathrm{KL}}, \delta)$ or $\mathcal{P}(D_{f_k}, \delta)$ and Assumptions 1 and 2 are in force. Then for any $n \ge 32\mathfrak{p}_\wedge^{-1}\log(2|\mathbf{S}|^2|\mathbf{A}|/\beta)$, the policy $\widehat{\pi}^*$ and value function $V_{\widehat{\mathcal{P}}}^*/\sqrt{n}$ returned by Algorithm 2 satisfies*

$$0 \le g_{\mathcal{P}}^* - g_{\mathcal{P}}^{\widehat{\pi}^*} \le \frac{c \cdot t_{\mathrm{minorize}}}{\sqrt{n\mathfrak{p}_\wedge}} \sqrt{\log(2|\mathbf{S}|^2|\mathbf{A}|/\beta)} \text{ and}$$

$$\left\| \frac{V_{\widehat{\mathcal{P}}}^*}{\sqrt{n}} - g_{\mathcal{P}}^* \right\|_\infty \le \frac{c' \cdot t_{\mathrm{minorize}}}{\sqrt{n\mathfrak{p}_\wedge}} \sqrt{\log(2|\mathbf{S}|^2|\mathbf{A}|/\beta)}$$

(4.4)

*with probability $1 - \beta$, where the constants $c, c' \le 120\sqrt{2}$ for both the KL and $f_k$ cases.*

Again, we remark that Theorem 4.4 implies that to achieve an $\varepsilon$-optimal policy as well as producing a uniform $\varepsilon$-error estimate of the optimal robust long-run average reward with high probability using Algorithm 2, we need $\widetilde{O}(|\mathbf{S}||\mathbf{A}|t_{\mathrm{minorize}}^2\mathfrak{p}_\wedge^{-1}\varepsilon^{-2})$ samples.

## 4.2 Anchored DR-AMDP

In this section, we develop *anchored DR-AMDP* Algorithm 3 that avoids solving a DR-DMDP subproblem. Inspired by Fruit et al. [10], Zurek and Chen [58]'s anchoring approach for classical MDPs, our anchored DR-AMDP approach modifies the entire uncertainty set of controlled transition kernels via a uniform anchoring state $s_0$ and a calibrated strength parameter $\xi$. We show that Algorithm 3 enjoys the same error and sample complexity upper bounds to Algorithm 2.

**Theorem 4.5.** *Suppose Assumption 1 and 2 are in force. Then for any $n \ge 32\mathfrak{p}_\wedge^{-1}\log(2|\mathbf{S}|^2|\mathbf{A}|/\beta)$, the policy $\widehat{\pi}^*$ and value function $g_{\widehat{\mathcal{P}}}^*$ returned by Algorithm 3 satisfies (4.4) with $V_{\widehat{\mathcal{P}}}^*/\sqrt{n}$ replaced by $g_{\widehat{\mathcal{P}}}^*$ with probability at least $1 - \beta$.*

This theorem implies the same $\widetilde{O}(|\mathbf{S}||\mathbf{A}|t_{\mathrm{minorize}}^2\mathfrak{p}_\wedge^{-1}\varepsilon^{-2})$ sample complexity to achieve an $\varepsilon$-optimal policy and value estimation.

**Sketched Proof of Theorems 4.4 and 4.5.** Our proof begin with establishing that, under Assumption 2, each adversarial transition kernel $Q \in \mathcal{P}$ consists of conditional distributions $q_{s,a}$ that are absolutely continuous with respect to the nominal distributions $p_{s,a}$, with a uniform lower bound $1 - \frac{1}{2m_\vee}$ on its Radon-Nikodym derivative. This guarantees that $t_{\mathrm{minorize}}(Q_\pi) \le O(t_{\mathrm{minorize}})$ for all $Q \in \mathcal{P}$ and $\pi \in \Pi$.

---

**Algorithm 3** Anchored DR-AMDP

---

**Input:** Sample size $n \geq 1$ and divergence $D = D_{\mathrm{KL}}$ or $D_{f_k}$.
For all $(s,a) \in \mathbf{S} \times \mathbf{A}$, compute the $n$-sample empirical transition probability $\widehat{p}_{s,a}$ as in (4.1).
Let $\xi = 1/\sqrt{n}$ and fixed any anchoring point $s_0 \in \mathbf{S}$. Construct the anchored empirical uncertainty set as $\widehat{\underline{\mathcal{P}}} = \times_{(s,a) \times \mathbf{S} \times \mathbf{A}} \widehat{\underline{\mathcal{P}}}_{s,a}$, where $\widehat{\underline{\mathcal{P}}}_{s,a} = \left\{ (1-\xi)p + \xi \mathbf{1} e_{s_0}^\top : D(p \| \widehat{p}_{s,a}) \leq \delta \right\}$.
Solve the empirical DR average reward Bellman equation

$$v_{\widehat{\underline{\mathcal{P}}}}^*(s) = \max_{a \in \mathbf{A}} \left\{ r(s,a) + \Gamma_{\widehat{\underline{\mathcal{P}}}_{s,a}}(v_{\widehat{\underline{\mathcal{P}}}}^*) \right\} - g_{\widehat{\underline{\mathcal{P}}}}^*(s)$$

for a solution pair $(g_{\widehat{\underline{\mathcal{P}}}}^*, v_{\widehat{\underline{\mathcal{P}}}}^*)$.

Extract an optimal policy $\widehat{\pi}^* \in \Pi$ as $\widehat{\pi}^*(s) \in \arg\max_{a \in \mathbf{A}} \left\{ r(s,a) + \Gamma_{\widehat{\underline{\mathcal{P}}}_{s,a}}(v_{\widehat{\underline{\mathcal{P}}}}^*) \right\}$.
**return** $\widehat{\pi}^*, g_{\widehat{\underline{\mathcal{P}}}}^*$

---

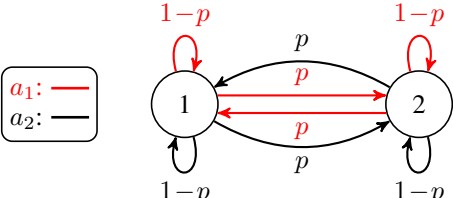

Figure 1: Transition diagram of the hard MDP instance in Wang et al. [37].

Next, combining Theorem D.1 with Lemma C.3, we establish that the policy error satisfies

$$\left\| V_{\mathcal{P}}^* - V_{\mathcal{P}}^{\widehat{\pi}^*} \right\|_\infty \leq \frac{2}{1-\gamma} \max_{\pi \in \Pi} \left\| \widehat{\mathcal{T}}_\gamma^\pi(V_{\mathcal{P}}^\pi) - \mathcal{T}_\gamma^\pi(V_{\mathcal{P}}^\pi) \right\|_\infty.$$

This reduces the analysis of the policy error to bounding the estimation error of the DR Bellman operator evaluated at $V_{\mathcal{P}}^\pi$. As the rewards cancel out, it remains to show that the DR functional applied to $V_{\mathcal{P}}^\pi$ satisfy appropriate concentration bound.

To this end, we apply the strong duality for the DR functional, the bound in Lemma C.6, and a Bernstein-type inequality to show that for any function $V$, the deviation satisfies

$$\left\| \Gamma_{\widehat{\mathcal{P}}_{s,a}}(V) - \Gamma_{\mathcal{P}_{s,a}}(V) \right\|_\infty \leq \widetilde{O}\left( \frac{\mathrm{Span}(V)}{\sqrt{n \mathfrak{p}_\wedge}} \right)$$

with high probability.

Finally, by selecting the parameters $\gamma = 1 - 1/\sqrt{n}$ and $\eta = 1/\sqrt{n}$, and noting that $\mathrm{Span}(V_{\mathcal{P}}^\pi) \leq O(t_{\mathrm{minorize}})$, we complete the proof for the KL-divergence case of Theorems 4.4 and 4.5. The argument under the $f_k$-divergence formulation proceeds in an analogous manner.

## 5   Numerical Experiments

In this section, we present numerical experiments to validate our theoretical results. We employ the Hard MDP family introduced in Wang et al. [37], which confirms a minimax sample complexity lower bound of $\Omega(t_{\mathrm{minorize}} \varepsilon^{-2})$ for estimating the average reward to within an $\varepsilon$ absolute error in the non-robust setting, matching the known upper bound. Our experiments show an empirical convergence rate of $n^{-1/2}$ for both algorithms, validating them as the first algorithms that achieve this rate in the DR-AMDP setting.

**Definition 5.1** (Hard MDP Family in Wang et al. [37]). This family of MDP instances has $\mathbf{S} = \{1,2\}$, $\mathbf{A} = \{1,2\}$, and reward function $r(1,\cdot) = 1$ and $r(2,\cdot) = 0$. The controlled transition kernel $P$ is parameterized by $\mathfrak{p}$ with transition diagram given in Figure 1.

Observe that under this controlled transition kernel, all stationary policies induce the same transition matrix $P_\pi$. Moreover, restricting $\mathfrak{p} \in (0, \frac{1}{2}]$ we have $P_\pi^m = (1 - (1 - 2\mathfrak{p})^m) \frac{1}{2} J + (1 - 2\mathfrak{p})^m I$, where

$J$ is the matrix of all 1 and $I$ is the identity matrix. Therefore, $P_{\pi_i}$ is $\left(m, (1 - (1 - 2\mathfrak{p})^m)\right)$-Doeblin. Thus, the minorization time of $P$ is $\inf_{m \geq 1} m/(1 - (1 - 2\mathfrak{p})^m) = \frac{1}{2\mathfrak{p}}$.

This example clarifies our use of $m_\vee$ in Definition 3.3: while $m_\pi \equiv 1$ for all $\mathfrak{p} \in (0, 1/2]$, the minorization time $t_{\text{minorize}}$ is unbounded, approaching infinity as $\mathfrak{p}$ goes to 0.

Next, we evaluate the performance of Algorithm 2 and 3 by analyzing their value approximation errors under both KL and $\chi^2$ uncertainty sets. $\chi^2$ is a special case of $f_k$-divergence with $k = 2$.

The sub-figures in Figure 2 presents the error achieved by the algorithms using a total of $n$ transition samples for every state-action pair. Each data point in the plots corresponds to *a single estimate* generated by one independent run of the corresponding algorithm. Then, we compute the $l_\infty$-error between the estimator and the ground-truth average-reward, which is computed via value iteration.

We then perform regression on data points on each MDP instance with the same parameter $\mathfrak{p}$. The plots demonstrate the error converging with rate $n^{-1/2}$, evidenced by the slope of $-1/2$ in Figure 2 on a log-log scale. We observe a remarkably low variance around the regression line of both algorithms, given that each data point is a single independent run of the corresponding algorithm.

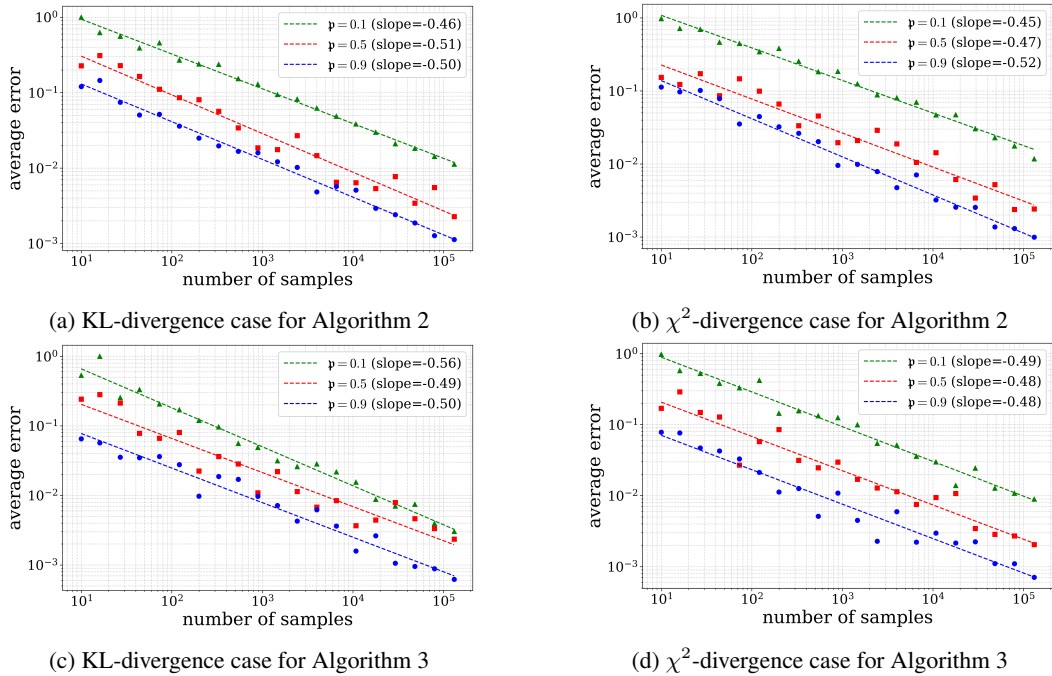

(a) KL-divergence case for Algorithm 2

(b) $\chi^2$-divergence case for Algorithm 2

(c) KL-divergence case for Algorithm 3

(d) $\chi^2$-divergence case for Algorithm 3

Figure 2: Comparative numerical experiments on (a-b) Algorithm 2 and (c-d) Algorithm 3 for the hard MDP instance, demonstrating $\varepsilon$-dependence under different divergence measures.

In addition to these experiment, we also perform a larger scale experiment to stress test our algorithm. Due to space limitations, the report is provided in Appendix H.

## 6 Conclusion and Future Work

In this work, we study distributionally robust average-reward reinforcement learning under a generative model. We first establish an instance-dependent bound of $\widetilde{O}(|\mathbf{S}||\mathbf{A}|t_{\text{minorize}}^2(1 - \gamma)^{-2}\varepsilon^{-2})$ for DR-DMDP. Building on this result, we propose two a priori knowledge-free algorithms with finite-sample complexity $\widetilde{O}(|\mathbf{S}||\mathbf{A}|t_{\text{minorize}}^2\varepsilon^{-2})$. Our work provides novel insights into the relationship between uniform ergodicity and sample complexity under distributional robustness.

While our results rely on the assumptions of uniform ergodicity and constraints on the uncertainty size, we acknowledge these as potential limitations. For future work, we plan to generalize these results to weakly communicating settings and, potentially, multichain MDPs, and investigate broader uncertainty sets (e.g., $l_p$-balls and Wasserstein metrics).

## Acknowledgement

N. Si gratefully acknowledges the support from the Hong Kong Research Grants Council [Theme-based Research Scheme T32-615/24-R].

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

# Appendices

## A  Notations and Basic Properties

In this section, we present the technical proof for DR-MDPs. Before introducing the theoretical foundations and analyzing related statistical properties, we first define key notations and auxiliary quantities to facilitate subsequent analysis.

For ARMDPs, it is useful to consider the *span semi-norm* Puterman [30]. For vector $u \in \mathbb{R}^d$, let $e = (1, \cdots, 1)^\top$, and define:

$$\begin{aligned}
\text{Span}(u) &:= \max_{1 \leq i \leq d}(u_i) - \min_{1 \leq i \leq d}(u_i) \\
&= 2 \inf_{c \in \mathbb{R}} \|u - ce\|_\infty.
\end{aligned} \tag{A.1}$$

Note that the span semi-norm satisfies the triangle inequality

$$\text{Span}(v_1 + v_2) \leq \text{Span}(v_1) + \text{Span}(v_2).$$

Our analysis relies extensively on two fundamental operators: the DR discounted policy Bellman operator $\mathcal{T}_\gamma^\pi$ and its optimal counterpart $\mathcal{T}_\gamma^*$. These operators are defined as follows:

$$\mathcal{T}_\gamma^\pi(V)(s) := \sum_{a \in \mathbf{A}} \pi(s|a)\left(r(s,a) + \gamma \Gamma_{\mathcal{P}_{s,a}}(V)\right) \tag{A.2}$$

$$\mathcal{T}_\gamma^*(V)(s) := \max_{a \in \mathbf{A}} \left\{r(s,a) + \gamma \Gamma_{\mathcal{P}_{s,a}}(V)\right\}. \tag{A.3}$$

Similarly, we define the empirical DR discounted policy operator $\widehat{\mathcal{T}}_\gamma^\pi$ and its optimal counterpart $\widehat{\mathcal{T}}_\gamma^*$ as:

$$\widehat{\mathcal{T}}_\gamma^\pi(V)(s) := \sum_{a \in \mathbf{A}} \pi(s|a)\left(r(s,a) + \gamma \Gamma_{\widehat{\mathcal{P}}_{s,a}}(V)\right) \tag{A.4}$$

$$\widehat{\mathcal{T}}_\gamma^*(V)(s) := \max_{a \in \mathbf{A}} \left\{r(s,a) + \gamma \Gamma_{\widehat{\mathcal{P}}_{s,a}}(V)\right\}. \tag{A.5}$$

It has been shown that the DR value function $V_\mathcal{P}^\pi$ is the unique fixed-point of the DR discounted policy operator (A.2), a.k.a. $V_\mathcal{P}^\pi$ is the solution to the DR discounted Bellman equation: $V_\mathcal{P}^\pi = \mathcal{T}_\gamma^\pi(V_\mathcal{P}^\pi)$ Iyengar [16], Puterman [30], Nilim and El Ghaoui [27].

We introduce some technical notations. For function $v : \mathbf{S} \to \mathbb{R}$, let

$$p[v] := \sum_{s \in \mathbf{S}} p(s)v(s)$$

Notice that with the above notation, we simplify the expectation as $E_p[v] = p[v]$.

For probability measure $p, q \in \Delta(\mathbf{S})$, we say that $p$ is absolutely continuous w.r.t. $q$, denoted by $p \ll q$, if $q(s) = 0$ implies that $p(s) = 0$. If $p \ll q$, we define the likelihood ratio, a.k.a. Radon-Nikodym derivative,

$$\frac{p}{q}(s) := \frac{p(s)}{q(s)} \text{ if } q(s) > 0, \text{ else } 0.$$

We say that $p$ and $q$ are mutually absolutely continuous, denoted by $p \sim q$ if $p \ll q$ and $q \ll p$.

For $p \in \Delta(\mathbf{S})$, we also define the $L^\infty(p)$ norm of a function $v : \mathbf{S} \to \mathbb{R}$ by

$$\|v\|_{L^\infty(p)} := \operatorname*{ess\,sup}_p |v| = \max_{s : p(s) > 0} |v(s)|.$$

In the DR setting, given uncertainty set $\mathcal{P}_{s,a}$ and a function $V : \mathbf{S} \to \mathbb{R}$, we say $p^*$ is a worst-case measure if

$$p^*[V] = \inf_{p \in \mathcal{P}_{s,a}} p[V]$$

Also, recall Definition 4.1, we define the minimal support probability $\mathfrak{p}_\wedge$ in measuring the samples required.

$$\mathfrak{p}_\wedge := \min_{(s,a)\in\mathbf{S}\times\mathbf{A}} \left\{ \min_{s'\in\mathbf{S}} p_{s,a}(s') : p_{s,a}(s') > 0 \right\}$$

The sample complexity's dependence on $\mathfrak{p}_\wedge$ emerges from two theoretical requirements. First, accurate estimation of the worst-case transition kernel demands that samples capture the distribution's support, necessitating at least $\Omega(1/\mathfrak{p}_\wedge)$ samples to ensure all non-zero probability transitions are observed. Second, the perturbed transition kernel needs to preserve certain mixing characteristics. This is crucial for us to establish a uniform high probability bound on the minorization times of the controlled kernels in the uncertainty set.

Specifically, we consider the "good events" set $\Omega_{n,d}$, as the collection of empirical measures that remain sufficiently close to the nominal transition kernel $P$. Recall from (4.1) that

$$\widehat{p}_{s,a}(s') := \frac{1}{n}\sum_{i=1}^{n} \mathbb{1}\left\{ S_{s,a}^{(i)} = s' \right\}.$$

For any $d > 0$ we define,

$$\Omega_{n,d}(p_{s,a}) := \left\{ \omega : \left\| \frac{\widehat{p}_{s,a} - p_{s,a}}{p_{s,a}} \right\|_{L^\infty(p_{s,a})} \leq d \right\} \tag{A.6}$$

as the relative difference between $\widehat{p}_{s,a}$ and $p_{s,a}$ is close up to $d$. Then, define:

$$\Omega_{n,d} := \bigcap_{(s,a)\in\mathbf{S}\times\mathbf{A}} \Omega_{n,d}(p_{s,a}). \tag{A.7}$$

**Theorem A.1** (Bernstein's inequality, Theorem 3 in Boucheron et al. [4]). *Let $X_1, X_2, \ldots, X_n$ be independent random variables with $E[X_i] = \mu$ and $|X_i - \mu| \leq M$ almost surely. Then we have:*

$$\left| \frac{1}{n}\sum_{i=1}^{n} X_i - \mu \right| \leq \frac{M}{3n}\log\frac{2}{\beta} + \sqrt{\frac{2\mathbf{Var}(X)}{n}\log\frac{2}{\beta}}. \tag{A.8}$$

*with probability at least $1 - \beta$.*

By Bernstein's inequality, we could bound the probability measure of $\Omega_{n,d}$:

**Lemma A.2.** *When the relative difference $d$ satisfies:*

$$d = \frac{1}{3n\mathfrak{p}_\wedge}\log\frac{2|\mathbf{S}|^2|\mathbf{A}|}{\beta} + \sqrt{\frac{2}{n\mathfrak{p}_\wedge}\log\frac{2|\mathbf{S}|^2|\mathbf{A}|}{\beta}}$$

*then*

$$P(\Omega_{n,d}^c) \leq \beta$$

*Proof.* Let $\mathrm{supp}(p_{s,a}) := \{s' : p_{s,a}(s') > 0\}$. Given $n$ i.i.d. samples $\left\{ S_{s,a}^{(1)}, S_{s,a}^{(2)}, \cdots, S_{s,a}^{(n)} \right\}$ drawn from $p_{s,a}$. We define the indicator variables:

$$X_{s,a}^i(s') = \mathbb{1}\left\{ S_{s,a}^{(i)} = s' \right\}.$$

Note that $X_{s,a}^i(s') \sim \mathrm{Bernoulli}(p_{s,a}(s'))$ for all $1 \leq i \leq n$. Then:

$$\widehat{p}_{s,a}(s') = \frac{1}{n}\sum_{i=1}^{n} X_{s,a}^i(s')$$

By Union bound, we have:

$$\begin{aligned}
P(\Omega_{n,d}^c) &\leq \sum_{(s,a)\in\mathbf{S}\times\mathbf{A}} P\left( \left\| \frac{\widehat{p}_{s,a} - p_{s,a}}{p_{s,a}} \right\|_{L^\infty(p_{s,a})} \geq d \right) \\
&\leq \sum_{(s,a)\in\mathbf{S}\times\mathbf{A}} \sum_{s'\in\mathrm{supp}(p_{s,a})} P\left( |\widehat{p}_{s,a}(s') - p_{s,a}(s')| \geq dp_{s,a}(s') \right) \\
&\leq \sum_{(s,a)\in\mathbf{S}\times\mathbf{A}} \sum_{s'\in\mathrm{supp}(p_{s,a})} P\left( \left| \frac{1}{n}\sum_{i=1}^{n} X_{s,a}^i(s') - p_{s,a}(s') \right| \geq dp_{s,a}(s') \right)
\end{aligned} \tag{A.9}$$

Then, by Bernstein's inequality (A.8), for any action-state pairs $(s,a) \in \mathbf{S} \times \mathbf{A}$, and next state $s' \in \mathrm{supp}(p_{s,a})$, since $E_{p_{s,a}}[X_{s,a}^i(s')] = p_{s,a}(s')$, and $|X_{s,a}^i(s') - p_{s,a}| \le 1$, we have:

$$\left| \frac{1}{n} \sum_{i=1}^{n} X_{s,a}^i(s') - p_{s,a}(s') \right| \le \frac{1}{3n} \log \frac{2|\mathbf{S}|^2|\mathbf{A}|}{\beta} + \sqrt{\frac{2\mathbf{Var}(\mathrm{Bernoulli}(p_{s,a}(s')))}{n} \log \frac{2|\mathbf{S}|^2|\mathbf{A}|}{\beta}}$$
(A.10)

with probability at least $1 - \frac{\beta}{|\mathbf{S}|^2|\mathbf{A}|}$. Thus, let

$$d = \frac{1}{3n\mathfrak{p}_\wedge} \log \frac{2|\mathbf{S}|^2|\mathbf{A}|}{\beta} + \sqrt{\frac{2}{n\mathfrak{p}_\wedge} \log \frac{2|\mathbf{S}|^2|\mathbf{A}|}{\beta}}$$

then, for all $(s,a) \in \mathbf{S} \times \mathbf{A}$, and $s' \in \mathrm{supp}(p_{s,a})$:

$$\begin{aligned}
dp_{s,a}(s') &= \frac{p_{s,a}(s')}{3n\mathfrak{p}_\wedge} \log \frac{2|\mathbf{S}|^2|\mathbf{A}|}{\beta} + \sqrt{\frac{2p_{s,a}(s')}{n\mathfrak{p}_\wedge} \log \frac{2|\mathbf{S}|^2|\mathbf{A}|}{\beta}} \\
&\ge \frac{1}{3n} \log \frac{2|\mathbf{S}|^2|\mathbf{A}|}{\beta} + \sqrt{\frac{2p_{s,a}(s')}{n} \log \frac{2|\mathbf{S}|^2|\mathbf{A}|}{\beta}} \\
&\overset{(i)}{\ge} \frac{1}{3n} \log \frac{2|\mathbf{S}|^2|\mathbf{A}|}{\beta} + \sqrt{\frac{2\mathbf{Var}\left(\mathrm{Bernoulli}(p_{s,a}(s'))\right)}{n} \log \frac{2|\mathbf{S}|^2|\mathbf{A}|}{\beta}}
\end{aligned}$$
(A.11)

Where $(i)$ relies on $\mathbf{Var}(\mathrm{Bernoulli}(p_{s,a}(s'))) = p_{s,a}(s')(1 - p_{s,a}(s'))$, then we conclude, for all $(s,a) \in \mathbf{S} \times \mathbf{A}$ and $s' \in \mathrm{supp}(p_{s,a})$, with probability $1 - \frac{\beta}{|\mathbf{S}|^2|\mathbf{A}|}$, we have:

$$\left| \frac{1}{n} \sum_{i=1}^{n} X_{s,a}^i(s') - p_{s,a}(s') \right| \le dp_{s,a}(s').$$
(A.12)

Then we conclude that:

$$P \left( \left| \frac{1}{n} \sum_{i=1}^{n} X_{s,a}^i(s') - p_{s,a}(s') \right| \ge dp_{s,a}(s') \right) \le \frac{\beta}{|\mathbf{S}|^2|\mathbf{A}|}$$

And further:

$$\begin{aligned}
P(\Omega_{n,d}^c) &\le \sum_{(s,a) \in \mathbf{S} \times \mathbf{A}} \sum_{s' \in \mathrm{supp}(p_{s,a})} P \left( \left| \frac{1}{n} \sum_{i=1}^{n} X_{s,a}^i(s') - p_{s,a}(s') \right| \ge dp_{s,a}(s') \right) \\
&\le |\mathbf{S}|^2|\mathbf{A}| \cdot \frac{\beta}{|\mathbf{S}|^2|\mathbf{A}|} \\
&= \beta.
\end{aligned}$$
(A.13)

Proved. □

**Lemma A.3.** *Let the transition kernel $K$ be uniformly ergodic. If $t_{\mathrm{minorize}}(K) < \infty$, then there exists an $(m,p)$ pair, such that:*

$$\frac{m}{p} = t_{\mathrm{minorize}}(K)$$

*Proof.* By definition:

$$t_{\mathrm{minorize}}(K) := \inf \left\{ m/p : \min_{s \in \mathbf{S}} K^m(s, \cdot) \ge p\psi(\cdot) \text{ for some } \psi \in \Delta(\mathbf{S}) \right\}.$$

As $t_{\mathrm{minorize}}(K) < \infty$, then, there exists a constant $C > 0$, such that

$$t_{\mathrm{minorize}} \le C.$$

As the feasible $(m,p)$-pair such that:

$$(m,p) \in \left\{ (m,p) : \min_{s \in \mathbf{S}} K^m \ge p\psi(\cdot) \text{ for some } \psi \in \Delta(\mathbf{S}) \right\},$$

$\frac{m}{p} \leq C$, then $m \leq Cp \leq C$ because $p \in (0, 1]$, we conclude:

$$m \in \mathcal{C} := \{1, 2, \cdots, \lfloor C \rfloor\} \quad \text{and} \quad |\mathcal{C}| < \infty.$$

Define $\mathcal{C}_m$ and $p_{\max}(m)$ as:

$$\mathcal{C}_m = \{p : \exists \psi \in \Delta(\mathbf{S}), K^m(s, \cdot) \geq p\psi(\cdot)\}$$
$$p_{\max}(m) = \sup_{p \in \mathcal{C}_m} p \tag{A.14}$$

With this definition, note that

$$t_{\text{minorize}}(K) = \inf_{m \in \mathcal{C}, p \in \mathcal{C}_m} \frac{m}{p}.$$

We will show that the set $\mathcal{C}_m$ is closed, hence $p_{\max}(m) \in \mathcal{C}_m$ is achieved.

Since $\mathbf{S}$ is finite, $\Delta(\mathbf{S}) \subset \mathbb{R}^{|\mathbf{S}|}$ is compact. Consider any sequence $\{p_n\} \subseteq \mathcal{C}_m$ such that $p_n \to p$. Then, there exists $\psi_n(\mathbf{S})$, such that

$$K^m(s, \cdot) \geq p_n \psi_n(\cdot), \quad s \in \mathbf{S}.$$

As $\Delta(\mathbf{S})$ is compact, sequence $\{\psi_n\}$ has subsequence $\{\psi_{n_k}\}$, such that:

$$\psi_{n_k} \to \psi \in \Delta(\mathbf{S}) \quad \text{pointwise.}$$

and a corresponding $\{p_{n_k}\}$ such that $p_{n_k} \to p$. Then for any $(s, s') \in \mathbf{S} \times \mathbf{S}$:

$$K^m(s, s') \geq p_{n_k} \cdot \psi_{n_k}(s'), \quad \forall k$$

We have $p_{n_k} \to p$ and $\psi_{n_k}(s') \to \psi(s')$:

$$p_{n_k} \cdot \psi_{n_k}(s') \to p \cdot \psi(s').$$

Thus:

$$K^m(s, \cdot) \geq p\psi(\cdot), \quad \forall s \in \mathbf{S};$$

i.e. $p \in \mathcal{C}_m$. Hence, $\mathcal{C}_m$ is a closed and $p_{\max}(m) \in \mathcal{C}_m$.

Therefore,

$$t_{\text{minorize}}(K) = \inf_{m \in \mathcal{C}, p \in \mathcal{C}_m} \frac{m}{p} = \min_{m \in \mathcal{C}} \frac{m}{p_{\max}(m^*)}.$$

Since $\mathcal{C}$ is finite, there exists a $m^* \in \{1, 2, \cdots, \lfloor C \rfloor\}$ such that

$$t_{\text{minorize}}(K) = \frac{m^*}{p_{\max}(m^*)}.$$

$\square$

**Lemma A.4.** *Suppose the controlled transition kernel $P$ is uniformly ergodic. If for all $Q \in \mathcal{P}$, $p_{s,a} \ll q_{s,a}$ holds for all $(s, a) \in \mathbf{S} \times \mathbf{A}$, then $\mathcal{P}$ is uniformly ergodic.*

*Proof.* Since $P$ is a uniformly ergodic, then for all $\pi \in \Pi$, $t_{\text{minorize}}(P_\pi)$, by Lemma A.3 there exists $(m_\pi, p_\pi)$ such that:

$$\frac{m_\pi}{p_\pi} = t_{\text{minorize}}(P_\pi).$$

For any $Q \in \mathcal{P}$ and state pair $(s_0, s_{m_\pi}) \in \mathbf{S} \times \mathbf{S}$, the $m_\pi$-step transition probability of $P_\pi$, and $Q_\pi$ can be expressed as:

$$P_\pi^{m_\pi}(s_0, s_{m_\pi}) = \sum_{s_1, s_2, \cdots, s_{m_\pi - 1}} p_{s_0, \pi(s_0)}(s_1) p_{s_1, \pi(s_1)}(s_2) \cdots p_{s_{m_\pi - 1}, \pi(s_{m_\pi - 1})}(s_{m_\pi})$$

$$Q_\pi^{m_\pi}(s_0, s_{m_\pi}) = \sum_{s_1, s_2, \cdots, s_{m_\pi - 1}} q_{s_0, \pi(s_0)}(s_1) q_{s_1, \pi(s_1)}(s_2) \cdots q_{s_{m_\pi - 1}, \pi(s_{m_\pi - 1})}(s_{m_\pi}). \tag{A.15}$$

Define:

$$\mathcal{U}(s_0, s_{m_\pi}) := \left\{ (s_1, s_2, \cdots, s_{m_\pi - 1}) \in \mathbf{S}^{m_\pi - 1} : p_{s_i, \pi(s_i)}(s_{i+1}) > 0, 0 \leq i \leq m_\pi - 1 \right\}.$$

Since $p_{s,a} \ll q_{s,a}$ holds for all $(s,a)$ pairs, we have for any tuple $(s_1, s_2, \cdots, s_{m_\pi - 1}) \in \mathcal{U}(s_0, s_{m_\pi})$,

$$\prod_{i=1}^{m_\pi - 1} p_{s_i, \pi(s_i)}(s_{i+1}) > 0,$$

by absolute continuity, it implies:

$$\prod_{i=1}^{m_\pi - 1} q_{s_i, \pi(s_i)}(s_{i+1}) > 0$$

Then:

$$\begin{aligned}
&Q_\pi^{m_\pi}(s_0, s_{m_\pi}) \\
&= \sum_{s_1, s_2, \cdots, s_{m_\pi - 1}} q_{s_0, \pi(s_0)}(s_1) q_{s_1, \pi(s_1)}(s_2) \cdots q_{s_{m_\pi - 1}, \pi(s_{m_\pi - 1})}(s_{m_\pi}) \\
&\geq \sum_{(s_1, s_2, \cdots, s_{m_\pi - 1}) \in \mathcal{U}(s_0, s_{m_\pi})} q_{s_0, \pi(s_0)}(s_1) q_{s_1, \pi(s_1)}(s_2) \cdots q_{s_{m_\pi - 1}, \pi(s_{m_\pi - 1})}(s_{m_\pi}) \\
&= \sum_{(s_1, s_2, \cdots, s_{m_\pi - 1}) \in \mathcal{U}(s_0, s_{m_\pi})} \left( \frac{q_{s_0, \pi(s_0)}}{p_{s_0, \pi(s_0)}}(s_1) \right) p_{s_0, \pi(s_0)}(s_1) \left( \frac{q_{s_1, \pi(s_1)}}{p_{s_1, \pi(s_1)}}(s_2) \right) p_{s_1, \pi(s_1)}(s_2) \\
&\quad \cdots \left( \frac{q_{s_{m_\pi - 1}, \pi(s_{m_\pi - 1})}}{p_{s_{m_\pi - 1}, \pi(s_{m_\pi - 1})}}(s_{m_\pi}) \right) p_{s_{m_\pi - 1}, \pi(s_{m_\pi - 1})}(s_{m_\pi}) \\
&\geq c(s_0, s_{m_\pi}) \cdot \sum_{(s_1, s_2, \cdots, s_{m_\pi - 1}) \in \mathcal{U}(s_0, s_{m_\pi})} p_{s_0, \pi(s_0)}(s_1) p_{s_1, \pi(s_1)}(s_2) \cdots p_{s_{m_\pi - 1}, \pi(s_{m_\pi - 1})}(s_{m_\pi}) \\
&= c(s_0, s_{m_\pi}) \cdot P_\pi^{m_\pi}(s_0, s_{m_\pi})
\end{aligned}$$

(A.16)

where:

$$c(s_0, s_{m_\pi}) = \min_{(s_1, s_2, \cdots, s_{m_\pi - 1}) \in \mathcal{U}} \prod_{i=0}^{m_\pi - 1} \frac{q_{s_i, \pi(s_i)}}{p_{s_i, \pi(s_i)}}(s_{i+1}) > 0$$

Denote $c := \min_{(s,s') \in \mathbf{S} \times \mathbf{S}} c(s, s') > 0$, we conclude:

$$Q_\pi^{m_\pi}(s_0, s_{m_\pi}) \geq c P_\pi^{m_\pi}(s_0, s_{m_\pi}) \quad \text{for all} \quad (s_0, s_{m_\pi}) \in \mathbf{S} \times \mathbf{S}.$$

Since $P_\pi$ satisfies $(m_\pi, p_\pi)$-Doeblin condition, for some $\psi \in \Delta(\mathbf{S})$, we have:

$$Q_\pi^{m_\pi}(s_0, s_{m_\pi}) \geq c P_\pi^{m_\pi}(s_0, s_{m_\pi}) \geq c p_\pi \psi(s_0) \quad \text{for all} \quad (s_0, s_{m_\pi}) \in \mathbf{S} \times \mathbf{S}$$

$Q_\pi$ satisfies $(m_\pi, q_\pi)$-Doeblin condition where $q_\pi = c p_\pi$. Thus, we concludes for all $Q \in \mathcal{P}$ and $\pi \in \Pi$, $Q_\pi$ satisfies $(m_\pi, q_\pi)$-Doeblin condition, and:

$$t_{\text{minorize}}(Q_\pi) \leq \frac{m_\pi}{c q_\pi} < \infty \quad \text{for all} \quad \pi \in \Pi \implies \max_{\pi \in \Pi} t_{\text{minorize}}(Q_\pi) < \infty$$

Finally, we concludes, for all $Q \in \mathcal{P}$:

$$\max_{\pi \in \Pi} t_{\text{minorize}}(Q_\pi) < \infty.$$

$\mathcal{P}$ is uniformly ergodic. $\qquad\square$

Having established the uniformly ergodic property preservation under absolute continuity, we now introduce additional technical tools central to our analysis. Our proof strategy fundamentally relies on the span semi-norm framework for value functions in DMDPs, the following proposition from Wang et al. [37] formalizes this connection:

**Proposition A.5** (Proposition 6.1 in Wang et al. [37]). *Suppose $P_\pi$ satisfies $(m, p)$-Doeblin condition, and $V_P^\pi$ is the value function associate with kernel $P$ under policy $\pi$, then $\text{Span}(V_P^\pi) \leq 3m/p$*

Our core approach involves approximating the DR-AMDP through its DR-DMDP counterpart. To establish this connection rigorously, we require the following fundamental lemma that bridges discounted and average-reward value functions:

**Lemma A.6** (Lemma 1 in Wang et al. [38]). *Suppose $P_\pi$ satisfies $(m_\pi, p_\pi)$-Doeblin condition and $t_{\mathrm{minorize}}(P_\pi) < \infty$, then:*

$$\|(1-\gamma)V_P^\pi - g_P^\pi\|_\infty \le 9(1-\gamma)t_{\mathrm{minorize}}(P_\pi)$$

**Proposition A.7** (Restatement of Proposition 3.6). *If $\mathcal{P}$ is uniformly ergodic with a uniformly bounded minorization time, then $g_\mathcal{P}^*(s) \equiv g_\mathcal{P}^*$ is constant in $s \in \mathbf{S}$. Moreover, there exists a solution $(g, v)$ of $v(s) = \mathcal{T}^*(v)(s) - g^*(s)$ for all $s \in \mathbf{S}$ and any such solution satisfies $g(s) = g_\mathcal{P}^*$ for all $s \in \mathbf{S}$. Moreover, the policy $\pi^*(s) \in \arg\max_{a \in \mathbf{A}} \{r(s,a) + \Gamma_{\mathcal{P}_{s,a}}(v)\}$ achieves the optimal average-reward $g_\mathcal{P}^*$.*

*Proof.* Since $\mathcal{P}$ is uniformly ergodic with uniformly bounded minorization time, for any stationary policy $\pi$ the kernel $Q_\pi$ satisfies an $(m_{P_\pi}, p_{Q_\pi})$-Doeblin condition. Then, by Theorem UE in Wang et al. [37],

$$\sup_{s \in \mathcal{S}} \|Q_\pi^n(s, \cdot) - \rho_{Q_\pi}(\cdot)\|_{\mathrm{TV}} \le 2(1 - p_{Q_\pi})^{\lfloor n/m_{Q_\pi} \rfloor}, \qquad \forall n \ge 0,$$

where $\rho_{Q_\pi}$ denotes the unique stationary distribution of $Q_\pi$. Let $\overline{m} := \sup_{Q \in \mathcal{P}, \pi \in \Pi} m_{Q_\pi} < \infty$ and $\underline{p} := \inf_{Q \in \mathcal{P}, \pi \in \Pi} p_{Q_\pi} > 0$. Choose $s' \in \mathcal{S}$ such that $\rho_{Q_\pi}(s') \ge \frac{1}{|\mathcal{S}|}$. Let

$$J := \left\lceil \overline{m} \cdot \frac{\ln\big(1/(8|\mathcal{S}|)\big)}{\ln(1 - \underline{p})} \right\rceil.$$

Then, for any $s \in \mathcal{S}$,

$$Q_\pi^J(s, s') \ge \rho_{Q_\pi}(s') - 4(1 - p_{Q_\pi})^{\lfloor J/m_{Q_\pi} \rfloor} \ge \frac{1}{|\mathcal{S}|} - \frac{1}{2|\mathcal{S}|} \ge \frac{1}{2|\mathcal{S}|} > 0.$$

Now, for a fixed $s_0 \in \mathcal{S}$, consider the iteration

$$\begin{cases} v_{t+1} \leftarrow \mathcal{T}^*(\omega_t), \\ \omega_{t+1} \leftarrow v_{t+1} - v_{t+1}(s_0)\, e, \end{cases}$$

where $e$ is the all-ones vector. Combine the fact there exists a positive integer $J$ such that for all $Q \in \mathcal{P}$ and any stationary deterministic policy $\pi$, there exists a state $s' \in \mathbf{S}$, such that $Q_\pi^J(s, s') > 0$, applying Theorem 8 in Wang et al. [45], $(\omega_t, v_t)$ converges to a solution $(g, v)$ of

$$v(s) = \mathcal{T}^*(v)(s) - g(s), \qquad \forall s \in \mathcal{S},$$

which proves existence. Combining Theorem 7 in Wang et al. [45], we have that $g(s) = g_\mathcal{P}^*$ for all $s \in \mathcal{S}$ and

$$\pi^*(s) \in \arg\max_{a \in \mathcal{A}} \left\{ r(s,a) + \Gamma_{\mathcal{P}_{s,a}}(v) \right\}$$

achieves the optimal average reward $g_\mathcal{P}^*$. $\qquad\square$

# B   Uniform Ergodicity of the KL Uncertainty Set

In this section, we prove the uniform ergodic properties over $\mathcal{P}$ and $\widehat{\mathcal{P}}$ under Assumption 1 and 2. To achieve this, we establish the uniform Doeblin condition through a careful analysis of the Radon-Nikodym derivatives between perturbed and nominal transition kernels $q_{s,a}$ and $p_{s,a}$. we propose some concepts in facilitating to bound the Radon-Nikodym derivative between derivative between perturbed kernel $q_{s,a}$ and nominal $p_{s,a}$.

**Proposition B.1.** *Suppose $\delta \le \frac{1}{8m_\vee^2}\mathfrak{p}_\wedge$, then for all $q_{s,a} \in \mathcal{P}_{s,a}$, the Radon-Nikodym derivative satisfies*

$$\left\| \frac{q_{s,a}}{p_{s,a}} \right\|_{L^\infty(p_{s,a})} \ge 1 - \frac{1}{2m_\vee}$$

*holds for all $(s,a) \in \mathbf{S} \times \mathbf{A}$*

*Proof.* Consider $q_{s,a} \in \mathcal{P}_{s,a}$, then with KL-constraint, for any $s' \in \mathbf{S}$, where $p_{s,a}(s') > 0$, we have:

$$
\begin{aligned}
\delta &\geq D_{\mathrm{KL}}\left(q_{s,a}\|p_{s,a}\right) \\
&= q_{s,a}\left[\log \frac{q_{s,a}}{p_{s,a}}\right] \\
&\geq q_{s,a}(s') \log \left(\frac{q_{s,a}(s')}{p_{s,a}(s')}\right) + (1 - q_{s,a}(s')) \log \left(\frac{1 - q_{s,a}(s')}{1 - p_{s,a}(s')}\right)
\end{aligned}
\tag{B.1}
$$

While the last inequality is derived by log-sum inequality. Let:

$$
h(x, y) = x \log \left(\frac{x}{y}\right) + (1 - x) \log \left(\frac{1 - x}{1 - y}\right)
$$

where $y \in [\mathbb{p}_\wedge, 1 - \mathbb{p}_\wedge]$, since:

$$
\frac{\partial^2 h(x, y)}{\partial x^2} = \frac{1}{x} + \frac{1}{1 - x}
$$

And $h(y, y) = 0$, we know for any fixed $y$, $h(x, y)$ is convex with respect to $x$ on $x \in (0, y)$. Since $h(0, y) = \log\left(\frac{1}{1-y}\right) \geq \log\left(\frac{1}{1-\mathbb{p}_\wedge}\right) \geq \mathbb{p}_\wedge > \delta$. By mean value theorem there exists a unique $x^*(y)$ s.t. $h(x^*(y), y) = \delta$. Hence, define $x^*(y) := \min_{x \in (0,y)}\{x : h(x, y) = \delta\}$, and for any fixed $t \in (0, 1)$, if $x^*(y) < ty$, then:

$$
\begin{aligned}
\delta \geq h(x^*(y), y) &> h(ty, y) \\
&= ty \log t + (1 - ty) \log \left(\frac{1 - ty}{1 - y}\right) \\
&\overset{(1)}{\geq} (1 - t + t \log t) y \\
&\overset{(2)}{>} \frac{(1 - t)^2}{2} y
\end{aligned}
\tag{B.2}
$$

Here (1) refers to the $h(ty, y)$ expansion at $y = 0$,

$$
\begin{aligned}
\lim_{y \to 0} h(ty, y)/y &= 1 - t + t \log t \\
\frac{\partial^2 h(ty, y)}{\partial y^2} &\geq 0
\end{aligned}
$$

then

$$
h(ty, y) \geq (1 - t + t \log t) y
$$

And (2) refers to the fact that

$$
1 - t + t \log t \geq \frac{(1 - t)^2}{2} \quad \text{on} \quad t \in (0, 1)
$$

The above functional dependence is optimal in polynomial. Hence, when $t = 1 - \frac{1}{2m_\vee}$, we have:

$$
\delta \geq h(x^*(y), y) \geq h\left(\left(1 - \frac{1}{2m_\vee}\right) y, y\right) > \frac{1}{8m_\vee^2} y \geq \frac{1}{8m_\vee^2} \mathbb{p}_\wedge
$$

However, under the assumption that $\delta \leq \frac{1}{8m_\vee^2} \mathbb{p}_\wedge$, the preceding inequality leads to a contradiction. As $y \in [\mathbb{p}_\wedge, 1 - \mathbb{p}_\wedge]$, let $y = p_{s,a}(s')$, we establish the uniform lower bound:

$$
q_{s,a}(s') \geq \left(1 - \frac{1}{2m_\vee}\right) p_{s,a}(s')
$$

This inequality holds uniformly across all:

- State-action pairs $(s, a) \in \mathbf{S} \times \mathbf{A}$
- Next states $s' \in \operatorname{supp}(p_{s,a})$

Consequently, the Radon-Nikodym derivative admits a uniform lower bound over the uncertainty set $\mathcal{P}_{s,a}$:

$$\inf_{q_{s,a} \in \mathcal{P}_{s,a}} \left\| \frac{q_{s,a}}{p_{s,a}} \right\|_{L^\infty(p_{s,a})} \geq 1 - \frac{1}{2m_\vee},$$

for all $(s, a) \in \mathbf{S} \times \mathbf{A}$. $\qquad\square$

Followed by the boundedness of the Radon-Nikodym derivative, we are able to show the uniform ergodic properties on the uncertainty $\mathcal{P}$

**Proposition B.2** (Restatement of the KL case in Proposition 4.2)**.** *Suppose $P$ is uniformly ergodic, and $\delta \leq \frac{1}{8m_\vee^2}\mathfrak{p}_\wedge$, then $\mathcal{P} = \mathcal{P}(D_{\mathrm{KL}}, \delta)$ is uniformly ergodic and for all $Q \in \mathcal{P}$ and $\pi \in \Pi$:*

$$t_{\mathrm{minorize}}(Q_\pi) \leq 2t_{\mathrm{minorize}}, \tag{B.3}$$

*where $t_{\mathrm{minorize}}$ is from Assumption 1.*

*Proof.* By Lemma A.3, since $P$ is uniformly ergodic, then there exists an $(m_\pi, p_\pi)$ pair, such that:

$$\frac{m_\pi}{p_\pi} = t_{\mathrm{minorize}}(P_\pi) \quad \text{and} \quad m_\pi \leq m_\vee$$

For all $Q \in \mathcal{P}$, by Proposition B.1, we have for all $(s, a) \in \mathbf{S} \times \mathbf{A}$,

$$\left\| \frac{q_{s,a}}{p_{s,a}} \right\|_{L^\infty(p_{s,a})} \geq 1 - \frac{1}{2m_\vee}$$

Then, for all state pairs $(s_0, s_{m_\pi}) \in \mathbf{S} \times \mathbf{S}$, consider $Q_\pi^{m_\pi}$, we have:

$$
\begin{aligned}
Q_\pi^{m_\pi}(s_0, s_{m_\pi}) &\geq \sum_{s_1, s_2, \cdots, s_{m_\pi-1}} q_{s_0, \pi(s_0)}(s_1) q_{s_1, \pi(s_1)}(s_2) \cdots q_{s_{m_\vee-1}, \pi(s_{m_\pi-1})}(s_{m_\pi}) \\
&\geq \sum_{s_1, s_2, \cdots, s_{m_\pi-1}} p_{s_0, \pi(s_0)}(s_1)(1 - \frac{1}{2m_\vee}) p_{s_1, \pi(s_1)}(s_2)(1 - \frac{1}{2m_\vee}) \cdots \\
&\quad p_{s_{m_\pi-1}, \pi(s_{m_\pi-1})}(s_{m_\pi})(1 - \frac{1}{2m_\vee}) \\
&\geq \left(1 - \frac{1}{2m_\vee}\right)^{m_\pi} P_\pi^{m_\pi}(s_0, s_{m_\pi}) \\
&\overset{(1)}{\geq} \frac{1}{2} P_\pi^{m_\pi}(s_0, s_{m_\pi}) \\
&\overset{(2)}{\geq} \frac{p_\pi}{2} \psi(s_{m_\pi}).
\end{aligned}
\tag{B.4}
$$

The inequality $(1)$ follows from:

$$\left(1 - \frac{1}{2m_\vee}\right)^m \geq \left(1 - \frac{1}{2m_\pi}\right)^{m_\pi} \geq \frac{1}{2}.$$

$(1)$ holds. The result $(2)$ is derived from the $(m_\pi, p_\pi)$-Doeblin condition satisfied by $P_\pi$. This implies that for every policy $\pi \in \Pi$, the perturbed kernel $Q_\pi$ maintains a $(m_\pi, \frac{p_\pi}{2})$-Doeblin condition. Crucially, this conclusion holds uniformly across all policies in $\Pi$. Furthermore, the minorization time satisfies:

$$t_{\mathrm{minorize}}(Q_\pi) \leq \frac{m_\pi}{\frac{p_\pi}{2}} \leq 2t_{\mathrm{minorize}}(P_\pi) \leq 2t_{\mathrm{minorize}},$$

where $t_{\mathrm{minorize}} = \max_{\pi \in \Pi} t_{\mathrm{minorize}}(P_\pi)$ by Assumption 1. Thus $t_{\mathrm{minorize}}(Q_\pi)$ is uniformly bounded over $Q \in \mathcal{P}$ and $\pi \in \Pi$:

$$\sup_{Q \in \mathcal{P}} \max_{\pi \in \Pi} t_{\mathrm{minorize}}(Q_\pi) < 2t_{\mathrm{minorize}} < \infty,$$

$\mathcal{P}$ is uniformly ergodic. $\qquad\square$

Building upon Proposition B.2, we establish that all perturbed transition kernels $Q \in \mathcal{P}$, $Q_\pi$ satifies the $(m_\pi, \frac{p_\pi}{2})$-Doeblin condition, and $\mathcal{P}$ preserves uniformly ergodic given $P$ is uniformly ergodic with appropriate adversarial power constraint. We further extends the uniform ergodicity to empirical kernels $\widehat{P}_\pi$ and empirical uncerainty sets $\widehat{P}$. Although these results are not essential for proving our main theorems, they provide valuable methodological insights for uniform ergodicity theory:

(i) Offering a technical blueprint for extending classical ergodic theory to DR settings

(ii) Laying the theoretical foundation for analyzing uncertainty sets in Markov models

(iii) Opening new research directions for perturbation analysis of ergodic processes.

These findings may prove particularly useful for future studies in DR-MDP and related ares.

**Lemma B.3.** *Suppose $P$ is uniformly ergodic. When the sample size:*

$$n \geq \frac{32m_\vee^2}{\mathfrak{p}_\wedge} \log \frac{2|\mathbf{S}|^2|\mathbf{A}|}{\beta}$$

*the empirical nominal transition kernel $\widehat{P}$ is also uniformly ergodic with:*

$$\max_{\pi \in \Pi} t_{\mathrm{minorize}}(\widehat{P}_\pi) \leq 2t_{\mathrm{minorize}}$$

*on the event set $\Omega_{n, \frac{1}{2m_\vee}}$, and*

$$P\left(\Omega_{n, \frac{1}{2m_\vee}}\right) \geq 1 - \beta$$

*Proof.* When the sample size satisfies:

$$n \geq \frac{32m_\vee^2}{\mathfrak{p}_\wedge} \log \frac{2|\mathbf{S}|^2|\mathbf{A}|}{\beta}$$

then:

$$
\begin{aligned}
&\frac{1}{3n\mathfrak{p}_\wedge} \log \frac{2|\mathbf{S}|^2|\mathbf{A}|}{\beta} + \sqrt{\frac{2}{n\mathfrak{p}_\wedge} \log \frac{2|\mathbf{S}|^2|\mathbf{A}|}{\beta}} \\
&\leq \frac{1}{48m_\vee^2} + \frac{1}{4m_\vee} \\
&\leq \frac{1}{2m_\vee}
\end{aligned}
\tag{B.5}
$$

Then by Bernsteins' inequality and A.2:

$$\frac{1}{n\mathfrak{p}_\wedge} \log \frac{2|\mathbf{S}|^2|\mathbf{A}|}{\beta} + \sqrt{\frac{2}{n\mathfrak{p}_\wedge} \log \frac{2|\mathbf{S}|^2|\mathbf{A}|}{\beta}} \leq \frac{1}{2m_\vee}$$

the probability of the event set $\Omega_{n, \frac{1}{2m_\vee}}$ is bounded by:

$$P\left(\Omega_{n, \frac{1}{2m_\vee}}\right) \geq 1 - \beta$$

For any fixed $(s, a) \in \mathbf{S} \times \mathbf{A}$, and $d \in [0, 1]$ on the event of $\Omega_{n,d}(p_{s,a})$, we have, for all $s' \in \mathrm{supp}(p_{s,a})$:

$$\widehat{p}_{s,a}(s') \geq p_{s,a}(s') - \frac{1}{2m_\vee} p_{s,a}(s') \geq \left(1 - \frac{1}{2m_\vee}\right) p_{s,a}(s')$$

Then, by Lemma A.3, for any $\pi \in \Pi$, there exists a $(m_\pi, p_\pi)$ pair such that $\frac{m_\pi}{p_\pi} = t_{\text{minorize}}(P_\pi)$. Consider the transition matrix $\widehat{P}_\pi$, we have, for all $(s, s') \in \mathbf{S} \times \mathbf{S}$:

$$
\begin{aligned}
\widehat{P}_\pi^{m_\pi}(s, s') &= \sum_{(s_1, s_2, \cdots, s_{m-1}) \in |\mathbf{S}|^{m_\pi - 1}} \widehat{p}_{s, \pi(s)}(s_1) \widehat{p}_{s_1, \pi(s_1)}(s_2) \cdots \widehat{p}_{s_{m-1}, \pi(s_{m_\pi-1})}(s') \\
&\geq \left(1 - \frac{1}{2m_\vee}\right)^{m_\pi} \sum_{(s_1, s_2, \cdots, s_{m_\pi-1}) \in |\mathbf{S}|^{m_\pi - 1}} p_{s, \pi(s)}(s_1) p_{s_1, \pi(s_1)}(s_2) \cdots p_{s_{m_\pi-1}, \pi(s_{m_\pi-1})}(s') \\
&\geq \left(1 - \frac{1}{2m_\vee}\right)^{m_\pi} P_\pi^m(s, s') \\
&\geq \frac{p_\pi}{2} \psi(s')
\end{aligned}
$$

(B.6)

which implies $\widehat{P}_\pi$ satisfies the $\left(m_\pi, \frac{p_\pi}{2}\right)$-Doeblin Condition, and:

$$
t_{\text{minorize}}(\widehat{P}_\pi) \leq \frac{m_\pi}{\frac{p_\pi}{2}} = 2t_{\text{minorize}}(P_\pi) \leq 2t_{\text{minorize}}.
$$

$\widehat{P}$ is uniformly ergodic with probability $1 - \beta$. $\qquad\square$

Combining Proposition B.2 with Lemma B.3, we establish that for all empirical transition kernels $\widehat{Q} \in \widehat{\mathcal{P}}$, the minorization time satisfies

$$
t_{\text{minorize}}(\widehat{Q}_\pi) \leq 4t_{\text{minorize}}.
$$

This bound yields the following immediate Corollary.

**Corollary B.4.** *Suppose the nominal transition kernel $P$ is uniformly ergodic, and $\delta \leq \frac{1}{16m_\vee^2} \mathfrak{p}_\wedge$, then when the sample complexity*

$$
n \geq \frac{32m_\vee^2}{\mathfrak{p}_\wedge} \log \frac{2|\mathbf{S}|^2|\mathbf{A}|}{\beta}
$$

*the empirical uncertainty set $\widehat{P}$ is uniformly ergodic and satisfies: following holds with probability $1 - \beta$*

$$
\sup_{Q \in \mathcal{P}} \max_{\pi \in \Pi} t_{\text{minorize}}(\widehat{Q}_\pi) \leq 4t_{\text{minorize}}
$$

*with probability $1 - \beta$.*

*Proof.* First, by Lemma A.3, for any $\pi \in \Pi$, there exists a $(m_\pi, p_\pi)$ pair such that:

$$
\frac{m_\pi}{p_\pi} = t_{\text{minorize}}(P_\pi).
$$

Then, consider

$$
\widehat{\mathfrak{p}}_\wedge := \min_{(s,a) \in \mathbf{S} \times \mathbf{A}} \left\{ \min_{s' \in \text{supp}(p_{s,a})} \widehat{p}_{s,a}(s') \right\}
$$

by Lemma B.3, when

$$
n \geq \frac{32m_\vee^2}{\mathfrak{p}_\wedge} \log \frac{2|\mathbf{S}|^2|\mathbf{A}|}{\beta}
$$

$\widehat{\mathfrak{p}}_\wedge \geq \left(1 - \frac{1}{2m_\vee}\right) \mathfrak{p}_\wedge \geq \frac{1}{2}\mathfrak{p}_\wedge$, and $\widehat{P}_\pi$ satisfies $\left(m_\pi, \frac{p_\pi}{2}\right)$-Doeblin condition with probability $1 - \beta$. Then, by Proposition B.2, as

$$
\delta \leq \frac{1}{16m_\vee^2} \mathfrak{p}_\wedge \leq \frac{1}{8m_\vee^2} \widehat{\mathfrak{p}}_\wedge.
$$

It implies for all $\widehat{Q} \in \widehat{\mathcal{P}}$, $\widehat{Q}_\pi$ satisfies $\left(m_\pi, \frac{p_\pi}{4}\right)$-Doeblin condition, and:

$$
t_{\text{minorize}}(\widehat{Q}_\pi) \leq \frac{4m_\pi}{p_\pi} \leq 4t_{\text{minorize}}(P_\pi) \leq 4t_{\text{minorize}},
$$

$\widehat{\mathcal{P}}$ is uniformly ergodic with probability $1 - \beta$. $\qquad\square$

**Theorem B.5.** *Under Assumptions 1 and when $\delta \leq \frac{1}{16m_\vee^2}\mathfrak{p}_\wedge$, when sample size satisfies:*

$$n \geq \frac{32m_\vee^2}{\mathfrak{p}_\wedge} \log \frac{2|\mathbf{S}|^2|\mathbf{A}|}{\beta},$$

*then,*

- *$\mathcal{P}$ is uniformly ergodic, and for any $Q \in \mathcal{P}$ and $\pi \in \Pi$, the minorization time of $Q_\pi$:*

$$t_{\text{minorize}}(Q_\pi) \leq 2t_{\text{minorize}}$$

  *with probability $1$.*

- *$\widehat{P}$ is uniformly ergodic and for any $\pi \in \Pi$, the minorization time of $\widehat{P}_\pi$:*

$$t_{\text{minorize}}(\widehat{P}_\pi) \leq 2t_{\text{minorize}}.$$

  *with probability $1 - \beta$.*

- *$\widehat{\mathcal{P}}$ is uniformly ergodic, and for any $\widehat{Q} \in \widehat{\mathcal{P}}$, the minorization time of $\widehat{Q}_\pi$:*

$$t_{\text{minorize}}(\widehat{Q}_\pi) \leq 4t_{\text{minorize}}.$$

  *with probability $1 - \beta$.*

*Proof.* The result follows by synthesizing three key components:

1. The uniform Doeblin condition for KL-constrained uncertainty set (Proposition B.2)

2. The Doeblin condition for empirical transition kernel (Lemma B.3)

3. The uniform Doeblin condition for empirical KL-constrained uncertainty set (Corollary B.4)

The combination of 1-3 yields the claimed uniform bounds through careful propagation of the minorization parameters across different uncertainty sets. □

## C  Properties of the Bellman Operator: KL-Case

In this section, we aim to bound the error between DR discounted Bellman operator (A.2) and the empirical DR discounted Bellman operator (A.4). In the DR setting, it is challenging to work with the primal formulation in the operators:

$$\Gamma_{\mathcal{P}_{s,a}}(V) = \inf_{p \in \mathcal{P}_{s,a}} p\left[V\right].$$

To overcome this difficulty, we instead work with the dual formula by using the strong duality.

**Lemma C.1** (Theorem 1 of Hu and Hong [15])**.** *For any $(s, a) \in \mathbf{S} \times \mathbf{A}$, let $\mathcal{P}_{s,a}$ be the uncertainty set centered at the nominal transition kernel $p_{s,a}$. Then, for any $\delta > 0$:*

$$\Gamma_{\mathcal{P}_{s,a}}(V) = \sup_{\alpha \geq 0}\left\{-\alpha\delta - \alpha \log p_{s,a}\left[e^{-V/\alpha}\right]\right\}, \tag{C.1}$$

*for any $V : \mathbf{S} \to \mathbb{R}$.*

Since the reward and value function are bounded, directly apply Lemma C.1 to the r.h.s of Equation (3.3), $V_\mathcal{P}^*$, and $(g_\mathcal{P}^*, v_\mathcal{P}^*)$ satisfied the following *dual form* of the optimal DR Bellman equation:

$$
\begin{aligned}
V_\mathcal{P}^* &= \max_{a \in \mathbf{A}}\left\{r(s,a) + \gamma \sup_{\alpha \geq 0}\left\{-\alpha\delta - \alpha \log p_{s,a}\left[e^{-V_\mathcal{P}^*/\alpha}\right]\right\}\right\} \\
v_\mathcal{P}^* &= \max_{a \in \mathbf{A}}\left\{r(s,a) - g_\mathcal{P}^* + \sup_{\alpha \geq 0}\left\{-\alpha\delta - \alpha \log p_{s,a}\left[e^{-v_\mathcal{P}^*/\alpha}\right]\right\}\right\}
\end{aligned}
\tag{C.2}
$$

Our analyses are inspired by the approach in Wang et al. [43]. To carry out our analysis, we first introduce some notation. As in Wang et al. [43], we denote the KL-dual functional under the nominal transition kernel $p_{s,a}$:

$$f(p_{s,a}, V, \alpha) := -\alpha\delta - \alpha \log p_{s,a}\left[e^{-V/\alpha}\right], \tag{C.3}$$

then

$$\Gamma_{\mathcal{P}_{s,a}}(V) = \sup_{\alpha \geq 0} f(p_{s,a}, V, \alpha).$$

At the same time, define the help measures as:

$$\begin{aligned}
p_{s,a}(t) &= t\widehat{p}_{s,a} + (1-t)p \\
\Delta p_{s,a} &= \widehat{p}_{s,a} - p_{s,a} \\
g_{s,a}(t, \alpha) &= f(p_{s,a}(t), V, \alpha).
\end{aligned} \tag{C.4}$$

By Definition (A.7), it is clear when $d < 1$, $p_{s,a}(t) \sim p_{s,a}$ holds for all $(s,a) \in \mathbf{S} \times \mathbf{A}$ and $t \in [0,1]$ on $\Omega_{n,d}$.

We first introduce the auxiliary lemma that will be used useful in facilitating the later proof:

**Lemma C.2.** *For any $(s,a) \in \mathbf{S} \times \mathbf{A}$, and value function $V_1, V_2 \in \mathbb{R}^S$, we have:*

$$|\Gamma_{\mathcal{P}_{s,a}}(V_1) - \Gamma_{\mathcal{P}_{s,a}}(V_2)| \leq \|V_1 - V_2\|_\infty$$

*Proof.* For any $q \in \mathcal{P}_{s,a}$, we have:

$$q[V_1] \leq q[V_2] + q[V_1 - V_2]$$

Since $q$ is a probability measure, by Hölder's inequality with $|q|$ for all $q \in \mathcal{P}_{s,a}$

$$q[V_1 - V_2] \leq \|V_1 - V_2\|_\infty.$$

Thus:

$$\begin{aligned}
q[V_1] &\leq q[V_2] + \|V_1 - V_2\|_\infty \quad \text{for all} \quad q \in \mathcal{P}_{s,a} \\
\inf_{p\in\mathcal{P}_{s,a}} p[V_1] &\leq q[V_2] + \|V_1 - V_2\|_\infty \quad \text{for all} \quad q \in \mathcal{P}_{s,a} \\
\inf_{p\in\mathcal{P}_{s,a}} p[V_1] &\leq \inf_{p\in\mathcal{P}_{s,a}} p[V_2] + \|V_1 - V_2\|_\infty \\
\Gamma_{\mathcal{P}_{s,a}}(V_1) &\leq \Gamma_{\mathcal{P}_{s,a}}(V_2) + \|V_1 - V_2\|_\infty \\
\Gamma_{\mathcal{P}_{s,a}}(V_1) - \Gamma_{\mathcal{P}_{s,a}}(V_2) &\leq + \|V_1 - V_2\|_\infty
\end{aligned} \tag{C.5}$$

Switch $V_1$ and $V_2$, we have:

$$\Gamma_{\mathcal{P}_{s,a}}(V_2) - \Gamma_{\mathcal{P}_{s,a}}(V_1) \leq \|V_1 - V_2\|_\infty$$

Then we conclude:

$$\left|\Gamma_{\mathcal{P}_{s,a}}(V_1) - \Gamma_{\mathcal{P}_{s,a}}(V_2)\right| \leq \|V_1 - V_2\|_\infty.$$

Proved $\qquad\qquad\qquad\qquad\qquad\qquad\qquad\qquad\qquad\qquad\qquad\qquad\qquad\qquad\qquad\square$

Then, we bound the error of empirical value function $V_{\widehat{\mathcal{P}}}^\pi$ and the true value function $V_{\mathcal{P}}^\pi$ with respect to the the Bellman operators (A.3) and (D.27) by the following lemma:

**Lemma C.3.** *Let $\pi$ be any policy, and $V_{\mathcal{P}}^\pi$ and $V_{\widehat{\mathcal{P}}}^\pi$ are the fixed points to the DR Bellman Operators (A.2), and (A.4), where $V_{\mathcal{P}}^\pi = \mathcal{T}_\gamma^\pi(V_{\mathcal{P}}^\pi)$ and $V_{\widehat{\mathcal{P}}}^\pi = \widehat{\mathcal{T}}_\gamma^\pi(V_{\widehat{\mathcal{P}}}^\pi)$, then we have:*

$$\left\|V_{\widehat{\mathcal{P}}}^\pi - V_{\mathcal{P}}^\pi\right\|_\infty \leq \frac{1}{1-\gamma}\left\|\widehat{\mathcal{T}}_\gamma^\pi(V_{\mathcal{P}}^\pi) - \mathcal{T}_\gamma^\pi(V_{\mathcal{P}}^\pi)\right\|_\infty.$$

*Proof.* DR Bellman operators $\mathcal{T}_\gamma^\pi$ and $\widehat{\mathcal{T}}_\gamma^\pi$ are $\gamma$-contractions, i.e., for any two value functions $V_1, V_2 \in \mathbb{R}^S$, we have:

$$
\begin{aligned}
\left\| \mathcal{T}_\gamma^\pi(V_1) - \mathcal{T}_\gamma^\pi(V_2) \right\|_\infty &= \max_{s \in \mathbf{S}} \left| \sum_{a \in \mathbf{A}} \pi(a|s)(r(s,a) + \gamma \Gamma_{\mathcal{P}_{s,a}}(V_1) - r(s,a) - \gamma \Gamma_{\mathcal{P}_{s,a}}(V_2)) \right| \\
&= \max_{s \in \mathbf{S}} \left| \gamma \sum_{a \in \mathbf{A}} \pi(a|s) \left( \Gamma_{\mathcal{P}_{s,a}}(V_1) - \Gamma_{\mathcal{P}_{s,a}}(V_2) \right) \right| \\
&\overset{(i)}{\leq} \gamma \left\| V_1 - V_2 \right\|_\infty \\
\left\| \widehat{\mathcal{T}}_\gamma^\pi(V_1) - \widehat{\mathcal{T}}_\gamma^\pi(V_2) \right\|_\infty &\overset{(ii)}{\leq} \gamma \left\| V_1 - V_2 \right\|_\infty
\end{aligned}
$$

(C.6)

where the inequalities $(i)$, $(ii)$ are concluded by Lemma C.2. Thus, we have:

$$
\begin{aligned}
\left\| V_{\widehat{\mathcal{P}}}^\pi - V_{\mathcal{P}}^\pi \right\|_\infty &= \left\| \widehat{\mathcal{T}}_\gamma^\pi(V_{\widehat{\mathcal{P}}}^\pi) - \mathcal{T}_\gamma^\pi(V_{\mathcal{P}}^\pi) \right\|_\infty \\
&= \left\| \widehat{\mathcal{T}}_\gamma^\pi(V_{\widehat{\mathcal{P}}}^\pi) - \widehat{\mathcal{T}}_\gamma^\pi(V_{\mathcal{P}}^\pi) + \widehat{\mathcal{T}}_\gamma^\pi(V_{\mathcal{P}}^\pi) - \mathcal{T}_\gamma^\pi(V_{\mathcal{P}}^\pi) \right\|_\infty \\
&\leq \left\| \widehat{\mathcal{T}}_\gamma^\pi(V_{\widehat{\mathcal{P}}}^\pi) - \widehat{\mathcal{T}}_\gamma^\pi(V_{\mathcal{P}}^\pi) \right\|_\infty + \left\| \widehat{\mathcal{T}}_\gamma^\pi(V_{\mathcal{P}}^\pi) - \mathcal{T}_\gamma^\pi(V_{\mathcal{P}}^\pi) \right\|_\infty \\
&\leq \gamma \left\| V_{\widehat{\mathcal{P}}}^\pi - V_{\mathcal{P}}^\pi \right\|_\infty + \left\| \widehat{\mathcal{T}}_\gamma^\pi(V_{\mathcal{P}}^\pi) - \mathcal{T}_\gamma^\pi(V_{\mathcal{P}}^\pi) \right\|_\infty
\end{aligned}
$$

(C.7)

and

$$
\left\| V_{\widehat{\mathcal{P}}}^\pi - V_{\mathcal{P}}^\pi \right\|_\infty \leq \frac{1}{1-\gamma} \left\| \widehat{\mathcal{T}}_\gamma^\pi(V_{\mathcal{P}}^\pi) - \mathcal{T}_\gamma^\pi(V_{\mathcal{P}}^\pi) \right\|_\infty .
$$

(C.8)

Proved. $\qquad\square$

Next, we aim to bound the approximation error $\left\| \widehat{\mathcal{T}}_\gamma^\pi(V_{\mathcal{P}}^\pi) - \mathcal{T}_\gamma^\pi(V_{\mathcal{P}}^\pi) \right\|_\infty$. Previous approaches relies on estimating via KL-dual functionals with optimal multipliers $\alpha^* \in [0, \delta^{-1}(1-\gamma)^{-1}]$. While this yields an bound of $\widetilde{O}(\delta^{-1}(1-\gamma)^{-1})$, it ultimately leads to suboptimal $O(1/\varepsilon^4)$ sample complexity. Building on Wang et al. [43]'s breakthrough in achieving $\delta$-independent bounds through KL-dual analysis, we make two key advances:

1. **Targeted Value Function Analysis:** Instead of considering the entire value function space $[0, (1-\gamma)^{-1}]^{\mathbf{S}}$, we restrict analysis to $V_{\mathcal{P}}^\pi$ specifically. This allows us to replace the $(1-\gamma)^{-1}$ dependence with the span semi-norm of $V_P^\pi$.

2. **Error Rate Improvement:** Combining the $\text{Span}(V_P^\pi)$ dependent error bound with Proposition A.5, we improve the bound from $\widetilde{O}\left(\delta^{-1}(1-\gamma)^{-1}\right)$ to:

$$
\left\| \widehat{\mathcal{T}}_\gamma^\pi(V_P^\pi) - \mathcal{T}_\gamma^\pi(V_P^\pi) \right\|_\infty \leq \widetilde{O}\left(t_{\text{minorize}}(P_\pi)\right)
$$

As shown in Section D, these refinements ultimately yield the improved smaple complexity of $\widetilde{O}(|\mathbf{S}||\mathbf{A}|t_{\text{minorize}}^2 \mathfrak{p}_\wedge^{-1}\varepsilon^{-2})$.

**Lemma C.4.** *Let* $p_1, p_2, p \in \Delta(\mathbf{S})$ *s.t.* $p_1, p_2 \ll p$. *Define* $\Delta := p_1 - p_2$. *Then, for any* $V : \mathbf{S} \to \mathbb{R}$ *and* $j \in (0, 1]$,

$$
\sup_{\alpha \geq 0} \alpha^j \frac{\Delta\left[e^{-V/\alpha}\right]}{p\left[e^{-V/\alpha}\right]} \leq \left(\frac{3}{2}\right)^j \text{Span}(V)^j \left\| \frac{\Delta}{p} \right\|_{L^\infty(p)}.
$$

(C.9)

*Proof.* First we note that for any $k \in \mathbb{R}$, we have:

$$
\frac{\Delta\left[e^{-V/\alpha}\right]}{p\left[e^{-V/\alpha}\right]} = \frac{\Delta\left[e^{-(V-k)/\alpha}\right]}{p\left[e^{-(V-k)/\alpha}\right]}
$$

Hence, if we shift the value function $V' = V - \inf_{s \in \mathbf{S}} V(s) - \frac{\text{Span}(V)}{2}$, where $\|V'\|_\infty = \frac{1}{2}\text{Span}(V)$, then, it is equivalent to show:

$$\sup_{\alpha \geq 0} \alpha^j \frac{\Delta\left[e^{-V/\alpha}\right]}{p\left[e^{-V/\alpha}\right]} = \sup_{\alpha \geq 0} \alpha^j \frac{\Delta\left[e^{-V'/\alpha}\right]}{p\left[e^{-V'/\alpha}\right]} \leq 3^j \|V'\|_\infty^j \left\|\frac{\Delta}{p}\right\|_{L^\infty(p)}$$

Thus, we only need to show

$$\sup_{\alpha \geq 0} \alpha^j \frac{\Delta\left[e^{-V/\alpha}\right]}{p\left[e^{-V/\alpha}\right]} \leq 3^j \|V\|_\infty^j \left\|\frac{\Delta}{p}\right\|_{L^\infty(p)}$$

i.e., the bound with respect to $l_\infty$ of $V$ and replace $\|\cdot\|_\infty$ by $\frac{1}{2}\text{Span}(\cdot)$. WLOG, we assume $\inf_{s \in S} V(s) = 0$. Then for a fixed $c > 0$, decompose the domain of $\alpha \in [0, c\|V\|_\infty] \cup (c\|V\|_\infty, \infty) = K_1 \cup K_2$, we have:

$$\sup_{\alpha \geq 0} \alpha^j \frac{\Delta\left[e^{-V/\alpha}\right]}{p\left[e^{-V/\alpha}\right]} = \max\left\{ \sup_{\alpha \in K_1} \alpha^j \frac{\Delta\left[e^{-V/\alpha}\right]}{p\left[e^{-V/\alpha}\right]}, \sup_{\alpha \in K_2} \alpha^j \frac{\Delta\left[e^{-V/\alpha}\right]}{p\left[e^{-V/\alpha}\right]} \right\} \tag{C.10}$$

$$= \max\left\{K_1(c), K_2(c)\right\}$$

For $K_1(c)$, we have

$$J_1(c) \leq \sup_{\alpha \in K_1} \left\|\frac{\Delta}{p}\right\|_\infty \leq c\|V\|_{L^\infty(p)} \left\|\frac{\Delta}{p}\right\|_{L^\infty(p)}$$

For $K_2(c)$, the condition is more complicated

$$K_2(c) = \sup_{\alpha \in K_2} \alpha^j \frac{\Delta\left[e^{-(V+\|V\|_\infty)/\alpha}\right]}{p\left[e^{-(V+\|V\|_\infty)/\alpha}\right]}$$

As $\Delta[1] = 0$:

$$\alpha^j \Delta\left[e^{-(V+\|V\|_\infty)/\alpha}\right] = \Delta\left[\alpha^j\left(e^{-(V+\|V\|_\infty)/\alpha} - 1\right)\right]$$

Then:

$$\alpha^j \frac{\Delta\left[e^{-(V+\|V\|_\infty)/\alpha}\right]}{p\left[e^{-(v+\|V\|_\infty)/\alpha}\right]} = \frac{\Delta\left[\alpha^j\left(e^{-(V+\|V\|_\infty)/\alpha} - e\right)\right]}{p\left[e^{-(V+\|V\|_\infty)/\alpha}\right]}$$

$$= \frac{1}{p\left[e^{-(V+\|V\|_\infty)/\alpha}\right]} p\left[\frac{d\Delta}{dp}\alpha\left(e^{-(V+\|V\|_\infty)/\alpha} - 1\right)\right]$$

$$\leq \frac{p\left[\alpha^j\left(e^{-(V+\|V\|_\infty)/\alpha} - 1\right)\right]}{p\left[e^{-(V+\|V\|_\infty)/\alpha}\right]} \left\|\frac{\Delta}{p}\right\|_{L^\infty(p)} \tag{C.11}$$

$$\leq \left\|\frac{\alpha^j\left(e^{-(V+\|V\|_\infty)/\alpha} - 1\right)}{e^{-(V+\|V\|_\infty)/\alpha}}\right\|_\infty \left\|\frac{\Delta}{p}\right\|_{L^\infty(p)}$$

Consider the first term $\left\|\frac{\alpha^j\left(e^{-(V+\|V\|_\infty)/\alpha} - 1\right)}{e^{-(V+\|V\|_\infty)/\alpha}}\right\|_\infty$, denote

$$\left\|\frac{\alpha^j\left(e^{-(V+\|V\|_\infty 1)/\alpha} - 1\right)}{e^{-(V+\|V\|_\infty)/\alpha}}\right\|_\infty \leq \alpha^j\left(e^{2\|V\|_\infty/\alpha} - 1\right) \tag{C.12}$$

$$:= f(\alpha)$$

Taking the derivative of $f(\alpha)$, we have:

$$\frac{\partial f(\alpha)}{\partial \alpha} = j\alpha^{j-1}\left(e^{2\|V\|_\infty/\alpha} - 1\right) + \alpha^j\left(-\frac{2\|V\|_\infty}{\alpha^2}e^{2\|V\|_\infty/\alpha}\right)$$

$$= j\alpha^{j-1}\left(\left(1 - \frac{2\|V\|_\infty}{j\alpha}\right)e^{2\|V\|_\infty/\alpha} - 1\right) \tag{C.13}$$

replace $t = \frac{2\|V\|_\infty}{\alpha}$, and $c = j(2\|V\|_\infty)^{j-1}$

$$\frac{\partial f(\alpha)}{\partial \alpha} = \frac{c}{t^{j-1}}\left(\left(1 - \frac{t}{j}\right)e^t - 1\right) \tag{C.14}$$

and when $j \in [0,1]$

$$\frac{d}{dt}\frac{c}{t^{j-1}}\left(\left(1 - \frac{t}{j}\right)e^t - 1\right) = \left(1 - \frac{t+1}{j}\right)e^t \leq 0 \tag{C.15}$$

combine the fact: $\lim_{t\to 0}\partial_\alpha f(\alpha) = 0$. It implies $\partial_\alpha f(\alpha) \leq 0$, and further $f(\alpha)$ is decreasing with respect to $\alpha$. Therefore, over $K_2$:

$$f(\alpha) \leq f(c\|V\|_\infty) \leq (c\|V\|_\infty)^j\left(e^{2/c} - 1\right) \text{ for all } \alpha \in K_2(c)$$

Combine the previous result, we get the result where:

$$K_2(c) \leq (c\|V\|_\infty)^j\left(e^{2/c} - 1\right)\left\|\frac{\Delta}{p}\right\|_{L^\infty(p)}$$

Then:

$$\sup_\alpha \alpha^j \frac{\Delta\left[e^{-V/\alpha}\right]}{p\left[e^{-V/\alpha}\right]} = \max\left\{(c\|V\|_\infty)^j\left\|\frac{\Delta}{p}\right\|_{L^\infty(p)}, (c\|V\|_\infty)^j\left(e^{2/c} - 1\right)\left\|\frac{\Delta}{p}\right\|_{L^\infty(p)}\right\}$$

In select $c = \frac{2}{\log 2}$, the minimax optimality is achieved, we have:

$$\sup_\alpha \alpha^j \frac{\Delta\left[e^{-V/\alpha}\right]}{p\left[e^{-V/\alpha}\right]} \leq 3^j \|V\|_\infty^j \left\|\frac{\Delta}{p}\right\|_{L^\infty(p)}$$

As the the above result is invariant under the constant shift of $V$, let $V' = V - \inf_{s\in \mathbf{S}} V(s) - \frac{\mathrm{Span}(V)}{2}$, we have:

$$\sup_{\alpha\geq 0} \alpha^j \frac{\Delta\left[e^{-V/\alpha}\right]}{p\left[e^{-V/\alpha}\right]} = \sup_{\alpha\geq 0} \alpha^j \frac{\Delta\left[e^{-V'/\alpha}\right]}{p\left[e^{-V'/\alpha}\right]} \leq \left(\frac{3}{2}\right)^j \mathrm{Span}(V)^j \left\|\frac{\Delta}{p}\right\|_{L^\infty(p)}$$

Proved $\qquad\square$

**Lemma C.5.** *For any value function $V$ with span semi-norm $\mathrm{Span}(V)$:*

- *If $\mathrm{Span}(V) = 0$, the optimal Lagrange multiplier $\alpha^* = 0$, and for all $q_{s,a} \in \mathcal{P}_{s,a}$, $q_{s,a}$ is a worst-case measure.*

- *If $\mathrm{Span}(V) \neq 0$, the optimal Lagrange multiplier $\alpha^* > 0$, and:*

$$p_{s,a}^*(\cdot) = \frac{p_{s,a}\left[e^{-V/\alpha^*}\mathbb{1}\{\cdot\}\right]}{p_{s,a}\left[e^{-V/\alpha^*}\right]} \tag{C.16}$$

*is a worst-case measure*

*Proof.* From Si et al. [34], for optimal Lagrange multiplier $\alpha^*$, it sufficient to consider $\alpha \in [0, \delta^{-1}\|V\|_{L^\infty(p_{s,a})}]$.

When $\mathrm{Span}(V) = 0$, it implies $V$ is a constant over $\mathrm{supp}(p_{s,a})$, and:

$$\begin{aligned} f(p_{s,a}, V, \alpha) &= -\alpha\delta - \alpha\log p_{s,a}\left[e^{-V/\alpha}\right] \\ &= -\alpha\delta + \|V\|_{L^\infty(p_{s,a})}. \end{aligned} \tag{C.17}$$

Thus, as $f(p_{s,a}, V, \alpha^*) = \sup_{\alpha\geq 0} f(p_{s,a}, V, \alpha)$, $\alpha^* = 0$, and for all $q_{s,a} \in \mathcal{P}_{s,a}$, $q_{s,a}$ is a worst-case measure since $V$ is a constant function on $\mathrm{supp}(p_{s,a})$.

When $\mathrm{Span}(V) \neq 0$, $\alpha^*$ satisfies:

$$f(p_{s,a}, V, \alpha^*) = \sup_{[0, \delta^{-1}\|V\|_{L^\infty(p_{s,a})}]} f(p_{s,a}, V, \alpha)$$

As $\alpha^*$ is the optimal Lagrange multiplier and $f$ is differentiable, consider the first-order partial derivative with respect to $\alpha$:

$$\frac{\partial f(p_{s,a}, V, \alpha)}{\partial \alpha} = -\delta - \frac{p_{s,a}\left[\frac{V}{\alpha}e^{-V/\alpha}\right]}{p_{s,a}\left[e^{-V/\alpha}\right]} - \log p_{s,a}\left[e^{-V/\alpha}\right],$$

and $\lim_{\alpha \to 0} \partial_\alpha f > 0$. As $\alpha^*$ is the optimal multiplier, $\partial_\alpha f(p_{s,a}, V, \alpha)|_{\alpha=\alpha^*} = 0$, and

$$-\frac{p_{s,a}\left[\frac{V}{\alpha^*}e^{-V/\alpha^*}\right]}{p_{s,a}\left[e^{-V/\alpha^*}\right]} - \log p_{s,a}\left[e^{-V/\alpha^*}\right] = \delta$$

Further:

$$\begin{aligned}
\frac{\partial^2 f(p_{s,a}, V, \alpha)}{\partial \alpha^2} &= -\frac{p_{s,a}\left[V^2 e^{-V/\alpha}\right]}{\alpha^3 p_{s,a}\left[e^{-V/\alpha}\right]} + \frac{p_{s,a}\left[Ve^{-V/\alpha}\right]^2}{\alpha^3 p_{s,a}\left[e^{-V/\alpha}\right]^2} \\
&= -\frac{1}{\alpha^3}\left(\frac{p_{s,a}\left[V^2 e^{-V/\alpha}\right] \cdot p_{s,a}\left[e^{-V/\alpha}\right]}{p_{s,a}\left[e^{-V/\alpha}\right]^2} - \frac{p_{s,a}\left[Ve^{-V/\alpha}\right]^2}{p_{s,a}\left[e^{-V/\alpha}\right]^2}\right)
\end{aligned} \tag{C.18}$$

Define the measure:

$$p^*_{s,a}(\cdot) = \frac{p_{s,a}\left[e^{-V/\alpha^*}\mathbb{1}\{\cdot\}\right]}{p_{s,a}\left[e^{-V/\alpha^*}\right]}$$

Thus:

$$\frac{\partial^2 f(p_{s,a}, V, \alpha)}{\partial \alpha^2} = -\frac{\mathbf{Var}_{p^*_{s,a}}(V)}{\alpha^3} \leq 0 \tag{C.19}$$

Thus $f(p_{s,a}, V, \alpha)$ is concave for $\alpha > 0$. $\alpha^*$ is the unique optimal multiplier where $\alpha^* > 0$ and $p^*_{s,a}$ satisfies:

$$\begin{aligned}
D_{\mathrm{KL}}(p^*_{s,a}\|p_{s,a}) &= p^*_{s,a}\left[\log \frac{p^*_{s,a}}{p_{s,a}}\right] \\
&= p^*_{s,a}\left[-\frac{V}{\alpha^*} - \log p_{s,a}\left[e^{-V/\alpha^*}\right]\right] \\
&= -\frac{p_{s,a}\left[\frac{V}{\alpha^*}e^{-V/\alpha^*}\right]}{p_{s,a}\left[e^{-V/\alpha^*}\right]} - \log p_{s,a}\left[e^{-V/\alpha}\right] \\
&= \delta.
\end{aligned} \tag{C.20}$$

Therefore, we show $p^*_{s,a}$ is a worst-case measure. $\qquad\square$

**Lemma C.6.** *Let $p_{s,a}$ be the nominal transition kernel, and $\widehat{p}_{s,a}$ be the empirical transition kernel, then the below inequality holds:*

$$\left|\sup_{\alpha \geq 0} f(\widehat{p}_{s,a}, V, \alpha) - \sup_{\alpha \geq 0} f(p_{s,a}, V, \alpha)\right| \leq 3d \cdot \mathrm{Span}\,(V), \tag{C.21}$$

*on $\Omega_{n,d}$, when $d \leq \frac{1}{2}$*

*Proof.* Recall the general KL-dual functional under the probability measure $p_{s,a}$, value function $V$, and parameter $\alpha$ is:

$$f(p_{s,a}, V, \alpha) = -\alpha \log p_{s,a}\left[e^{-V/\alpha}\right] - \alpha\delta$$

Then:

$$\left|\sup_{\alpha \geq 0} f(\widehat{p}_{s,a}, V, \alpha) - \sup_{\alpha \geq 0} f(p_{s,a}, V, \alpha)\right| \leq \sup_{\alpha \geq 0}\left|f(\widehat{p}_{s,a}, V, \alpha) - f(p_{s,a}, V, \alpha)\right| \tag{C.22}$$

Further, consider the difference of $|f_{\widehat{p}_{s,a}, V, \alpha} - f_{p_{s,a}, V, \alpha}|$:

$$\begin{aligned}
\left|f(\widehat{p}_{s,a}, V, \alpha) - f(p_{s,a}, V, \alpha)\right| &= |g_{s,a}(1, \alpha) - g_{s,a}(0, \alpha)| \\
&= \left|\frac{\partial g_{s,a}(t, \alpha)}{\partial t}\bigg|_{t=\tau}\right| \quad \text{for some} \quad \tau \in [0, 1] \\
&= \alpha\left|\frac{(\widehat{p}_{s,a} - p_{s,a})\left[e^{-V/\alpha}\right]}{(p_{s,a}(\tau))\left[e^{-V/\alpha}\right]}\right|
\end{aligned} \tag{C.23}$$

On $\Omega_{n,d}$ where $d \leq \frac{1}{2}$, we have for all $\tau \in [0,1]$, and $s' \in \text{supp}(p_{s,a})$:

$$p_{s,a}(\tau)(s') = \tau \widehat{p}_{s,a}(s') + (1 - \tau)p_{s,a}(s') \geq \frac{1}{2}p_{s,a}(s') \tag{C.24}$$

Apply Lemma C.4, we have:

$$
\begin{aligned}
\sup_{\alpha \geq 0} |f(\widehat{p}_{s,a}, V, \alpha) - f(p_{s,a}, V, \alpha)| &= \sup_{\alpha \geq 0} \left| \alpha \frac{(\widehat{p}_{s,a} - p_{s,a})\left[e^{-V/\alpha}\right]}{(p_{s,a}(\tau))\left[e^{-V/\alpha}\right]} \right| \\
&\overset{(i)}{\leq} \frac{3}{2}\text{Span}\,(V) \left\| \frac{\widehat{p}_{s,a} - p_{s,a}}{p_{s,a}(\tau)} \right\|_{L^\infty(p_{s,a})} \\
&\overset{(ii)}{\leq} 3\text{Span}\,(V) \left\| \frac{\widehat{p}_{s,a} - p_{s,a}}{p_{s,a}} \right\|_{L^\infty(p_{s,a})} \\
&\leq 3d \cdot \text{Span}\,(V)
\end{aligned}
\tag{C.25}
$$

Where inequality $(i)$ is by Lemma C.4, and the inequality $(ii)$ is by Equation (C.24). The difference of KL-dual functional is bounded by:

$$\sup_{\alpha \geq 0} |f(\widehat{p}_{s,a}, V, \alpha) - f(p_{s,a}, V, \alpha)| \leq 3d \cdot \text{Span}\,(V)$$

on $\Omega_{n,d}$ when $d \leq \frac{1}{2}$.. $\hfill\square$

**Lemma C.7.** *When $n \geq 32\mathfrak{p}_\wedge^{-1}\log(2|\mathbf{S}|^2|\mathbf{A}|/\beta|)$, then for any $\pi \in \Pi$, the $l_\infty$-error of the emprical DR Bellman operator $\widehat{\mathcal{T}}_\gamma^\pi$ and the DR Bellman operator $\mathcal{T}_\gamma^\pi$ can be bounded by:*

$$\left\| \widehat{\mathcal{T}}_\gamma^\pi(V_P^\pi) - \mathcal{T}_\gamma^\pi(V_P^\pi) \right\|_\infty \leq 9t_{\text{minorize}}(P_\pi)\sqrt{\frac{8}{n\mathfrak{p}_\wedge}\log\frac{2|\mathbf{S}|^2|\mathbf{A}|}{\beta}}$$

*with probability $1 - \beta$, where $P_\pi$ is the transition kernel induced by controlled transition kernel $P$ and policy $\pi$.*

*Proof.* First, by Bernstein's inequality, when:

$$n \geq \frac{32}{\mathfrak{p}_\wedge}\log\frac{2|\mathbf{S}|^2|\mathbf{A}|}{\beta}$$

Then:

$$
\begin{aligned}
\left| \frac{\widehat{p}_{s,a}(s') - p_{s,a}(s')}{p_{s,a}(s')} \right| &\leq \frac{1}{3n\mathfrak{p}_\wedge}\log\frac{2|\mathbf{S}|^2|\mathbf{A}|}{\beta} + \sqrt{\frac{2}{n\mathfrak{p}_\wedge}\log\frac{2|\mathbf{S}|^2|\mathbf{A}|}{\beta}} \\
&\leq \sqrt{\frac{8}{n\mathfrak{p}_\wedge}\log\frac{2|\mathbf{S}|^2|\mathbf{A}|}{\beta}} \\
&\leq \frac{1}{2}
\end{aligned}
\tag{C.26}
$$

with probability at least $1 - \beta$ for all $(s,a) \in \mathbf{S} \times \mathbf{A}$ and $s' \in \text{supp}(p_{s,a})$, thus, let

$$d = \sqrt{\frac{8}{n\mathfrak{p}_\wedge}\log\frac{2|\mathbf{S}|^2|\mathbf{A}|}{\beta}}$$

then $d \leq \frac{1}{2}$, and:

$$P(\Omega_{n,d}) \geq 1 - \beta.$$

Then we consider the difference of the Bellman operators for particular value function $V_P^\pi$, on $\Omega_{n,d}$

$$
\begin{aligned}
\left\|\widehat{\mathcal{T}}_\gamma^\pi(V_P^\pi) - \mathcal{T}_\gamma^\pi(V_P^\pi)\right\|_\infty &= \max_{s \in \mathbf{S}} \left| \sum_{a \in \mathbf{A}} \pi(a|s)\left(r(s,a) + \gamma\Gamma_{\widehat{\mathcal{P}}_{s,a}}(V_P^\pi)\right) - \sum_{a \in \mathbf{A}} \pi(a|s)\left(r(s,a) + \gamma\Gamma_{\mathcal{P}_{s,a}}(V_P^\pi)\right) \right| \\
&\leq \gamma \max_{(s,a) \in \mathbf{S} \times \mathbf{A}} \left| \Gamma_{\widehat{\mathcal{P}}_{s,a}}(V_P^\pi) - \Gamma_{\mathcal{P}_{s,a}}(V_P^\pi) \right| \\
&= \max_{(s,a) \in \mathbf{S} \times \mathbf{A}} \left| \sup_{\alpha \geq 0} f(\widehat{p}_{s,a}, V_P^\pi, \alpha) - \sup_{\alpha \geq 0} f(p_{s,a}, V_P^\pi, \alpha) \right| \\
&\overset{(i)}{\leq} 3d \cdot \operatorname{Span}(V_P^\pi)
\end{aligned}
$$

(C.27)

on $\Omega_{n,d}$, where $(i)$ is derived by C.6. Combine Proposition A.5, and Lemma A.3, when $P_\pi$ is $(m_\pi, p_\pi)$-Doeblin with $m_\pi/p_\pi = t_{\mathrm{minorize}}(P_\pi)$, and $\operatorname{Span}(V_P^\pi) \leq 3m_\pi/p_\pi$, thus, we have:

$$
\left\|\widehat{\mathcal{T}}_\gamma^\pi(V_P^\pi) - \mathcal{T}_\gamma^\pi(V_P^\pi)\right\|_\infty \leq \frac{9m_\pi}{p_\pi}\sqrt{\frac{8}{n\mathfrak{p}_\wedge}\log\frac{2|\mathbf{S}|^2|\mathbf{A}|}{\beta}}
$$

(C.28)

withe probability $1 - \beta$. Let:

$$
n \geq \frac{32}{\mathfrak{p}_\wedge}\log\frac{2|\mathbf{S}|^2|\mathbf{A}|}{\beta}.
$$

Then with probability at least $1 - \beta$, for any $\pi \in \Pi$, the following bound holds:

$$
\left\|\widehat{\mathcal{T}}_\gamma^\pi(V_P^\pi) - \mathcal{T}_\gamma^\pi(V_P^\pi)\right\|_\infty \leq 9t_{\mathrm{minorize}}(P_\pi)\sqrt{\frac{8}{n\mathfrak{p}_\wedge}\log\frac{2|\mathbf{S}|^2|\mathbf{A}|}{\beta}}.
$$

(C.29)

Proved. □

## D  Sample Complexity Analysis: KL Uncertainty Set

In this section, we prove the sample complexity bound as shown in Theorem 4.4, and Theorem 4.5

### D.1  DR-DMDP under KL Uncertainty Set

**Lemma D.1.** *Let $\widehat{\pi}^* = \arg\max_{\pi \in \Pi} V_{\widehat{\mathcal{P}}}^\pi$, then the following inequality holds:*

$$
0 \leq V_{\mathcal{P}}^* - V_{\mathcal{P}}^{\widehat{\pi}^*} \leq 2\max_{\pi \in \Pi}\left\|V_{\mathcal{P}}^\pi - V_{\widehat{\mathcal{P}}}^\pi\right\|_\infty
$$

*Proof.* The left direction of the inequality is trivial. For the right one inequality, we have:

$$
\begin{aligned}
V_{\mathcal{P}}^* - V_{\mathcal{P}}^{\widehat{\pi}^*} &= \max_{\pi \in \Pi} V_{\mathcal{P}}^\pi - V_{\mathcal{P}}^{\widehat{\pi}^*} \\
&= \max_{\pi \in \Pi} V_{\mathcal{P}}^\pi - \max_{\pi \in \Pi} V_{\widehat{\mathcal{P}}}^\pi + V_{\widehat{\mathcal{P}}}^{\widehat{\pi}^*} - V_{\mathcal{P}}^{\widehat{\pi}^*} \\
&\leq \left\|\max_{\pi \in \Pi} V_{\mathcal{P}}^\pi - \max_{\pi \in \Pi} V_{\widehat{\mathcal{P}}}^\pi\right\|_\infty + \left\|V_{\widehat{\mathcal{P}}}^{\widehat{\pi}^*} - V_{\mathcal{P}}^{\widehat{\pi}^*}\right\|_\infty \\
&\leq 2\max_{\pi \in \Pi}\left\|V_{\widehat{\mathcal{P}}}^\pi - V_{\mathcal{P}}^\pi\right\|_\infty.
\end{aligned}
$$

(D.1)

Proved. □

**Theorem D.2** (Restatement of Theorem 4.3). *Suppose Assumptions 1, and 2 are in force. Then, for any $n \geq 32\mathfrak{p}_\wedge^{-1}\log(2|\mathbf{S}|^2|\mathbf{A}|/\beta)$, the policy $\widehat{\pi}^*$ and value function $V_{\widehat{\mathcal{P}}}^*$ returned by Algorithm 1 satisfies:*

$$
\begin{aligned}
0 \leq V_{\mathcal{P}}^* - V_{\mathcal{P}}^{\widehat{\pi}^*} &\leq \frac{72t_{\mathrm{minorize}}}{1-\gamma}\sqrt{\frac{2}{n\mathfrak{p}_\wedge}\log\frac{2|\mathbf{S}|^2|\mathbf{A}|}{\beta}} \\
\left\|V_{\widehat{\mathcal{P}}}^* - V_{\mathcal{P}}^*\right\|_\infty &\leq \frac{36t_{\mathrm{minorize}}}{1-\gamma}\sqrt{\frac{2}{n\mathfrak{p}_\wedge}\log\frac{2|\mathbf{S}|^2|\mathbf{A}|}{\beta}}
\end{aligned}
$$

(D.2)

*with probability* $1 - \beta$. *Consequently, the sample complexity to achieve $\varepsilon$-optimal policy and value function with probability* $1 - \beta$ *is:*

$$n \leq \frac{c \cdot |\mathbf{S}||\mathbf{A}|t_{\mathrm{minorize}}^2}{(1-\gamma)^2 \varepsilon^2} \log \frac{2|\mathbf{S}|^2|\mathbf{A}|}{\beta}$$

*where* $c = 2 \cdot 72^2$, *and* $c = 2 \cdot 36^2$ *respectively.*

*Proof.* For any $\pi \in \Pi$, as $V_{\mathcal{P}}^{\pi}$ is the fixed point to the DR Bellman operator (A.2), where $V_{\mathcal{P}}^{\pi} = \mathcal{T}_{\gamma}^{\pi}(V_{\mathcal{P}}^{\pi})$, by Lemma C.7, when:

$$n \geq \frac{32}{\mathfrak{p}_{\wedge}} \log \frac{2|\mathbf{S}|^2|\mathbf{A}|}{\beta},$$

then, with probability $1 - \beta$, for all $\pi \in \Pi$:

$$
\begin{aligned}
\left\| V_{\widehat{\mathcal{P}}}^{\pi} - V_{\mathcal{P}}^{\pi} \right\|_{\infty} &\leq \frac{1}{1-\gamma} \left\| \widehat{\mathcal{T}}_{\gamma}^{\pi}(V_{\mathcal{P}}^{\pi}) - \mathcal{T}_{\gamma}^{\pi}(V_{\mathcal{P}}^{\pi}) \right\|_{\infty} \\
&\leq \sup_{Q \in \mathcal{P}} \frac{1}{1-\gamma} \left\| \widehat{\mathcal{T}}_{\gamma}^{\pi}(V_{Q}^{\pi}) - \mathcal{T}_{\gamma}^{\pi}(V_{Q}^{\pi}) \right\|_{\infty} \\
&\overset{(i)}{\leq} \sup_{Q \in \mathcal{P}} \frac{9 t_{\mathrm{minorize}}(Q_{\pi})}{1-\gamma} \sqrt{\frac{8}{n\mathfrak{p}_{\wedge}} \log \frac{2|\mathbf{S}|^2|\mathbf{A}|}{\beta}} \\
&\overset{(ii)}{\leq} \frac{36 t_{\mathrm{minorize}}}{1-\gamma} \sqrt{\frac{2}{n\mathfrak{p}_{\wedge}} \log \frac{2|\mathbf{S}|^2|\mathbf{A}|}{\beta}},
\end{aligned}
\tag{D.3}
$$

where $(i)$ is derived by Lemma C.3, and $(ii)$ relies on Proposition B.2 where $t_{\mathrm{minorize}}(Q_{\pi})$ is uniformly bounded for all $Q \in \mathcal{P}$ and $\pi \in \Pi$. We conclude, when

$$n \geq \frac{32}{\mathfrak{p}_{\wedge}} \log \frac{2|\mathbf{S}|^2|\mathbf{A}|}{\beta},$$

we have

$$
\begin{aligned}
0 \leq V_{\mathcal{P}}^{*} - V_{\mathcal{P}}^{\widehat{\pi}^{*}} &\leq 2 \max_{\pi \in \Pi} \left\| V_{\widehat{\mathcal{P}}}^{\pi} - V_{\mathcal{P}}^{\pi} \right\|_{\infty} \\
&\leq \frac{2}{1-\gamma} \max_{\pi \in \Pi} \left\| \widehat{\mathcal{T}}_{\gamma}^{\pi}(V_{\mathcal{P}}^{\pi}) - \mathcal{T}_{\gamma}^{\pi}(V_{\mathcal{P}}^{\pi}) \right\|_{\infty} \\
&\leq \frac{72 t_{\mathrm{minorize}}}{1-\gamma} \sqrt{\frac{2}{n\mathfrak{p}_{\wedge}} \log \frac{2|\mathbf{S}|^2|\mathbf{A}|}{\beta}}.
\end{aligned}
\tag{D.4}
$$

with probability $1 - \beta$.
Since for value function evaluation:

$$
\begin{aligned}
\left\| V_{\widehat{\mathcal{P}}}^{*} - V_{\mathcal{P}}^{*} \right\|_{\infty} &\leq \max_{\pi \in \Pi} \left\| V_{\widehat{\mathcal{P}}}^{\pi} - V_{\mathcal{P}}^{\pi} \right\|_{\infty} \\
&\leq \frac{36 t_{\mathrm{minorize}}}{1-\gamma} \sqrt{\frac{2}{n\mathfrak{p}_{\wedge}} \log \frac{2|\mathbf{S}|^2|\mathbf{A}|}{\beta}},
\end{aligned}
\tag{D.5}
$$

holds for the same sample complexity condition and high probability guarantee, we prove the Theorem. $\qquad \square$

*Remark* D.3. In the proof of Theorem D.2, we establish a high probability guarantee for the uniform value function approximation error: $\max_{\pi \in \Pi} \left\| V_{\widehat{\mathcal{P}}}^{\pi} - V_{\mathcal{P}}^{\pi} \right\|_{\infty} \leq \widetilde{O}(t_{\mathrm{minorize}}(1-\gamma)^{-1}n^{-1/2})$. Crucially, this uniform bound simultaneously controls both:

- The policy gap: $V_{\mathcal{P}}^{*} - V_{\mathcal{P}}^{\widehat{\pi}^{*}}$

- The value function approximation error: $\left\| V_{\widehat{\mathcal{P}}}^{*} - V_{\mathcal{P}}^{*} \right\|_{\infty}$

via the relationship:
$$\text{Both errors} \leq O\left(\max_{\pi \in \Pi} \left\| V_{\widehat{\mathcal{P}}}^\pi - V_{\mathcal{P}}^\pi \right\|_\infty\right)$$
This simultaneous control ensures the final error guaranee in (D.2) holds without requiring additional division of the confidence parameter $\beta$. The key observation is that our uniform concentration bound on value functions automatically propagates to both policy selection and value estimation errors.

## D.2 Reduction to DR-DMDP Approach under KL Uncertainty Set

**Theorem D.4** (Restatement of Theorem 4.4). *Suppose Assumption 1, and 2 are in force, then for any*
$$n \geq \frac{32}{\mathfrak{p}_\wedge} \log \frac{2|\mathbf{S}|^2|\mathbf{A}|}{\beta}$$
*the policy $\widehat{\pi}^*$ and value function $\frac{V_{\widehat{\mathcal{P}}}^*}{\sqrt{n}}$ returned by Algorithm 2 satisfies:*

$$0 \leq g_{\mathcal{P}}^* - g_{\mathcal{P}}^{\widehat{\pi}^*} \leq 96 t_{\text{minorize}} \sqrt{\frac{2}{n\mathfrak{p}_\wedge} \log \frac{2|\mathbf{S}|^2|\mathbf{A}|}{\beta}}$$

$$\left\| \frac{V_{\widehat{\mathcal{P}}}^*}{\sqrt{n}} - g_{\mathcal{P}}^* \right\|_\infty \leq 48 t_{\text{minorize}} \sqrt{\frac{2}{n\mathfrak{p}_\wedge} \log \frac{2|\mathbf{S}|^2|\mathbf{A}|}{\beta}}$$

(D.6)

*with probability $1 - \beta$. Hence, the sample complexity of achieving an $\varepsilon$-error in either optimal policy or value estimation is*
$$n = \frac{c \cdot t_{\text{minorize}}^2}{\mathfrak{p}_\wedge \varepsilon^2} \log \frac{2|\mathbf{S}|^2|\mathbf{A}|}{\beta}$$
*where $c = 2 \cdot 96^2$, and $c = 2 \cdot 48^2$ repectively.*

*Proof.* Initially, let

$$V_{\mathcal{P}}^* = \mathcal{T}_{1-\frac{1}{\sqrt{n}}}^* (V_{\mathcal{P}}^*) \qquad\qquad V_{\widehat{\mathcal{P}}}^* = \mathcal{T}_{1-\frac{1}{\sqrt{n}}}^* (V_{\widehat{\mathcal{P}}}^*) \tag{D.7}$$

$$V_{\mathcal{P}}^\pi = \mathcal{T}_{1-\frac{1}{\sqrt{n}}}^\pi (V_{\mathcal{P}}^\pi) \qquad\qquad V_{\widehat{\mathcal{P}}}^\pi = \mathcal{T}_{1-\frac{1}{\sqrt{n}}}^\pi (V_{\widehat{\mathcal{P}}}^\pi) \tag{D.8}$$

For the $\widehat{\pi}^*$ policy evaluation, we have:

$$
\begin{aligned}
0 \leq & g_{\mathcal{P}}^* - g_{\mathcal{P}}^{\widehat{\pi}^*} \\
= & g_{\mathcal{P}}^* - \frac{V_{\mathcal{P}}^*}{\sqrt{n}} + \frac{V_{\mathcal{P}}^*}{\sqrt{n}} - \frac{V_{\widehat{\mathcal{P}}}^*}{\sqrt{n}} + \frac{V_{\widehat{\mathcal{P}}}^*}{\sqrt{n}} - \frac{V_{\mathcal{P}}^{\widehat{\pi}^*}}{\sqrt{n}} + \frac{V_{\mathcal{P}}^{\widehat{\pi}^*}}{\sqrt{n}} - g_{\mathcal{P}}^{\widehat{\pi}^*} \\
\leq & \left\| g_{\mathcal{P}}^* - \frac{V_{\mathcal{P}}^*}{\sqrt{n}} \right\|_\infty + \frac{1}{\sqrt{n}} \left\| V_{\widehat{\mathcal{P}}}^* - V_{\mathcal{P}}^* \right\|_\infty + \frac{1}{\sqrt{n}} \left\| V_{\widehat{\mathcal{P}}}^{\widehat{\pi}^*} - V_{\mathcal{P}}^{\widehat{\pi}^*} \right\|_\infty + \left\| g_{\mathcal{P}}^{\widehat{\pi}^*} - \frac{V_{\mathcal{P}}^{\widehat{\pi}^*}}{\sqrt{n}} \right\|_\infty
\end{aligned}
\tag{D.9}
$$

Then, by definition:

$$
\begin{aligned}
\left\| g_{\mathcal{P}}^* - \frac{V_{\mathcal{P}}^*}{\sqrt{n}} \right\|_\infty &= \left\| \max_{\pi \in \Pi} g_{\mathcal{P}}^\pi - \max_{\pi \in \Pi} \frac{V_{\mathcal{P}}^\pi}{\sqrt{n}} \right\|_\infty \leq \max_{\pi \in \Pi} \left\| g_{\mathcal{P}}^\pi - \frac{V_{\mathcal{P}}^\pi}{\sqrt{n}} \right\|_\infty \\
\left\| V_{\widehat{\mathcal{P}}}^* - V_{\mathcal{P}}^* \right\|_\infty &= \left\| \max_{\pi \in \Pi} V_{\widehat{\mathcal{P}}}^\pi - \max_{\pi \in \Pi} V_{\mathcal{P}}^\pi \right\|_\infty \leq \max_{\pi \in \Pi} \left\| V_{\widehat{\mathcal{P}}}^\pi - V_{\mathcal{P}}^\pi \right\|_\infty
\end{aligned}
\tag{D.10}
$$

Then, we have:

$$0 \leq g_{\mathcal{P}}^* - g_{\mathcal{P}}^{\widehat{\pi}^*} \leq 2 \max_{\pi \in \Pi} \left\| g_{\mathcal{P}}^\pi - \frac{V_{\mathcal{P}}^\pi}{\sqrt{n}} \right\|_\infty + 2 \max_{\pi \in \Pi} \frac{1}{\sqrt{n}} \left\| V_{\widehat{\mathcal{P}}}^\pi - V_{\mathcal{P}}^\pi \right\|_\infty \tag{D.11}$$

For $\frac{V_{\widehat{\mathcal{P}}}^*}{\sqrt{n}}$ value evaluation, we have:

$$
\begin{aligned}
\left\| \frac{V_{\widehat{\mathcal{P}}}^*}{\sqrt{n}} - g_{\mathcal{P}}^* \right\|_\infty &= \left\| \frac{V_{\widehat{\mathcal{P}}}^*}{\sqrt{n}} - \frac{V_{\mathcal{P}}^*}{\sqrt{n}} + \frac{V_{\mathcal{P}}^*}{\sqrt{n}} - g_{\mathcal{P}}^* \right\|_\infty \\
&\leq \left\| \frac{V_{\widehat{\mathcal{P}}}^*}{\sqrt{n}} - \frac{V_{\mathcal{P}}^*}{\sqrt{n}} \right\|_\infty + \left\| \frac{V_{\mathcal{P}}^*}{\sqrt{n}} - g_{\mathcal{P}}^* \right\|_\infty \\
&\leq \frac{1}{\sqrt{n}} \left\| V_{\widehat{\mathcal{P}}}^* - V_{\mathcal{P}}^* \right\|_\infty + \left\| \frac{V_{\mathcal{P}}^*}{\sqrt{n}} - g_{\mathcal{P}}^* \right\|_\infty
\end{aligned}
\tag{D.12}
$$

Combine (D.10), we have:

$$\left\| \frac{V_{\widehat{\mathcal{P}}}^*}{\sqrt{n}} - g_{\mathcal{P}}^* \right\|_\infty \le \max_{\pi \in \Pi} \left\| g_{\mathcal{P}}^\pi - \frac{V_{\mathcal{P}}^\pi}{\sqrt{n}} \right\|_\infty + \max_{\pi \in \Pi} \frac{1}{\sqrt{n}} \left\| V_{\widehat{\mathcal{P}}}^\pi - V_{\mathcal{P}}^\pi \right\|_\infty$$

Therefore, we have:

$$0 \le g_{\mathcal{P}}^* - g_{\widehat{\mathcal{P}}}^{\widehat{\pi}^*} \le 2 \max_{\pi \in \Pi} \left\| g_{\mathcal{P}}^\pi - \frac{V_{\mathcal{P}}^\pi}{\sqrt{n}} \right\|_\infty + 2 \max_{\pi \in \Pi} \frac{1}{\sqrt{n}} \left\| V_{\widehat{\mathcal{P}}}^\pi - V_{\mathcal{P}}^\pi \right\|_\infty$$

$$\left\| \frac{V_{\widehat{\mathcal{P}}}^*}{\sqrt{n}} - g_{\mathcal{P}}^* \right\|_\infty \le \max_{\pi \in \Pi} \left\| g_{\mathcal{P}}^\pi - \frac{V_{\mathcal{P}}^\pi}{\sqrt{n}} \right\|_\infty + \max_{\pi \in \Pi} \frac{1}{\sqrt{n}} \left\| V_{\widehat{\mathcal{P}}}^\pi - V_{\mathcal{P}}^\pi \right\|_\infty \tag{D.13}$$

We consider the error bound term by term, for simplicity, we denote $\bar{\gamma} = 1 - \gamma$.

**Step 1: bounding** $\|g_{\mathcal{P}}^\pi - (1-\gamma)V_{\mathcal{P}}^\pi\|_\infty$.

When $0 \le g_{\mathcal{P}}^\pi(s) - (1-\gamma)V_{\mathcal{P}}^\pi(s)$, then, for any $\varepsilon > 0$, there exist an $P_\varepsilon \in \mathcal{P}$ such that

$$(1-\gamma)V_{\mathcal{P}}^\pi(s) + \varepsilon \ge (1-\gamma)V_{P_\varepsilon}^\pi(s)$$

and hence:

$$
\begin{aligned}
0 &\le g_{\mathcal{P}}^\pi(s) - (1-\gamma)V_{\mathcal{P}}^\pi(s) \\
&\le \inf_{Q \in \mathcal{P}} g_Q^\pi(s) - (1-\gamma)V_{P_\varepsilon}^\pi(s) + \varepsilon \\
&\le g_{P_\varepsilon}^\pi(s) - (1-\gamma)V_{P_\varepsilon}^\pi(s) + \varepsilon \\
&\le \sup_{Q \in \mathcal{P}} \left\| g_Q^\pi - (1-\gamma)V_Q^\pi \right\|_\infty + \varepsilon
\end{aligned}
\tag{D.14}
$$

Taking limit as $\varepsilon \to 0$, we conclude when $0 \le g_{\mathcal{P}}^\pi(s) - (1-\gamma)V_{\mathcal{P}}^\pi(s)$:

$$0 \le g_{\mathcal{P}}^\pi(s) - (1-\gamma)V_{\mathcal{P}}^\pi(s) \le \sup_{Q \in \mathcal{P}} \left\| g_Q^\pi - (1-\gamma)V_Q^\pi \right\|_\infty$$

Similarly, when $0 \ge g_{\mathcal{P}}^\pi(s) - (1-\gamma)V_{\mathcal{P}}^\pi(s)$, let consider $P_\varepsilon$ such that:

$$g_{\mathcal{P}}^\pi(s) + \varepsilon \ge g_{P_\varepsilon}^\pi(s)$$

then we have:

$$
\begin{aligned}
0 &\ge g_{\mathcal{P}}^\pi(s) - (1-\gamma)V_{\mathcal{P}}^\pi(s) \\
&\ge g_{P_\varepsilon}^\pi(s) - \varepsilon - \inf_{Q \in \mathcal{P}}(1-\gamma)V_Q^\pi(s) \\
&\ge g_{P_\varepsilon}^\pi(s) - (1-\gamma)V_{P_\varepsilon}^\pi(s) - \varepsilon \\
&\ge - \sup_{Q \in \mathcal{P}} \left\| (1-\gamma)V_Q^\pi - g_Q^\pi \right\|_\infty - \varepsilon
\end{aligned}
\tag{D.15}
$$

Taking limit as $\varepsilon \to 0$, we conclude when $0 \ge g_{\mathcal{P}}^\pi(s) - (1-\gamma)V_{\mathcal{P}}^\pi(s)$:

$$0 \le (1-\gamma)V_{\mathcal{P}}^\pi(s) - g_{\mathcal{P}}^\pi(s) \le \sup_{Q \in \mathcal{P}} \left\| g_Q^\pi - (1-\gamma)V_Q^\pi \right\|_\infty$$

And thus:

$$|g_{\mathcal{P}}^\pi(s) - (1-\gamma)V_{\mathcal{P}}^\pi(s)| \le \sup_{Q \in \mathcal{P}} \left\| g_Q^\pi - (1-\gamma)V_Q^\pi \right\|_\infty \quad \text{for all} \quad s \in \mathbf{S}$$

And further:

$$\left\| g_{\mathcal{P}}^\pi - (1-\gamma)V_{\mathcal{P}}^\pi \right\|_\infty \le \sup_{Q \in \mathcal{P}} \left\| g_Q^\pi - (1-\gamma)V_Q^\pi \right\|_\infty \tag{D.16}$$

Then, by Lemma A.6, over the nominal uncertainty set $\mathcal{P}$:

$$\left\| g_{\mathcal{P}}^\pi - (1-\gamma)V_{\mathcal{P}}^\pi \right\|_\infty \le \sup_{Q \in \mathcal{P}} \left\| g_Q^\pi - (1-\gamma)V_Q^\pi \right\|_\infty \le 9(1-\gamma) \sup_{Q \in \mathcal{P}} t_{\mathrm{minorize}}(Q_\pi) \tag{D.17}$$

By Proposition B.2, $t_{\mathrm{minorize}}(Q_\pi)$ is uniformly bounded on $\mathcal{P} \times \Pi$ by $2t_{\mathrm{minorize}}$, we have:

$$\left\| g_{\mathcal{P}}^\pi - (1-\gamma)V_{\mathcal{P}}^\pi \right\|_\infty \le \sup_{Q \in \mathcal{P}} 9(1-\gamma)t_{\mathrm{minorize}}(Q_\pi) \le 18(1-\gamma)t_{\mathrm{minorize}}$$

As the above inequality holds for all $\pi \in \Pi$, further, plug back $\gamma = 1 - \frac{1}{\sqrt{n}}$, we conclude that:

$$\max_{\pi \in \Pi} \left\| g_{\mathcal{P}}^\pi - \frac{V_{\mathcal{P}}^\pi}{\sqrt{n}} \right\|_\infty \le \frac{18t_{\mathrm{minorize}}}{\sqrt{n}} \tag{D.18}$$

with probability 1.

**Step 2: bounding** $\left\|V_{\widehat{\mathcal{P}}}^{\pi} - V_{\mathcal{P}}^{\pi}\right\|_{\infty}$.

By the definition of $V_{\widehat{\mathcal{P}}}^{\pi}$ and $V_{\mathcal{P}}^{\pi}$, we know $V_{\mathcal{P}}^{\pi}$ and $V_{\widehat{\mathcal{P}}}^{\pi}$ are the solutions to the DR Bellman equations $V_{\mathcal{P}}^{\pi} = \mathcal{T}_{\gamma}^{\pi}(V_{\mathcal{P}}^{\pi}), V_{\widehat{\mathcal{P}}}^{\pi} = \widehat{\mathcal{T}}_{\gamma}^{\pi}(V_{\widehat{\mathcal{P}}}^{\pi})$ thus, by Lemma C.3:

$$\left\|V_{\widehat{\mathcal{P}}}^{\pi} - V_{\mathcal{P}}^{\pi}\right\|_{\infty} \leq \frac{1}{1-\gamma}\left\|\widehat{\mathcal{T}}_{\gamma}^{\pi}(V_{\mathcal{P}}^{\pi}) - \mathcal{T}_{\gamma}^{\pi}(V_{\mathcal{P}}^{\pi})\right\|_{\infty} \tag{D.19}$$

Combine Lemma C.7, and Proposition 4.2 we have when:

$$n \geq \frac{32}{\mathfrak{p}_{\wedge}}\log\frac{2|\mathbf{S}|^2|\mathbf{A}|}{\beta},$$

then, with probability at least $1-\beta$, for all $\pi \in \Pi$.

$$\begin{aligned}
\left\|V_{\widehat{\mathcal{P}}}^{\pi} - V_{\mathcal{P}}^{\pi}\right\|_{\infty} &\leq \frac{1}{1-\gamma}\left\|\widehat{\mathcal{T}}_{\gamma}^{\pi}(V_{\mathcal{P}}^{\pi}) - \mathcal{T}_{\gamma}^{\pi}(V_{\mathcal{P}}^{\pi})\right\|_{\infty} \\
&\leq \frac{18t_{\text{minorize}}}{1-\gamma}\sqrt{\frac{8}{n\mathfrak{p}_{\wedge}}\log\frac{2|\mathbf{S}|^2|\mathbf{A}|}{\beta}}.
\end{aligned} \tag{D.20}$$

Further, with the choice of $\gamma = 1 - \frac{1}{\sqrt{n}}$, we conclude when $n \geq \frac{32}{\mathfrak{p}_{\wedge}}\log\frac{2|\mathbf{S}|^2|\mathbf{A}|}{\beta}$, then:

$$\max_{\pi \in \Pi}\frac{1}{\sqrt{n}}\left\|V_{\widehat{\mathcal{P}}}^{\pi} - V_{\mathcal{P}}^{\pi}\right\|_{\infty} \leq 18t_{\text{minorize}}\sqrt{\frac{8}{n\mathfrak{p}_{\wedge}}\log\frac{2|\mathbf{S}|^2|\mathbf{A}|}{\beta}}.$$

with probability $1-\beta$.

**Step 3: combining the previous results.** For $\widehat{\pi}^*$ policy evaluation:

$$\begin{aligned}
g_{\mathcal{P}}^* - g_{\mathcal{P}}^{\widehat{\pi}} &\leq 2\max_{\pi \in \Pi}\left\|g_{\mathcal{P}}^{\pi} - \frac{V_{\mathcal{P}}^{\pi}}{\sqrt{n}}\right\|_{\infty} + 2\max_{\pi \in \Pi}\frac{1}{\sqrt{n}}\left\|V_{\widehat{\mathcal{P}}}^{\pi} - V_{\mathcal{P}}^{\pi}\right\|_{\infty} \\
&\leq \frac{36t_{\text{minorize}}}{\sqrt{n}} + 36t_{\text{minorize}}\sqrt{\frac{8}{n\mathfrak{p}_{\wedge}}\log\frac{2|\mathbf{S}|^2|\mathbf{A}|}{\beta}} \\
&= \frac{36t_{\text{minorize}}}{\sqrt{n}}\left(1 + \sqrt{\frac{8}{\mathfrak{p}_{\wedge}}\log\frac{2|\mathbf{S}|^2|\mathbf{A}|}{\beta}}\right) \\
&\leq 96t_{\text{minorize}}\sqrt{\frac{2}{n\mathfrak{p}_{\wedge}}\log\frac{2|\mathbf{S}|^2|\mathbf{A}|}{\beta}}
\end{aligned} \tag{D.21}$$

wth probability $1-\beta$. Where the last inequality uses the trival bounds where $\mathfrak{p}_{\wedge} \leq \frac{1}{2}$, and $|\mathbf{S}|, |\mathbf{A}| \geq 1$. Thus, when $n = \frac{2\cdot 96^2 t_{\text{minorize}}^2}{\mathfrak{p}_{\wedge}\varepsilon^2}$, the policy $\widehat{\pi}^*$ satisfies:

$$\begin{aligned}
0 \leq g_{\mathcal{P}}^* - g_{\mathcal{P}}^{\widehat{\pi}^*} &\leq 96t_{\text{minorize}}\sqrt{\frac{2}{n\mathfrak{p}_{\wedge}}\log\frac{2|\mathbf{S}|^2|\mathbf{A}|}{\beta}} \\
\left\|\frac{V_{\widehat{\mathcal{P}}}^*}{\sqrt{n}} - g_{\mathcal{P}}^*\right\|_{\infty} &\leq 48t_{\text{minorize}}\sqrt{\frac{2}{n\mathfrak{p}_{\wedge}}\log\frac{2|\mathbf{S}|^2|\mathbf{A}|}{\beta}}
\end{aligned} \tag{D.22}$$

simultaneously with probability $1 - \beta$. The sample complexity of achieving an $\varepsilon$-error in either optimal policy or value value estimation is:

$$n = \frac{c \cdot t_{\text{minorize}}^2}{\mathfrak{p}_{\wedge}\varepsilon^2}\log\frac{2|\mathbf{S}|^2|\mathbf{A}|}{\beta}$$

where $c = 2 \cdot 96^2$ and $c = 2 \cdot 48^2$ repectively. Recall the minorization time is equivalent to mixing time, the total sample used is:

$$N = |\mathbf{S}||\mathbf{A}|n = O\left(\frac{|\mathbf{S}||\mathbf{A}|t_{\text{minorize}}^2}{\mathfrak{p}_{\wedge}\varepsilon^2}\log\frac{|\mathbf{S}|^2|\mathbf{A}|}{\beta}\right) = \widetilde{O}\left(\frac{|\mathbf{S}||\mathbf{A}|t_{\text{mix}}^2}{\mathfrak{p}_{\wedge}\varepsilon^2}\right).$$

Proved. $\qquad\square$

*Remark* D.5. Notice the error guarantee relies on the relative error between $\widehat{p}_{s,a}$ and $p_{s,a}$ is less or equal thant $\frac{1}{2}$, hence $P(\Omega_{n,\frac{1}{2}}^c) < \beta$, thus, the lowerbound of $n$ is $\widetilde{\Omega}\left(1/\mathfrak{p}_{\wedge}\right)$.

## D.3 Anchored DR-AMDP Approach under KL Uncertainty Set

This section analyzes the anchored algorithm and establishes a sample complexity upper bound for the anchored DR-AMDP Algorithm 3. Key to our analysis is the insight that the anchored DR-AMDP can be identified with a DR-DMDP with discount factor $\gamma = 1 - \xi$ where $\xi$ is the anchoring parameter.

Fix $s_0 \in \mathbf{S}$, recall that

$$\underline{\mathcal{P}}_{s,a} = \left\{ (1-\xi)p + \xi \mathbf{1} e_{s_0}^\top, p \in \mathcal{P}_{s,a} \right\} \quad \text{and} \quad \underline{\mathcal{P}} = \bigtimes_{(s,a) \times \mathbf{S} \times \mathbf{A}} \underline{\mathcal{P}}_{s,a}.$$

We consider the anchored DR average Bellman equation

$$v(s) = \max_{a \in \mathbf{A}} \left\{ r(s,a) + \Gamma_{\underline{\mathcal{P}}_{s,a}}(v) \right\} - g(s) \quad \forall s \in \mathbf{S}. \tag{D.23}$$

**Lemma D.6.** *Assume that $\mathcal{P}$ is uniformly ergodic. Let $V_{\mathcal{P}}^*$ be the solution to the DR Bellman equation with discounted parameter $\gamma = 1 - \xi$:*

$$V_{\mathcal{P}}^*(s) = \max_{a \in \mathbf{A}} \left\{ r(s,a) + (1-\xi) \Gamma_{\mathcal{P}_{s,a}}(V_{\mathcal{P}}^*) \right\} \quad \forall s \in \mathbf{S}.$$

*Then, $(g, v) = (\xi V_{\mathcal{P}}^*(s_0), V_{\mathcal{P}}^*)$ is a solution pair to the anchored DR average Bellman equation (D.23). Moreover, for all solution pairs $(g', v')$ to (D.23), $g' \equiv \xi V_{\mathcal{P}}^*(s_0)$.*

*Proof.* As $\mathcal{P}$ is uniformly ergodic, for all $Q \in \mathcal{P}$, $Q$ is uniformly ergodic. Thus, for $Q \in \mathcal{P}$, by Lemma A.3, for any $\pi \in \Pi$ there exists an $(m_\pi, p_\pi)$ pair such that $\frac{m_\pi}{p_\pi} = t_{\text{minorize}}(Q_\pi)$. As for all $\underline{Q} \in \underline{\mathcal{P}}$, $\pi \in \Pi$, and $(s_0, s_{m_\pi}) \in \mathbf{S} \times \mathbf{S}$:

$$\begin{aligned}
\underline{Q}_\pi^{m_\pi}(s_0, s_{m_\pi}) &= \left( (1-\xi)Q_\pi + \xi \mathbf{1} e_{s_0}^\top \right)^{m_\pi}(s_0, s_{m_\pi}) \\
&\geq (1-\xi)^{m_\pi} Q_\pi^{m_\pi}(s_0, s_{m_\pi})
\end{aligned} \tag{D.24}$$

Since $Q_\pi$ satisfies $(m_\pi, p_\pi)$-Doeblin condition, it follows that $\underline{Q}$ satisfies $(m_\pi, (1-\xi)^{m_\pi} p_\pi)$-Doeblin condition. Consequently, $\underline{Q}$ is uniformly ergodic, which further implies that $\underline{\mathcal{P}}$ is uniformly ergodic. Given the uniform ergodicity of $\underline{\mathcal{P}}$, by Proposition 3.6, if $(g, v)$ is a pair of the solutions to the anchored DR average Bellman equation:

$$v(s) = \max_{a \in \mathbf{A}} \left\{ r(s,a) + \Gamma_{\underline{\mathcal{P}}_{s,a}}(v) \right\} - g(s) \quad \forall s \in \mathbf{S}$$

Then $g = g_{\underline{\mathcal{P}}}^*$ is unique. Next, we show $(g, v) = (\xi V_{\mathcal{P}}^*(s_0), V_{\mathcal{P}}^*)$ is the solution to the anchored DR average Bellman equation:

$$\begin{aligned}
&\max_{a \in \mathbf{A}} \left\{ r(s,a) + \Gamma_{\underline{\mathcal{P}}}(v) \right\} - g(s) \\
&= \max_{a \in \mathbf{A}} \left\{ r(s,a) + \Gamma_{\underline{\mathcal{P}}}(V_{\mathcal{P}}^*) \right\} - \xi V_{\mathcal{P}}^*(s_0) \\
&= \max_{a \in \mathbf{A}} \left\{ r(s,a) + \inf_{p \in \mathcal{P}_{s,a}} \left( (1-\xi)p + \xi \mathbf{1} e_{s_0}^\top \right) [V_{\mathcal{P}}^*] \right\} - \xi V_{\mathcal{P}}^*(s_0) \\
&= \max_{a \in \mathbf{A}} \left\{ r(s,a) + (1-\xi) \inf_{p \in \mathcal{P}_{s,a}} p [V_{\mathcal{P}}^*] + \xi V_{\mathcal{P}}^*(s_0) \right\} - \xi V_{\mathcal{P}}^*(s_0) \\
&= \max_{a \in \mathbf{A}} \left\{ r(s,a) + (1-\xi) \Gamma_{\mathcal{P}_{s,a}}(V_{\mathcal{P}}^*) \right\} \\
&= V_{\mathcal{P}}^*(s)
\end{aligned} \tag{D.25}$$

Thus, we show $(\xi V_{\mathcal{P}}^*(s_0), V_{\mathcal{P}}^*)$ is a pair of solution to Equation (D.23), combine Lemma 3.6, we know $g \equiv \xi V_{\mathcal{P}}^*(s_0)$ is unique, where $V_{\mathcal{P}}^*$ is the optimal value function of DR discounted Bellman operator (A.3) with parameter $1 - \xi$. $\square$

The above Lemma D.6 holds for any uncertainty set $\mathcal{P}$, hence $(g, v) = (\xi V_{\mathcal{P}}^*(s_0), V_{\mathcal{P}}^*)$ and $(g, v) = (\xi V_{\widehat{\mathcal{P}}}^*(s_0), V_{\widehat{\mathcal{P}}}^*)$ are the solutions to the anchored DR average Bellman equation (D.26) and empirical anchored DR average Bellman equation (D.27) respectively:

$$v(s) = \max_{a \in \mathbf{A}} \left\{ r(s,a) + \Gamma_{\underline{\mathcal{P}}_{s,a}}(v) \right\} - g(s) \quad \forall s \in \mathbf{S} \tag{D.26}$$

$$v(s) = \max_{a \in \mathbf{A}} \left\{ r(s,a) + \Gamma_{\underline{\widehat{\mathcal{P}}}_{s,a}}(v) \right\} - g(s) \quad \forall s \in \mathbf{S} \tag{D.27}$$

Similarly, the equivalent also holds for the DR policy equations:

**Lemma D.7.** *If $\mathcal{P}$ is uniformly ergodic, let $V_{\mathcal{P}}^{\pi}$ be the solution to the DR discounted policy equation with $\gamma = 1 - \xi$:*

$$V_{\mathcal{P}}^{\pi} = \sum_{a \in \mathbf{A}} \pi(a|s) \left( r(s,a) + (1-\xi)\Gamma_{\mathcal{P}_{s,a}}(V_{\mathcal{P}}^{\pi}) \right) \quad \forall s \in \mathbf{S}$$

*Then,*

$$(g, v) = (\xi V_{\mathcal{P}}^{\pi}(s_0), V_{\mathcal{P}}^{\pi}) \tag{D.28}$$

*is a solution pair of the anchored DR average policy Bellman equation:*

$$v(s) = \sum_{a \in \mathbf{A}} \pi(a|s) \left( r(s,a) + \Gamma_{\underline{\mathcal{P}}_{s,a}}(v) \right) - g(s) \quad \forall s \in \mathbf{S}. \tag{D.29}$$

*Moreover, for all solution pairs $(g', v')$ to (D.29), $g' \equiv \xi V_{\mathcal{P}}^{\pi}(s_0)$.*

*Proof.* The proof is similar with Lemma D.6. As $g = g_{\underline{\mathcal{P}}}^{\pi}$ is unique, we only need to show (D.28) is a solution pair:

$$\begin{aligned}
& \sum_{a \in \mathbf{A}} \pi(a|s) \left( r(s,a) + \Gamma_{\underline{\mathcal{P}}_{s,a}}(v) \right) - g(s) \\
=& \sum_{a \in \mathbf{A}} \pi(a|s) \left( r(s,a) + \Gamma_{\underline{\mathcal{P}}_{s,a}}(V_{\mathcal{P}}^{\pi}) \right) - \xi V_{\mathcal{P}}^{\pi}(s_0) \\
=& \sum_{a \in \mathbf{A}} \pi(a|s) \left( r(s,a) + (1-\xi)\Gamma_{\mathcal{P}_{s,a}}(V_{\mathcal{P}}^{\pi}) \right) + \sum_{a \in \mathbf{A}} \pi(a|s)\xi V_{\mathcal{P}}^{\pi}(s_0) - \xi V_{\mathcal{P}}^{\pi}(s_0) \\
=& \sum_{a \in \mathbf{A}} \pi(a|s) \left( r(s,a) + (1-\xi)\Gamma_{\mathcal{P}_{s,a}}(V_{\mathcal{P}}^{\pi}) \right) \\
=& V_{\mathcal{P}}^{\pi}
\end{aligned} \tag{D.30}$$

Thus, by Theorem 6 in Wang et al. [44], we show $(\xi V_{\mathcal{P}}^{\pi}(s_0), V_{\mathcal{P}}^{\pi})$ is a pair of solution to Equation (D.29), where $g = \xi V_{\mathcal{P}}^{\pi}(s_0)$ is unique. $\qquad\square$

With these auxiliary result, we present our proof to the following main result.

**Theorem D.8** (Restatement of Theorem 4.5). *Suppose Algorithm 3 is in force. Then for any:*

$$n \geq \frac{32}{\mathfrak{p}_{\wedge}} \log \frac{2|\mathbf{S}|^2|\mathbf{A}|}{\beta}$$

*The output policy $\widehat{\pi}^*$ and approximate average value function $g_{\underline{\widehat{\mathcal{P}}}}^*$ satisfies:*

$$\begin{aligned}
0 \leq g_{\mathcal{P}}^* - g_{\mathcal{P}}^{\widehat{\pi}^*} &\leq 96 t_{\text{minorize}} \sqrt{\frac{2}{n\mathfrak{p}_{\wedge}} \log \frac{2|\mathbf{S}|^2|\mathbf{A}|}{\beta}} \\
\left\| g_{\underline{\widehat{\mathcal{P}}}}^* - g_{\mathcal{P}}^* \right\|_{\infty} &\leq 48 t_{\text{minorize}} \sqrt{\frac{2}{n\mathfrak{p}_{\wedge}} \log \frac{2|\mathbf{S}|^2|\mathbf{A}|}{\beta}}
\end{aligned} \tag{D.31}$$

*with probability at least $1 - \beta$. Hence, the sample complexity of achieving an $\varepsilon$-error in both optimal policy and value estimation is*

$$n = \frac{c \cdot t_{\text{minorize}}^2}{\mathfrak{p}_{\wedge}\varepsilon^2} \log \frac{2|\mathbf{S}|^2|\mathbf{A}|}{\beta}.$$

*where $c = 2 \cdot 96^2$ and $c = 2 \cdot 48^2$ repectively.*

*Proof.* For policy evaluation, consider policy $\widehat{\pi}^*$ returned by Algorithm 3, by Lemma 3.6, we know $\widehat{\pi}^*$ is an optimal policy for the anchored empirical uncertainty set $\underline{\widehat{\mathcal{P}}}$:

$$g_{\underline{\widehat{\mathcal{P}}}}^{\widehat{\pi}^*} = g_{\underline{\widehat{\mathcal{P}}}}^*$$

further

$$g^*_{\mathcal{P}} - g^{\widehat{\pi}^*}_{\mathcal{P}} = \max_{\pi \in \Pi} g^\pi_{\mathcal{P}} - g^{\widehat{\pi}^*}_{\mathcal{P}}$$

$$= \max_{\pi \in \Pi} g^\pi_{\mathcal{P}} - \max_{\pi \in \Pi} g^\pi_{\underline{\widehat{\mathcal{P}}}} + g^{\widehat{\pi}^*}_{\underline{\widehat{\mathcal{P}}}} - g^{\widehat{\pi}^*}_{\mathcal{P}} \tag{D.32}$$

$$\leq 2 \max_{\pi \in \Pi} \left\| g^\pi_{\underline{\widehat{\mathcal{P}}}} - g^\pi_{\mathcal{P}} \right\|_\infty$$

For average value function evaluation $g^*_{\underline{\widehat{\mathcal{P}}}}$:

$$\left\| g^*_{\underline{\widehat{\mathcal{P}}}} - g^*_{\mathcal{P}} \right\|_\infty = \left\| \max_{\pi \in \Pi} g^\pi_{\underline{\widehat{\mathcal{P}}}} - \max_{\pi \in \Pi} g^\pi_{\mathcal{P}} \right\|_\infty$$

$$\leq \max_{\pi \in \Pi} \left\| g^\pi_{\underline{\widehat{\mathcal{P}}}} - g^\pi_{\mathcal{P}} \right\|_\infty \tag{D.33}$$

Then, we analysis $\left\| g^\pi_{\underline{\widehat{\mathcal{P}}}} - g^\pi_{\mathcal{P}} \right\|_\infty$, by Lemma D.7, we know $g^\pi_{\underline{\widehat{\mathcal{P}}}} = \xi V^\pi_{\widehat{\mathcal{P}}}(s_0)$, then:

$$\left\| g^\pi_{\underline{\widehat{\mathcal{P}}}} - g^\pi_{\mathcal{P}} \right\|_\infty = \left\| \xi V^\pi_{\widehat{\mathcal{P}}}(s_0) - \xi V^\pi_{\mathcal{P}}(s_0) + \xi V^\pi_{\mathcal{P}}(s_0) - g^\pi_{\mathcal{P}} \right\|_\infty$$

$$\leq \left\| \xi V^\pi_{\widehat{\mathcal{P}}}(s_0) - \xi V^\pi_{\mathcal{P}}(s_0) \right\|_\infty + \left\| \xi V^\pi_{\mathcal{P}}(s_0) - g^\pi_{\mathcal{P}} \right\|_\infty \tag{D.34}$$

Since for all $Q \in \mathcal{P}$, by Proposition B.1, we know $p_{s,a} \ll q_{s,a}$ for all $(s,a) \in \mathbf{S} \times \mathbf{A}$. Then by Lemma A.4, for all $Q \in \mathcal{P}$, $Q$ is also uniformly ergodic, and $\mathrm{Span}\,(g^\pi_{\mathcal{P}}) = 0$. Thus:

$$\left\| g^\pi_{\underline{\widehat{\mathcal{P}}}} - g^\pi_{\mathcal{P}} \right\|_\infty \leq \xi \left\| V^\pi_{\widehat{\mathcal{P}}} - V^\pi_{\mathcal{P}} \right\|_\infty + \left\| \xi V^\pi_{\mathcal{P}}(s_0) - g^\pi_{\mathcal{P}}(s_0) \right\|_\infty$$

$$\leq \xi \left\| V^\pi_{\widehat{\mathcal{P}}} - V^\pi_{\mathcal{P}} \right\|_\infty + \left\| \xi V^\pi_{\mathcal{P}} - g^\pi_{\mathcal{P}} \right\|_\infty. \tag{D.35}$$

With the choice $\xi = \frac{1}{\sqrt{n}}$, by Lemma C.3 and Lemma C.7, when $n \geq \frac{32}{\mathfrak{p}_\wedge} \log \frac{2|\mathbf{S}|^2|\mathbf{A}|}{\beta}$, with probability $1 - \beta$, for all $\pi \in \Pi$:

$$\left\| g^\pi_{\underline{\widehat{\mathcal{P}}}} - g^\pi_{\mathcal{P}} \right\|_\infty \leq \frac{18 t_{\mathrm{minorize}}}{\sqrt{n}} + 18 t_{\mathrm{minorize}} \sqrt{\frac{8}{n \mathfrak{p}_\wedge} \log \frac{2|\mathbf{S}|^2|\mathbf{A}|}{\beta}}$$

$$= \frac{18 t_{\mathrm{minorize}}}{\sqrt{n}} \left( 1 + \sqrt{\frac{8}{\mathfrak{p}_\wedge} \log \frac{2|\mathbf{S}|^2|\mathbf{A}|}{\beta}} \right) \tag{D.36}$$

$$\leq 24 t_{\mathrm{minorize}} \sqrt{\frac{8}{n \mathfrak{p}_\wedge} \log \frac{2|\mathbf{S}|^2|\mathbf{A}|}{\beta}}$$

$$\leq 48 t_{\mathrm{minorize}} \sqrt{\frac{2}{n \mathfrak{p}_\wedge} \log \frac{2|\mathbf{S}|^2|\mathbf{A}|}{\beta}}$$

Thus, combine the results:

$$g^*_{\mathcal{P}} - g^{\widehat{\pi}^*}_{\mathcal{P}} \leq 2 \max_{\pi \in \Pi} \left\| g^\pi_{\widehat{\mathcal{P}}} - g^\pi_{\mathcal{P}} \right\|_\infty \leq 96 t_{\mathrm{minorize}} \sqrt{\frac{2}{n \mathfrak{p}_\wedge} \log \frac{2|\mathbf{S}|^2|\mathbf{A}|}{\beta}} \tag{D.37}$$

with probability $1 - \beta$. And when:

$$n = \frac{2 \cdot 96^2 t^2_{\mathrm{minorize}}}{\mathfrak{p}_\wedge \varepsilon^2} \log \frac{2|\mathbf{S}|^2|\mathbf{A}|}{\beta}$$

we get:

$$0 \leq g^*_{\mathcal{P}} - g^{\widehat{\pi}^*}_{\mathcal{P}} \leq \varepsilon$$

with probability $1 - \beta$, $\widehat{\pi}^*$ is an $\varepsilon$-optimal policy.
Simultaneously,

$$\left\| g^*_{\underline{\widehat{\mathcal{P}}}} - g^*_{\mathcal{P}} \right\|_\infty \leq \max_{\pi \in \Pi} \left\| g^\pi_{\underline{\widehat{\mathcal{P}}}} - g^\pi_{\mathcal{P}} \right\|_\infty \leq 48 t_{\mathrm{minorize}} \sqrt{\frac{2}{n \mathfrak{p}_\wedge} \log \frac{2|\mathbf{S}|^2|\mathbf{A}|}{\beta}} \tag{D.38}$$

with probability $1 - \beta$. And when:

$$n = \frac{2 \cdot 48^* t_{\text{minorize}}^2}{\mathfrak{p}_\wedge \varepsilon^2} \log \frac{2|\mathbf{S}|^2 |\mathbf{A}|}{\beta}$$

we get:

$$\left\| g_{\widehat{\mathcal{P}}}^* - g_{\mathcal{P}}^* \right\|_\infty \leq \varepsilon$$

with probability $1 - \beta$, $g_{\widehat{\mathcal{P}}}^*$ is an $\varepsilon$-optimal value function. $\qquad \square$

# E  Uniform Ergodicity of the $f_k$ Uncertainty Set

In this section, we prove the technique results similar in $f_k$-divergence constraints given the Assumption 1 and 2. Like what we have in KL-diverence, we first bound the Radon-Nikodym derivative between perturbed kernel $q_{s,a}$ and nominal $p_{s,a}$.

**Lemma E.1.** *Suppose* $\delta \leq \frac{1}{\max\{8, 4k\} m_\vee^2} \mathfrak{p}_\wedge$*, then for all* $q_{s,a} \in \mathcal{P}_{s,a}$*, the Radon-Nikodym derivative of* $q_{s,a}$ *satisfies:*

$$\left\| \frac{q_{s,a}}{p_{s,a}} \right\|_{L^\infty(p_{s,a})} \geq 1 - \frac{1}{2m_\vee}$$

*for all* $(s,a) \in \mathbf{S} \times \mathbf{A}$

*Proof.* We prove the Lemma by contradiction. For any $(s,a) \in \mathbf{S} \times \mathbf{A}$, consider $q_{s,a} \in \mathcal{P}_{s,a}$, suppose there exists $s' \in \mathbf{S}$, such that

$$r := \frac{q_{s,a}}{p_{s,a}}(s') < 1 - \frac{1}{2m_\vee}$$

Then for any $q_{s,a} \in \mathcal{P}_{s,a}$, we have:

$$
\begin{aligned}
D_{f_k}(q_{s,a} \| p_{s,a}) &= p_{s,a} \left[ f_k \left( \frac{q_{s,a}}{p_{s,a}} \right) \right] \\
&\geq f_k \left( \frac{q_{s,a}}{p_{s,a}}(s') \right) p_{s,a}(s') \\
&\geq f_k(r) \, \mathfrak{p}_\wedge
\end{aligned}
\tag{E.1}
$$

Define the helper function $g_k(t) := \frac{1}{k}(t-1)^2$, when $k \geq 2$, we have:

$$
\begin{aligned}
f_k(t) - g_k(t) &= \frac{t^k - kt + k - 1}{k(k-1)} - \frac{1}{k}(t-1)^2 \\
&= \frac{t^k - kk + k - 1 - (k-1)(t^2 - 2t + 1)}{k(k+1)} \\
&= \frac{t^k - kt + k - 1 - (k-1)t^2 + 2(k-1)t - (k-1)}{k(k-1)} \\
&= \frac{t^k - (k-1)t^2 + (k-2)t}{k(k-1)}
\end{aligned}
\tag{E.2}
$$

The nominator is

$$t \cdot (t^{k-1} - (k-1)t + (k-2)),$$

let

$$p_k(t) := t^{k-1} - (k-1)t + (k-2).$$

Then:

$$\frac{dp_k(t)}{dt} = (k-1)t^{k-2} - (k-1) \leq 0 \quad \text{on} \quad t \in [0,1] \tag{E.3}$$

$$\left. \frac{dp_k(t)}{dt} \right|_{t=1} = 0 \quad \text{and} \quad \frac{dp_k(t)}{dt} \leq 0 \tag{E.4}$$

We conclude that $p_k(t) \geq 0$ on $t \in [0, 1]$, and thus:

$$f_k(t) - g_k(t) = \frac{tp_k(t)}{k(k-1)} \geq 0$$

Hence, when $t = r < 1 - \frac{1}{2m_\vee}$:

$$f_k(r) \geq g_k(r) > g\left(1 - \frac{1}{2m_\vee}\right) = \frac{1}{k} \cdot \frac{1}{4m_\vee^2} = \frac{1}{4km_\vee^2} \geq \frac{\delta}{\mathfrak{p}_\wedge} \qquad \text{(E.5)}$$
$$\Rightarrow f_k(r)\mathfrak{p}_\wedge > \delta$$

However, the above inequality contradict to (E.1) where $f_k(r)\mathfrak{p}_\wedge \leq \delta$.

For the case when $k \in (1, 2)$, consider the function:

$$h_k(t) = \frac{f_k(t)}{(t-1)^2} = \frac{t^k - kt + k - 1}{k(k-1)(t-1)^2} \qquad \text{(E.6)}$$

It is easy to see that $\lim_{t \to 1^-} h_k(t) = \frac{1}{2}$, and $h_k(0) = \frac{1}{k}$. Hence $h_k(0) \geq \lim_{t \to 1^-} h_k(t)$. Then, the derivative:

$$
\begin{aligned}
\frac{dh_k(t)}{dt} &= \frac{(kt^{k-1} - k)k(k-1)(t-1)^2 - (t^k - kt + k - 1)2k(k-1)(t-1)}{k^2(k-1)^2(t-1)^4} \\
&= \frac{(k-2)t^k - kt^{k-1} + kt - k + 2}{k(k-1)(t-1)^3}
\end{aligned}
\qquad \text{(E.7)}
$$

Denote the nominator as:

$$q_k(t) = (k-2)t^k - kt^{k-1} + kt - k + 2,$$

then we have:

$$
\begin{aligned}
\frac{dq_k(t)}{dt} &= k(k-2)t^{k-1} - k(k-1)t^{k-2} + k \\
\frac{d^2q_k(t)}{dt^2} &= k(k-1)(k-2)t^{k-2} - k(k-1)(k-2)t^{k-3} \\
&= k(k-1)(k-2)t^{k-3}(t-1)
\end{aligned}
\qquad \text{(E.8)}
$$

When $1 < k < 2$, the second order derivative $d_t^2 q_k(t) > 0$ on $t \in (0, 1)$. $dq_k(t)$ is monotone increasing, and as $d_t q_k(t)|_{t=1} = 0$, we have $d_t q_k(t) \leq 0$ on $t \in (0, 1]$, which sequentially implies $q_k(t)$ is monotone decreasing,

$$q_k(t) \geq q_k(1) = 0 \quad \text{on } t \in (0, 1]$$

And further $d_t h_k(t) \leq 0$, $h_k(t)$ is monotone decreasing from $\frac{1}{k}$ to $\frac{1}{2}$. Thus, for $1 < k < 2$:

$$\frac{f_k(t)}{(t-1)^2} \geq \frac{1}{2} \Rightarrow f_k(t) \geq \frac{1}{2}(t-1)^2 \qquad \text{(E.9)}$$

Combine the previous results, we have, when $t = r < 1 - \frac{1}{2m_\vee}$

$$f_k(r) \geq \frac{1}{2}(r-1)^2 > \frac{1}{2 \cdot 4m_\vee^2} \geq \frac{\delta}{\mathfrak{p}_\wedge}$$

which contradict to

$$\delta \geq D_{f_k}(q_{s,a} \| p_{s,a}) \geq f_k(r)\mathfrak{p}_\wedge.$$

Consequently, we prove, for all $k \in (1, \infty)$ when $\delta \leq \frac{1}{\cdot \max\{8, 4k\}m_\vee^2}\mathfrak{p}_\wedge$, the Radon-Nikodym derivative of between any $q_{s,a} \in \mathcal{P}_{s,a}$ and $p_{s,a}$ satisfies:

$$\left\| \frac{q_{s,a}}{p_{s,a}} \right\|_{L^\infty(p_{s,a})} \geq 1 - \frac{1}{2m_\vee}.$$

Proved. $\qquad\qquad\qquad\qquad\qquad\qquad\qquad\qquad\qquad\qquad\qquad\qquad\qquad\qquad\qquad \square$

Further, similar with Proposition B.2 we have when $\delta \leq \frac{1}{4 \max\{2,k\} m_\vee^2} \mathfrak{p}_\wedge$, $\mathcal{P}$ is uniformly ergodic with:

$$\sup_{Q \in \mathcal{P}} \max_{\pi \in \Pi} t_{\mathrm{minorize}}(Q_\pi) \leq 2t_{\mathrm{minorize}} \tag{E.10}$$

The idea is as same as the KL-divergence case, as under Assumption 2, the Radon-Nikodym derivative is uniform bounded by $1 - \frac{1}{2m_\vee}$ over $\mathcal{P}$, thus, we have for any $Q \in \mathcal{P}$, $Q_\pi^{m_\pi}(s, s') \geq \frac{p_\pi}{2} P_\pi^{m_\pi}(s, s')$ for all $(s, s') \in \mathbf{S} \times \mathbf{S}$.

**Proposition E.2** (Restatement of Proposition 4.2). *Suppose $P$ is uniformly ergodic, and $\delta \leq \frac{1}{\max\{8,4k\} m_\vee^2} \mathfrak{p}_\wedge$, then $\mathcal{P} = \mathcal{P}(D_{f_k}, \delta)$ is uniformly ergodic and for all $Q \in \mathcal{P}$, and $\pi \in \Pi$:*

$$t_{\mathrm{minorize}}(Q_\pi) \leq 2t_{\mathrm{minorize}} \tag{E.11}$$

*where $t_{\mathrm{minorize}}$ is from Assumption 1.*

*Proof.* By Lemma A.3, since $P$ is uniformly ergodic, then there exists an $(m_\pi, p_\pi)$ pair, such that:

$$\frac{m_\pi}{p_\pi} = t_{\mathrm{minorize}}(P_\pi) \quad \text{and} \quad m_\pi \leq m_\vee$$

For all $Q \in \mathcal{P}$, by Lemma E.1, we have for all $(s, a) \in \mathbf{S} \times \mathbf{A}$,

$$\left\| \frac{q_{s,a}}{p_{s,a}} \right\|_{L^\infty(p_{s,a})} \geq 1 - \frac{1}{2m_\vee}$$

Then, for all state pairs $(s_0, s_{m_\pi}) \in \mathbf{S} \times \mathbf{S}$, consider $Q_\pi^{m_\pi}$, we have:

$$\begin{aligned} Q_\pi^{m_\pi}(s_0, s_{m_\pi}) &\geq \sum_{s_1, s_2, \cdots, s_{m_\pi - 1}} q_{s_0, \pi(s_0)}(s_1) q_{s_1, \pi(s_1)}(s_2) \cdots q_{s_{m_\vee - 1}, \pi(s_{m_\pi - 1})}(s_{m_\pi}) \\ &\geq \frac{p_\pi}{2} \psi(s_{m_\pi}). \end{aligned} \tag{E.12}$$

The proof of the above inequality is the same as in (B.4). This implies that for every policy $\pi \in \Pi$, the perturbed kernel $Q_\pi$ maintains a $(m_\pi, \frac{p_\pi}{2})$-Doeblin condition. Furthermore, the minorization time satisfies:

$$t_{\mathrm{minorize}}(Q_\pi) \leq \frac{m_\pi}{\frac{p_\pi}{2}} \leq 2t_{\mathrm{minorize}}(P_\pi) \leq 2t_{\mathrm{minorize}},$$

where $t_{\mathrm{minorize}} = \max_{\pi \in \Pi} t_{\mathrm{minorize}}(P_\pi)$ by Assumption 1. Thus $t_{\mathrm{minorize}}(Q_\pi)$ is uniformly bounded over $Q \in \mathcal{P}$ and $\pi \in \Pi$:

$$\sup_{Q \in \mathcal{P}} \max_{\pi \in \Pi} t_{\mathrm{minorize}}(Q_\pi) < 2t_{\mathrm{minorize}} < \infty,$$

$\mathcal{P}$ is uniformly ergodic. $\square$

We furthr establish uniform ergodicity properties for both the empirical transition kernel $\widehat{P}$ and its empirical uncertainty set $\widehat{\mathcal{P}}$, serving as a probabilistic counterpart to Theorem B.5. Whle not directly impacting our sample complexity results, this analysis reveals that when the robustness parameter satisfies:

$$\delta \leq \frac{1}{\max\{16, 8k\} m_\vee^2} \mathfrak{p}_\wedge,$$

both $\widehat{\mathcal{P}}$ and $\widehat{\mathcal{P}}$ maintain uniform ergodicity with high probability.

**Corollary E.3.** *Under Assumption 1 and $\delta \leq \frac{1}{\max\{16,8k\} m_\vee^2} \mathfrak{p}_\wedge$, when the smaple size satisfies:*

$$n \geq \frac{32 m_\vee^2}{\mathfrak{p}_\wedge} \log \frac{2|\mathbf{S}|^2|\mathbf{A}|}{\beta}$$

*then:*

- *$\mathcal{P}$ is uniformly ergodic, for any $Q \in \mathcal{P}$ and $\pi \in \Pi$, the minorization time of $Q_\pi$:*

$$t_{\mathrm{minorize}}(Q_\pi) \leq 2t_{\mathrm{minorize}}$$

    *with probability $1$.*

- $\widehat{P}$ is uniformly ergodic and for any $\pi \in \Pi$, the minorization time of $\widehat{P}_\pi$:

$$t_{\text{minorize}}(\widehat{P}_\pi) \leq 2t_{\text{minorize}}$$

with probability $1 - \beta$.

- $\widehat{\mathcal{P}}$ is uniformly ergodic, for any $\widehat{Q} \in \widehat{\mathcal{P}}$ and $\pi \in \Pi$, the minorization time of $\widehat{Q}_\pi$:

$$t_{\text{minorize}}(\widehat{Q}_\pi) \leq 4t_{\text{minorize}}$$

with probability $1 - \beta$.

*Proof.* Since $P$ is uniformly ergodic, by Lemma A.3, for any $\pi \in \Pi$, there exists an $(m_\pi, p_\pi)$ pair such that $\frac{m_\pi}{p_\pi} = t_{\text{minorize}}(P_\pi)$.

First, for $\mathcal{P}$, by Lemma E.1, for all $Q \in \mathcal{P}$, we have:

$$\left\| \frac{q_{s,a}}{p_{s,a}} \right\|_{L^\infty(p_{s,a})} \geq 1 - \frac{1}{2m_\vee}$$

and there exists an $\psi \in \Delta(\mathbf{S})$, for any $(s_0, s_{m_\pi}) \in \mathbf{S} \times \mathbf{A}$:

$$Q_\pi^{m_\pi}(s_0, s_{m_\pi}) \geq \left( 1 - \frac{1}{2m_\vee} \right)^{m_\pi} P_\pi^{m_\pi}(s_0, s_{m_\pi}) \tag{E.13}$$
$$\geq \frac{p_\pi}{2} \psi(s')$$

which implies $Q_\pi$ satisifes $(m_\pi, \frac{p_\pi}{2})$-Doeblin condition with probability 1. And further, we have

$$t_{\text{minorize}}(Q_\pi) \leq \frac{m_\pi}{\frac{p_\pi}{2}} = 2t_{\text{minorize}}$$

And $\mathcal{P}$ is uniformly ergodic.

Second, for empirical kernel $\widehat{P}$, by Lemma B.3, we have when

$$n \geq \frac{32m_\vee^2}{\mathfrak{p}_\wedge} \log \frac{2|\mathbf{S}|^2|\mathbf{A}|}{\beta}$$

empiricla kernel $\widehat{P}_\pi$ satisifes $(m_\pi, \frac{p_\pi}{2})$-Doeblin condition, and $\widehat{P}$ is uniformly ergodic with probability $1 - \beta$.

Third, since when $n \geq \frac{32m_\vee^2}{\mathfrak{p}_\wedge} \log \frac{2|\mathbf{S}|^2|\mathbf{A}|}{\beta}$,

$$P\left( \Omega_{n, \frac{1}{2m_\vee}} \right) \geq 1 - \beta.$$

On $\Omega_{n, \frac{1}{2m_\vee}}$, $\widehat{\mathfrak{p}}_\wedge \geq \left( 1 - \frac{1}{2m_\vee} \right) \mathfrak{p}_\wedge \geq \frac{1}{2}\mathfrak{p}_\wedge$. By Lemma E.1

$$\delta \leq \frac{1}{8 \max\{2, k\} m_\vee^2} \mathfrak{p}_\wedge \leq \frac{1}{4 \max\{2, k\} m_\vee^2} \widehat{\mathfrak{p}}_\wedge.$$

It implies fo all $\widehat{Q} \in \widehat{\mathcal{P}}$, $\widehat{Q}_\pi$ satisifes $(m_\pi, \frac{1}{2}\left( \frac{p_\pi}{2} \right))$-Doeblin condition, $\widehat{\mathcal{P}}$ is uniformly ergodic and

$$\sup_{\widehat{Q} \in \widehat{\mathcal{P}}} \max_{\pi \in \Pi} t_{\text{minorize}}(\widehat{Q}_\pi) \leq 4t_{\text{minorize}}$$

with probability $1 - \beta$. $\qquad\qquad\qquad\qquad\qquad\qquad\qquad\qquad\qquad\qquad\qquad\qquad\square$

# F   Properties of the Bellman Operator: $f_k$-Case

In this section, we target to bound the error between DR discounted Bellman opertaor and empirical DR discounted Bellman operator under $f_k$-divergnce. Similar to the KL-case. We override the notations for the $f_k$-case, and introduce the $f_k$-duality

**Lemma F.1** (Lemma 1 of Duchi and Namkoong [9]). *For any $(s, a) \in \mathbf{S} \times \mathbf{A}$, let $\mathcal{P}_{s,a}$ be the $f_k$-uncertainty set centered at the nominal transition kernel $p_{s,a}$. Then, for any $\delta > 0$, let $k^* = \frac{k}{k-1}$:*

$$\Gamma_{\mathcal{P}_{s,a}}(V) = \sup_{\alpha \in \mathbb{R}} \left\{ \alpha - c_k(\delta) E_{p_{s,a}} \left[ (\alpha - V(S))_+^{k^*} \right]^{\frac{1}{k^*}} \right\} \tag{F.1}$$

*where $c_k(\delta) = (1 + k(k-1)\delta)^{\frac{1}{k}}$, $(\cdot)_+ = \max\{\cdot, 0\}$ and $V : \mathbf{S} \to \mathbb{R}$ is the value function.*

we denote the $f_k$-dual functional under the nominal transition kernel $p_{s,a}$:

$$f(p_{s,a}, V, \alpha) := \alpha - c_k(\delta) p_{s,a} \left[ (\alpha - V)_+^{k^*} \right]^{\frac{1}{k^*}} \tag{F.2}$$

then

$$\Gamma_{\mathcal{P}_{s,a}}(V) = \sup_{\alpha \in \mathbb{R}} f(p_{s,a}, V, \alpha)$$

At the same time, we follow the auxiliary measures and function used in KL-case:

$$\begin{aligned} p_{s,a}(t) &= t\widehat{p}_{s,a} + (1-t)p \\ \Delta p_{s,a} &= \widehat{p}_{s,a} - p_{s,a} \\ g_{s,a}(t, \alpha) &= f(p_{s,a}(t), V, \alpha). \end{aligned} \tag{F.3}$$

When $d < 1$, $p_{s,a}(t) \sim p_{s,a}$ holds for all $(s, a) \in \mathbf{S} \times \mathbf{A}$ and $t \in [0, 1]$ on $\Omega_{n,d}$.

**Lemma F.2.** *For any $\delta$, the supremum of $f(p_{s,a}, V, \alpha)$ is achieved at $\alpha^* \geq \operatorname{ess\,inf}_{p_{s,a}} V$. If $\alpha^* > \operatorname{ess\,inf}_{p_{s,a}} V$, then*

$$c_k(\delta) = \frac{p_{s,a} \left[ (\alpha^* - V)_+^{k^*} \right]^{1 - \frac{1}{k^*}}}{p_{s,a} \left[ (\alpha^* - V)_+^{k^* - 1} \right]}$$

*where $p_{s,a}^*$ defined as below:*

$$p_{s,a}^*(\cdot) := \frac{p_{s,a} \left[ (\alpha - V)_+^{k^* - 1} \mathbb{1}\{\cdot\} \right]}{p_{s,a} \left[ (\alpha - V)_+^{k^* - 1} \right]}$$

*is a worst-case measure. When $\alpha^* = \operatorname{ess\,inf}_{p_{s,a}} V$, the worst-case measure is given as:*

$$p_{s,a}^*(\cdot) = \frac{p_{s,a} \left[ \mathbb{1}\{U \cap \cdot\} \right]}{p_{s,a} \left[ \mathbb{1}\{U\} \right]}$$

*where $U = \left\{ s' : V(s') = \operatorname{ess\,inf}_{p_{s,a}} V \right\}$.*

*Proof.* A directly consequence for $f(p_{s,a}, V, \alpha)$ is

$$f(p_{s,a}, V, \alpha) = \alpha \quad \text{when} \quad \alpha \leq \operatorname{ess\,inf}_{p_{s,a}} V$$

Thus, $f$ is monotone increasing as $\alpha < \operatorname{ess\,inf}_{p_{s,a}} V$, thus, the supremum of $f(p_{s,a}, V, \alpha)$ is achieved at $\alpha^* \geq \operatorname{ess\,inf}_{p_{s,a}} V$. The first order derivative of $f(p_{s,a}, V, \alpha)$ with respect to $\alpha$ is given as:

$$\frac{\partial f(p_{s,a}, V, \alpha)}{\partial \alpha} = 1 - c_k(\delta) \left( p_{s,a} \left[ (\alpha - V)_+^{k^*} \right]^{\frac{1}{k^*} - 1} \cdot p_{s,a} \left[ (\alpha - V)_+^{k^* - 1} \right] \right) \tag{F.4}$$

By Proposition 1 of Duchi and Namkoong [9], the dual form of $f(p_{s,a}, V, \alpha)$ is concave with respect to $\alpha$, and the supremum is achieved at $\alpha^*$ where $\partial_\alpha f(p_{s,a}, V, \alpha)|_{\alpha=\alpha^*} = 0$. Since:

$$\frac{\partial f(p_{s,a}, V, \alpha)}{\partial \alpha} = 1 - c_k(\delta) \left( p_{s,a} \left[ (\alpha e - V)_+^{k^*} \right]^{\frac{1}{k^*} - 1} \cdot p_{s,a} \left[ (\alpha e - V)_+^{k^* - 1} \right] \right). \tag{F.5}$$

By the first order condition, we have:

$$\begin{aligned} 0 &= \frac{\partial f(p_{s,a}, V, \alpha)}{\partial \alpha} \bigg|_{\alpha=\alpha^*} \\ &= 1 - c_k(\delta) \left( p_{s,a} \left[ (\alpha^* e - V)_+^{k^*} \right]^{\frac{1}{k^*} - 1} \cdot p_{s,a} \left[ (\alpha^* e - V)_+^{k^* - 1} \right] \right) \end{aligned} \tag{F.6}$$

Whiche implies:

$$c_k(\delta) = \frac{p_{s,a}\left[(\alpha^* e - V)_+^{k^*}\right]^{1-\frac{1}{k^*}}}{p_{s,a}\left[(\alpha^* e - V)_+^{k^*-1}\right]} \tag{F.7}$$

Then, to show $p_{s,a}^*$ is a worst-case measure, it is sufficient to show $p_{s,a}^*[V] = f(p_{s,a}, V, \alpha^*)$ and $D_{f_k}(p_{s,a}^* \| p_{s,a}) = \delta$. We have:

$$
\begin{aligned}
p_{s,a}^*[V] &= \frac{p_{s,a}\left[(\alpha^* - V)_+^{k^*-1} V\right]}{p_{s,a}\left[(\alpha^* - V)_+^{k^*-1}\right]} \\
&= \frac{p_{s,a}\left[(\alpha^* - V)_+^{k^*-1}(V - \alpha^*)\right]}{p_{s,a}\left[(\alpha^* - V)_+^{k^*-1}\right]} + \frac{p_{s,a}\left[(\alpha^* - V)_+^{k^*-1}\alpha^*\right]}{p_{s,a}\left[(\alpha^* - V)_+^{k^*-1}\right]} \\
&= \alpha^* - \frac{p_{s,a}\left[(\alpha^* - V)_+^{k^*}\right]}{p_{s,a}\left[(\alpha^* - V)_+^{k^*-1}\right]} \\
&\overset{(i)}{=} \alpha^* - c_k(\delta) p_{s,a}\left[(\alpha^* - V)_+^{k^*}\right]^{\frac{1}{k^*}} \\
&= f(p_{s,a}, V, \alpha^*)
\end{aligned}
\tag{F.8}
$$

Where $(i)$ is derived by Equation (F.7). Moreover, by the definition of $f_k$-divergence we have:

$$
\begin{aligned}
D_{f_k}(p_{s,a}^* \| p_{s,a}) &= \frac{1}{k(k-1)} p_{s,a}\left[\left(\frac{(\alpha^* - V)_+^{k^*-1}}{p_{s,a}\left[(\alpha^* - V)_+^{k^*-1}\right]}\right)^k - \frac{k(\alpha^* - V)_+^{k^*-1}}{p_{s,a}\left[(\alpha^* - V)_+^{k^*-1}\right]} + k - 1\right] \\
&= \frac{1}{k(k-1)} p_{s,a}\left[\frac{(\alpha^* - V)^{k^*}}{p_{s,a}\left[(\alpha^* - V)_+^{k^*-1}\right]^k} - \frac{k(\alpha^* - V)_+^{k^*-1}}{p_{s,a}\left[(\alpha^* - V)_+^{k^*-1}\right]} + ke - e\right] \\
&= \frac{1}{k(k-1)}\left(\frac{p_{s,a}\left[(\alpha^* - V)_+^{k^*}\right]}{p_{s,a}\left[(\alpha^* - V)_+^{k^*-1}\right]^k} - \frac{k p_{s,a}\left[(\alpha^* - V)_+^{k^*-1}\right]}{p_{s,a}\left[(\alpha^* - V)_+^{k^*-1}\right]} + k - 1\right) \\
&= \frac{1}{k(k-1)}\left(c_k(\delta) - k + k - 1\right) \\
&= \delta.
\end{aligned}
\tag{F.9}
$$

Here we proved that when $\alpha^* > \operatorname{ess\,inf}_{p_{s,a}} V$:

$$p_{s,a}^*(\cdot) = \frac{p_{s,a}\left[(\alpha^* - V)_+^{k^*-1} \mathbb{1}\{\cdot\}\right]}{p_{s,a}\left[(\alpha^* - V)_+^{k^*-1}\right]}.$$

$p_{s,a}^*$ is a worst-case measure. Further we show when $\alpha^* = \operatorname{ess\,inf}_{p_{s,a}} V$, $p_{s,a}^*$ defined below is a worst-case measure:

$$p_{s,a}^*(\cdot) = \frac{p_{s,a}\left[\mathbb{1}\{U \cap \cdot\}\right]}{p_{s,a}\left[\mathbb{1}\{U\}\right]},$$

where $U = \{s' : V(s') = \operatorname{ess\,inf}_{p_{s,a}} V\}$. As $p_{s,a}^*(V) = \operatorname{ess\,inf}_{p_{s,a}} V$, then we only need to show $D_{f_k}(p_{s,a}^* \| p_{s,a}) \leq \delta$. To show this, we divide $V$ into two cases.

First, if $V$ is a constant function on $\operatorname{supp}(p_{s,a})$, then $U = \mathbf{S}$, then for all $q_{s,a} \in \mathcal{P}_{s,a}$, $q_{s,a}[V] = \operatorname{ess\,inf}_{p_{s,a}} V$, and $p_{s,a}^* = p_{s,a}$, $p_{s,a} \in \mathcal{P}_{s,a}$ is a worst-case measure.

Second, if $V$ is not a constnat, then $\mathbf{S} \setminus U \neq \emptyset$, observe that, by continuity, there exists an $\epsilon_0 > 0$, such that for all $0 < \epsilon \leq \epsilon_0$,

$$\operatorname*{ess\,inf}_{p_{s,a}} V \leq \alpha^* + \epsilon \leq \operatorname*{ess\,inf}_{s \sim p_{s,a}|_{\mathbf{S} \setminus U}} V(s)$$

At the same time, as $f(p_{s,a}, V, \alpha^*) = \sup_\alpha f(p_{s,a}, V, \alpha)$, then

$$\left.\frac{\partial f(p_{s,a}, V, \alpha)}{\partial \alpha}\right|_{\alpha^* + \varepsilon} \leq 0$$

which implies:

$$0 \geq 1 - c_k(\delta) \left( p_{s,a} \left[ (\alpha^* + \varepsilon - V)_+^{k^*} \right]^{\frac{1}{k^*}-1} \cdot p_{s,a} \left[ (\alpha^* + \varepsilon - V)_+^{k^*-1} \right] \right)$$

$$c_k(\delta) \geq \frac{p_{s,a} \left[ (\varepsilon \mathbb{1}\{U\})^{k^*} \right]^{1-\frac{1}{k^*}}}{p_{s,a} \left[ (\varepsilon \mathbb{1}\{U\})^{k^*-1} \right]} = \frac{1}{p_{s,a} \left[ \mathbb{1}\{U\} \right]^{\frac{1}{k^*}}} \tag{F.10}$$

The above inequality holds for all $\varepsilon \to 0$, thus, by $k^* = \frac{k}{k-1}$, we concludes:

$$c_k(\delta) \geq \frac{1}{p_{s,a} \left[ \mathbb{1}\{U\} \right]^{k-1}} \tag{F.11}$$

Then, we can compute the $f_k$-divergence as:

$$\begin{aligned}
D_{f_k}(p_{s,a}^* \| p_{s,a}) &= \frac{1}{k(k-1)} p_{s,a} \left[ \left( \frac{\mathbb{1}\{U\}}{p_{s,a} \left[ \mathbb{1}\{U\} \right]} \right)^k - \frac{k \mathbb{1}\{U\}}{p_{s,a} \left[ \mathbb{1}\{U\} \right]} + k - 1 \right] \\
&= \frac{1}{k(k-1)} \left( \frac{1}{p_{s,a} \left[ \mathbb{1}\{U\} \right]^{k-1}} - k + k - 1 \right) \\
&\overset{(i)}{\leq} \frac{1}{k(k-1)} \left( c_k(\delta) - 1 \right) \\
&= \delta.
\end{aligned} \tag{F.12}$$

Where $(i)$ follows by (F.11). Thus, we proved the Lemma. $\qquad\square$

**Lemma F.3.** *If* $\delta \leq \frac{1}{2k} \mathfrak{p}_\wedge$, *then* $\alpha^* \geq \|V\|_{L^\infty(p_{s,a})}$.

*Proof.* First, if $V$ is essentially the constant, then $\mathrm{Span}(V) = 0$, and

$$f(p_{s,a}, V, \alpha) = \alpha \quad \text{when} \quad \alpha \leq \|V\|_{L^\infty(p_{s,a})}$$

And hence $\alpha^* \geq \|V\|_{L^\infty(p_{s,a})}$.
When $V$ is not essentially the constant, let $U = \{s' : V(s') = \mathrm{ess\,inf}_{p_{s,a}} V\}$, and $\mathbf{S}\backslash U \neq \emptyset$, hence:

$$p_{s,a} \left[ \mathbb{1}\{U\} \right] \leq 1 - \mathfrak{p}_\wedge.$$

Recall Lemma F.2, the worst-case measure is given as:

$$p_{s,a}^*(\cdot) = \frac{p_{s,a} \left[ (\alpha^* - V)_+^{k^*-1} \mathbb{1}\{\cdot\} \right]}{p_{s,a} \left[ (\alpha^* - V)_+^{k^*-1} \right]}$$

Hence:

$$\begin{aligned}
\delta \geq D_{f_k}(p_{s,a}^* \| p_{s,a}) &= \frac{1}{k(k-1)} \left( \frac{1}{p_{s,a} [\mathbb{1}\{U\}]^{k-1}} - 1 \right) \\
&\geq \frac{1}{k(k-1)} \left( \frac{1}{(1-\mathfrak{p}_\wedge)^{k-1}} - 1 \right) \\
&\overset{(i)}{\geq} \frac{1}{k} \mathfrak{p}_\wedge
\end{aligned} \tag{F.13}$$

Inequality $(i)$ is derived by $\frac{1}{(1-x)^c} - 1 \geq \frac{x}{c}$ when $c > 0$. However, the above result contradict to the assumption where $\delta \leq \frac{1}{2k} \mathfrak{p}_\wedge$. Thus, we conclude $\alpha^* > \|V\|_{L^\infty(p_{s,a})}$. $\qquad\square$

As $\alpha^* > \|V\|_{L^\infty(p_{s,a})}$:

$$(\alpha^* - V)_+ = \alpha^* - V > 0$$

holds. Then, we show the following Lemma:

**Lemma F.4.** *Let $p_1, p_2, p \in \Delta(\mathbf{S})$ s.t. $p_1, p_2 \ll p$. Define $\Delta := p_1 - p_2$. Then, for any $V : \mathbf{S} \to \mathbb{R}$ and $k^* > 1$:*

$$\sup_{\alpha > \|V\|_{L^\infty(p)}} \frac{1}{k^*} \left| \frac{\Delta\left[(\alpha - V)^{k^*}\right]}{p\left[(\alpha - V)^{k^* - 1}\right]} \right| \leq \left\| \frac{2\Delta}{p} \right\|_\infty \cdot \text{Span}\,(V) \tag{F.14}$$

*Proof.* Notice that for all $c \in \mathbb{R}$:

$$\sup_{\alpha \geq \|V\|_{L^\infty(p)}} \frac{1}{k^*} \left| \frac{\Delta\left[(\alpha - V)^{k^*}\right]}{p\left[(\alpha - V)^{k^* - 1}\right]} \right| = \sup_{\alpha \geq \|V\|_{L^\infty(p)}} \frac{1}{k^*} \left| \frac{\Delta\left[((\alpha - c) - (V - c))^{k^*}\right]}{p\left[((\alpha - c) - (V - c))^{k^* - 1}\right]} \right| \tag{F.15}$$

let $c = \operatorname{ess\,inf}_p V$ and $V' = V - c$, then $\|V'\|_{L^\infty(p)} = \text{Span}\,(V)$, and we have:

$$\sup_{\alpha \geq \|V\|_{L^\infty(p)}} \frac{1}{k^*} \left| \frac{\Delta\left[(\alpha - V)^{k^*}\right]}{p\left[(\alpha - V)^{k^* - 1}\right]} \right| = \sup_{\alpha \geq \|V\|_\infty} \frac{1}{k^*} \left| \frac{\Delta\left[((\alpha - c) - V')^{k^*}\right]}{p\left[((\alpha - c) - V')^{k^* - 1}\right]} \right|$$

$$= \sup_{\alpha \geq \text{Span}(V)} \frac{1}{k^*} \left| \frac{\Delta\left[(\alpha - V')^{k^*}\right]}{p\left[(\alpha - V')^{k^* - 1}\right]} \right| \tag{F.16}$$

First, consider the case where $\alpha > 2k^* \text{Span}\,(V)$, then, for any $s \in \mathbf{S}$, and $V'(s) > 0$, we have:

$$\frac{\alpha}{V'(s)} \geq 2k^* \geq \frac{1}{1 - 2^{-\frac{1}{k^* - 1}}}$$

$$(\alpha - V')^{k^* - 1} \geq \frac{\alpha^{k^* - 1}}{2} \tag{F.17}$$

And when $V'(s) = 0$, $(\alpha - V'(s))^{k^* - 1} \geq \frac{\alpha^{k^* - 1}}{2}$ holds trivially.

Further, with tayler expansion, there exists a $\xi$ where $\xi(s) \in \left(0, \frac{\text{Span}(V)}{\alpha}\right)$, such that:

$$\left(1 - \frac{V'(s)}{\alpha}\right)^{k^*} = 1 - k^* \frac{V'(s)}{\alpha} + \frac{k^*(k^* - 1)}{2}(1 - \xi(s))^{k^* - 2} \frac{V'(s)^2}{\alpha^2}$$

Then, we can derive that:

$$\sup_{\alpha \geq 2k^* \text{Span}(V)} \frac{1}{k^*} \left| \frac{\Delta\left[(\alpha - V')^{k^*}\right]}{p_{s,a(\tau)}\left[(\alpha - V')\right]^{k^* - 1}} \right|$$

$$\leq \sup_{\alpha \geq 2k^* \text{Span}(V)} \frac{1}{k^*} \cdot \left| \frac{\Delta\left[\alpha^{k^*}\left(1 - k^* \frac{V'}{\alpha} + \frac{k^*(k^* - 1)}{2}(1 - \xi)^{k^* - 2}\frac{V'^2}{\alpha^2}\right)\right]}{p\left[\frac{\alpha^{k^* - 1}}{2}\right]} \right|$$

$$\overset{(i)}{=} \sup_{\alpha \geq 2k^* \text{Span}(V)} \frac{1}{k^*} \cdot \left| \frac{\Delta\left[\alpha^{k^*}\left(-k^* \frac{V'}{\alpha} + \frac{k^*(k^* - 1)}{2}(1 - \xi)^{k^* - 2}\frac{V'^2}{\alpha^2}\right)\right]}{p\left[\frac{\alpha^{k^* - 1}}{2}\right]} \right|$$

$$\leq \frac{1}{k^*} \left\| \frac{\Delta}{p} \right\|_{L^\infty(p)} \cdot \sup_{\alpha \geq 2k^* \text{Span}(V)} \left\| \frac{2\left(-k^* V' \alpha^{k^* - 1} + \frac{k^*(k - 1)}{2}(1 - \xi)^{k^* - 2} V'^2 \alpha^{k^* - 2}\right)}{\alpha^{k^* - 1}} \right\|_{L^\infty(p)}$$

$$\leq \frac{2}{k^*} \left\| \frac{\Delta}{p} \right\|_{L^\infty(p)} \cdot \sup_{\alpha \geq 2k^* \text{Span}(V)} \left\| \frac{k^* V' \alpha^{k^* - 1}\left(1 - \frac{k^* - 1}{2}(1 - \xi)^{k^* - 2}\frac{V'}{\alpha}\right)}{\alpha^{k^* - 1}} \right\|_{L^\infty(p)}$$

$$\overset{(ii)}{\leq} \left\| \frac{2\Delta}{p} \right\|_{L^\infty(p)} \cdot \sup_{\alpha \geq 2k^* \text{Span}(V)} \left\| V'\left(1 - \frac{k^* - 1}{4k^*}(1 - \frac{1}{2k^*})^{k^* - 2}\right) \right\|_{L^\infty(p)}$$

$$\overset{(iii)}{\leq} \left\| \frac{2\Delta}{p} \right\|_{L^\infty(p)} \cdot \text{Span}\,(V)$$

$$\tag{F.18}$$

The equality $(i)$ follows from the property that $\Delta\left[c\right] = 0$ for any constant function $c$, since the difference between two measurres $\Delta$ annihilates constants. The inequality $(ii)$ is obtained by applying the condition $\alpha \geq 2k^*\text{Span}\left(V'\right)$, which ensures sufficient regularization. Finally, $(iii)$ emerges from the fundamental constraint $k^* > 1$ in our parameter condition.

Second, we consider the case where $\text{Span}\left(V\right) \leq \alpha \leq 2k^*\text{Span}\left(V\right)$, actually, the the above result holds trivially:

$$\sup_{\text{Span}(V)\leq\alpha\leq 2k^*\text{Span}(V)} \frac{1}{k^*}\left|\frac{\Delta\left[(\alpha-V')^{k^*}\right]}{p\left[(\alpha-V)^{k^*-1}\right]}\right|$$

$$\leq \sup_{\text{Span}(V)\leq\alpha\leq 2k^*\text{Span}(V)} \frac{1}{k^*}\|\alpha-V'\|_{\infty}\left|\frac{\Delta\left[(\alpha-V')^{k^*-1}\right]}{p\left[(\alpha-V')^{k^*-1}\right]}\right| \qquad \text{(F.19)}$$

$$\leq \left\|\frac{2\Delta}{p}\right\|_{L^{\infty}(p)} \cdot \text{Span}\left(V\right)$$

Combine the previous two cases, we derived the result:

$$\sup_{\alpha>\|V\|_{L^{\infty}(p)}} \frac{1}{k^*}\left|\frac{\Delta\left[(\alpha-V)^{k^*}\right]}{p\left[(\alpha-V)^{k^*-1}\right]}\right| \leq \left\|\frac{2\Delta}{p}\right\|_{L^{\infty}(p)} \cdot \text{Span}\left(V\right)$$

Proved $\qquad\qquad\qquad\qquad\qquad\qquad\qquad\qquad\qquad\qquad\qquad\qquad\qquad\qquad\qquad\qquad\qquad\square$

To establish Lemma F.6 bounding the dual functional difference, we build on Lemma F.4. While this result is analogous to Lemma C.7 for the KL-divergence case, the analysis for $f_k$-divergence requires first applying the Envelope Theorem to characterize the variational behavior of the dual optimization problem.

**Lemma F.5** (Envelope Theorem, Corollary 3 of Milgrom and Segal [26]). *Denote $V$ as:*

$$V(t) = \sup_{x\in X} f(x,t)$$

*Where $X$ is a convex set in a linear space and $f : X \times [0,1] \to \mathbb{R}$ is a concave function. Also suppose that $t_0$, and that there is some $x^* \in X^*(t_0)$ such that $d_t f(x^*, t_0)$ exists. Then $V$ is differentiable at $t_0$ and*

$$\frac{dV(t_0)}{dt} = \frac{\partial f(x^*, t)}{\partial t}$$

**Lemma F.6.** *Let $p_{s,a}$ be the nominal transition kernel, and $\widehat{p}_{s,a}$ be the empirical transition kernel, when $\delta \leq \frac{1}{\max\{8,4k\}m_{\vee}^2}\mathfrak{p}_{\wedge}$, then the below inequality holds*

$$\left|\sup_{\alpha\in\mathbb{R}} f(\widehat{p}_{s,a}, V, \alpha) - \sup_{\alpha\in\mathbb{R}} f(p_{s,a}, V, \alpha)\right| \leq 4d \cdot \text{Span}\left(V\right)$$

*on $\Omega_{n,d}$, when $d \leq \frac{1}{2}$*

*Proof.* Since

$$\delta \leq \frac{1}{\max\{8,4k\}m_{\vee}^2}\mathfrak{p}_{\wedge} \leq \frac{1}{2k},$$

by Lemma F.3, we have $\alpha^* > \|V\|_{L^{\infty}(p_{s,a})}$, thus, we only need to consider the case where $\alpha \geq \|V\|_{L^{\infty}(p_{s,a})}$, then recall

$$g_{s,a}(t,\alpha) = f(p_{s,a}(t), V, \alpha)$$

$$= \alpha - c_k(\delta)p_{s,a}(t)\left[(\alpha-V)^{k^*}\right]^{\frac{1}{k^*}} \qquad \text{(F.20)}$$

is concave with respect to $\alpha$, then denote $G(t)$ and $\alpha^*(t)$ we have:

$$G(t) := \sup_{\alpha\geq\|V\|_{L^{\infty}(p_{s,a})}} g_{s,a}(t,\alpha)$$

$$= g_{s,a}(t, \alpha^*(t)) \qquad \text{(F.21)}$$

Combine previous results, we have:

$$| \sup_{\alpha \in \mathbb{R}} f(\widehat{p}_{s,a}, V, \alpha) - \sup_{\alpha \in \mathbb{R}} f(p_{s,a}, V, \alpha)| = |G(1) - G(0)| = \left| \frac{dG(t)}{dt} \Big|_{t=\tau} \right| \tag{F.22}$$

Then, by Lemma F.5, we have, as $G(t)$ is differentiable on $[0, 1]$, then:

$$\frac{dG(t)}{dt} = \frac{\partial g_{s,a}(t, \alpha)}{\partial t} \Big|_{\alpha = \alpha^*(t)}$$

Then we have:

$$\begin{aligned} \left| \sup_{\alpha \in \mathbb{R}} f(\widehat{p}_{s,a}, V, \alpha) - \sup_{\alpha \in \mathbb{R}} f(p_{s,a}, V, \alpha) \right| &= \left| \frac{dG}{dt} \Big|_{t=\tau} \right| \\ &= \left| \left( \frac{\partial g_{s,a}(t, \alpha)}{\partial t} \Big|_{\alpha = \alpha^*(t)} \right) \Big|_{t=\tau} \right| \\ &= \frac{c_k(\delta)}{k^*} \left| \frac{(\widehat{p}_{s,a} - p_{s,a}) \left[ (\alpha^*(t) - V)^{k^*} \right]}{p_{s,a}(t) \left[ (\alpha - V)^{k^*} \right]^{1 - \frac{1}{k^*}}} \Big|_{t=\tau} \right| \end{aligned} \tag{F.23}$$

Since $\alpha^*(t)$ is the optimal multiplier for $\sup_{\alpha \geq \|V\|_{L^\infty(p_{s,a})}} g_{s,a}(\tau, \alpha)$, Using Lemma F.2, we have, for all $t \in [0, 1]$:

$$c_k(\delta) = \frac{p_{s,a}(\tau) \left[ (\alpha^*(\tau) - V)^{k^*} \right]^{1 - \frac{1}{k^*}}}{p_{s,a}(\tau) \left[ (\alpha^*(\tau) - V)^{k^* - 1} \right]}.$$

Thus:

$$\begin{aligned} |\sup_{\alpha \in \mathbb{R}} f(\widehat{p}_{s,a}, V, \alpha) - \sup_{\alpha \in \mathbb{R}} f(p_{s,a}, V, \alpha)| &= \frac{1}{k^*} \left| \frac{(\widehat{p}_{s,a} - p_{s,a}) \left[ (\alpha^*(\tau) - V)^{k^*} \right]}{p_{s,a}(\tau) \left[ (\alpha^*(\tau) - V)^{k^* - 1} \right]} \right| \\ &\leq \sup_{\alpha \geq \|V\|_{L^\infty(p_{s,a})}} \frac{1}{k^*} \left| \frac{(\widehat{p}_{s,a} - p_{s,a}) \left[ (\alpha - V)^{k^*} \right]}{p_{s,a}(\tau) \left[ (\alpha - V)^{k^* - 1} \right]} \right|. \end{aligned} \tag{F.24}$$

Using the fact $\alpha^* \geq \|V\|_{L^\infty(p_{s,a})}$ by the formula where $k^* = \frac{k}{k-1}$, we know $k > 1$, then, apply Lemma F.4,

$$\sup_{\alpha \geq \|V\|_{L^\infty(p)}} \frac{1}{k^*} \left| \frac{(\widehat{p}_{s,a} - p_{s,a}) \left[ (\alpha - V)^{k^*} \right]}{p_{s,a}(\tau) \left[ (\alpha - V)^{k^* - 1} \right]} \right| \leq \left\| \frac{2(\widehat{p}_{s,a} - p_{s,a})}{p_{s,a}(\tau)} \right\|_{L^\infty(p_{s,a})} \cdot \mathrm{Span}(V)$$

Further, on the events set $\Omega_{n,d}$, we have, when $d \leq \frac{1}{2}$:

$$\left\| \frac{\widehat{p}_{s,a} - p_{s,a}}{p_{s,a}(\tau)} \right\|_{L^\infty(p_{s,a})} \leq \left\| \frac{d \cdot p_{s,a}}{\tau p_{s,a} + (1 - \tau) p_{s,a}(1 - d)} \right\|_{L^\infty(p_{s,a})} \leq \frac{d}{1 - d} \leq 2d.$$

We derived the result:

$$\sup_{\alpha} |f(\widehat{p}_{s,a}, V, \alpha, \alpha) - f(p_{s,a}, V, \alpha)| \leq 4d \cdot \mathrm{Span}(V).$$

Proved. $\qquad\qquad\qquad\qquad\qquad\qquad\qquad\qquad\qquad\qquad\qquad\qquad\qquad\qquad\qquad \square$

**Lemma F.7.** *When $n \geq 32 \mathfrak{p}_\wedge^{-1} \log(2|\mathbf{S}|^2|\mathbf{A}|/\beta)$, then for any $\pi \in \Pi$, the $l_\infty$-error of the empirical DR Bellmam operator $\widehat{\widetilde{\mathcal{T}}}_\gamma^\pi$ and the DR Bellman operator $\widehat{\mathcal{T}}_\gamma^\pi$ can be bounded as:*

$$\left\| \widehat{\widetilde{\mathcal{T}}}_\gamma^\pi(V_P^\pi) - \mathcal{T}_\gamma^\pi(V_P^\pi) \right\|_\infty \leq 12 t_{\mathrm{minorize}}(P_\pi) \sqrt{\frac{8}{n \mathfrak{p}_\wedge} \log \frac{2|\mathbf{S}|^2|\mathbf{A}|}{\beta}}$$

*with probability at least $1 - \beta$, where $P_\pi$ is the transition kernel induced by controlled transition kernel $P$ and policy $\pi$.*

*Proof.* By Union bound and Bernstein's inequality, we have, when:

$$n \geq \frac{32}{\mathfrak{p}_\wedge} \log \frac{2|\mathbf{S}|^2|\mathbf{A}|}{\beta}$$

the relative error of $\widehat{p}_{s,a}(s')$ could be bounded as:

$$
\begin{aligned}
\left| \frac{\widehat{p}_{s,a}(s') - p_{s,a}(s')}{p_{s,a}(s')} \right| &\leq \frac{1}{n\mathfrak{p}_\wedge} \log \frac{2|\mathbf{S}|^2|\mathbf{A}|}{\beta} + \sqrt{\frac{2}{n\mathfrak{p}_\wedge} \log \frac{2|\mathbf{S}|^2|\mathbf{A}|}{\beta}} \\
&\leq \sqrt{\frac{8}{n\mathfrak{p}_\wedge} \log \frac{2|\mathbf{S}|^2|\mathbf{A}|}{\beta}} \\
&\leq \frac{1}{2}.
\end{aligned}
\tag{F.25}
$$

Thus, let $d = \sqrt{\frac{8}{n\mathfrak{p}_\wedge} \log \frac{2|\mathbf{S}|^2|\mathbf{A}|}{\beta}} \leq \frac{1}{2}$, we have:

$$P(\Omega_{n,d}) \geq 1 - \beta.$$

Then, on $\Omega_{n,d}$, by Lemma F.6, we could bound the error as:

$$
\begin{aligned}
\left\| \widehat{\mathcal{T}}_\gamma^\pi(V_P^\pi) - \mathcal{T}_\gamma^\pi(V_P^\pi) \right\|_\infty &= \max_{s\in\mathbf{S}} \left| \sum_{a\in\mathbf{A}} \pi(a|s)\left( r(s,a) + \gamma\Gamma_{\widehat{\mathcal{P}}_{s,a}}(V_P^\pi) \right) - \sum_{a\in\mathbf{A}} \pi(a|s)\left( r(s,a) + \gamma\Gamma_{\mathcal{P}_{s,a}}(V_P^\pi) \right) \right| \\
&\leq \max_{(s,a)\in\mathbf{S}\times\mathbf{A}} \left| \gamma\Gamma_{\widehat{\mathcal{P}}_{s,a}}(V_P^\pi) - \gamma\Gamma_{\mathcal{P}_{s,a}}(V_P^\pi) \right| \\
&\leq \gamma \max_{(s,a)\in\mathbf{S}\times\mathbf{A}} \left| \sup_{\alpha\in\mathbb{R}} f(\widehat{p}_{s,a}, V_P^\pi, \alpha) - \sup_{\alpha\in\mathbb{R}} f(p_{s,a}, V_P^\pi, \alpha) \right| \\
&\leq \gamma \max_{(s,a)\in\mathbf{S}\times\mathbf{A}} \sup_{\alpha\in\mathbb{R}} \left| f(\widehat{p}_{s,a}, V_P^\pi, \alpha) - f(p_{s,a}, V_P^\pi, \alpha) \right| \\
&\overset{(i)}{\leq} \max_{s,a\in\mathbf{S}\times\mathbf{A}} 4d \cdot \mathrm{Span}(V_P^\pi)
\end{aligned}
\tag{F.26}
$$

The inequality $(i)$ is derived by Lemma F.4. Since Lemma A.3, there exists an $(m_\pi, p_\pi)$ pair such that $m_\pi/p_\pi = t_{\mathrm{minorize}}(P_\pi)$, combine Proposition A.5 , when $P_\pi$ is $(m_\pi, p_\pi)$-Doeblin, $\mathrm{Span}(V_P^\pi) \leq 3m_\pi/p_\pi$, we have: when

$$n \geq \frac{32}{\mathfrak{p}_\wedge} \log \frac{2|\mathbf{S}|^2|\mathbf{A}|}{\beta},$$

then

$$
\begin{aligned}
\left\| \widehat{\mathcal{T}}_\gamma^\pi(V_P^\pi) - \mathcal{T}_\gamma^\pi(V_P^\pi) \right\|_\infty &\leq 4d \cdot \mathrm{Span}(V_P^\pi) \\
&\leq 12 t_{\mathrm{minorize}}(P_\pi) \sqrt{\frac{8}{n\mathfrak{p}_\wedge} \log \frac{2|\mathbf{S}|^2|\mathbf{A}|}{\beta}}
\end{aligned}
\tag{F.27}
$$

with probability $1 - \beta$. $\qquad\square$

# G  Sample Complexity Analysis: $f_k$ Uncertainty Set

We proceed with the analysis of the Algorithm1, 2 and 3 in the $f_k$-divergence case.

## G.1  DR-DMDP under $f_k$ Uncertainty Set

Building upon these analytical foundations, we derive the following sample complexity bound for DR-MDPs with $f_k$-divergence uncertainty sets:

**Theorem G.1** (Restatement of Theorem 4.4). *Suppose Assumptions 1, and 2 are in force. Then for any $n \geq 32\mathfrak{p}_\wedge^{-1} \log(2|\mathbf{S}|^2|\mathbf{A}|/\beta)$, the policy $\widehat{\pi}^*$ and value function $V_{\widehat{\mathcal{P}}}^*$ returned by Algorithm 1*

*satisfies:*

$$0 \leq V_{\mathcal{P}}^* - V_{\mathcal{P}}^{\widehat{\pi}^*} \leq \frac{96 t_{\text{minorize}}}{1 - \gamma} \sqrt{\frac{2}{n \mathfrak{p}_\wedge} \log \frac{2|\mathbf{S}|^2|\mathbf{A}|}{\beta}}$$

$$\left\| V_{\widehat{\mathcal{P}}}^* - V_{\mathcal{P}}^* \right\|_\infty \leq \frac{48 t_{\text{minorize}}}{1 - \gamma} \sqrt{\frac{2}{n \mathfrak{p}_\wedge} \log \frac{2|\mathbf{S}|^2|\mathbf{A}|}{\beta}}$$

(G.1)

*with probability $1 - \beta$. Consequently, the sample complexity to achieve $\varepsilon$-optimal policy and value function with probability $1 - \beta$ is:*

$$n \leq \frac{c \cdot t_{\text{minorize}}^2}{(1 - \gamma)^2 \varepsilon^2} \log \frac{2|\mathbf{S}|^2|\mathbf{A}|}{\beta}$$

*where $c = 2 \cdot 96^2$, and $c = 2 \cdot 48^2$ repectively.*

*Proof.* For any $\pi$, as $V_{\mathcal{P}}^\pi$ is the solution to $V_{\mathcal{P}}^\pi = \mathcal{T}_\gamma^\pi(V_{\mathcal{P}})$, by Lemma F.7 when:

$$n \geq \frac{32}{\mathfrak{p}_\wedge} \log \frac{2|\mathbf{S}|^2|\mathbf{A}|}{\beta},$$

with probability $1 - \beta$, we have:

$$\left\| \widehat{\mathcal{T}}_\gamma^\pi(V_{\mathcal{P}}^\pi) - \mathcal{T}_\gamma^\pi(V_{\mathcal{P}}^\pi) \right\|_\infty \leq \sup_{Q \in \mathcal{P}} \left\| \widehat{\mathcal{T}}_\gamma^\pi(V_Q^\pi) - \mathcal{T}_\gamma^\pi(V_Q^\pi) \right\|_\infty$$

$$\leq 12 \sup_{Q \in \mathcal{P}} t_{\text{minorize}}(Q_\pi) \sqrt{\frac{8}{n \mathfrak{p}_\wedge} \log \frac{2|\mathbf{S}|^2|\mathbf{A}|}{\beta}}$$

(G.2)

$$\leq 24 t_{\text{minorize}} \sqrt{\frac{8}{n \mathfrak{p}_\wedge} \log \frac{2|\mathbf{S}|^2|\mathbf{A}|}{\beta}}.$$

Since the above result holds for all $\pi \in \Pi$, then:

$$\max_{\pi \in \Pi} \left\| V_{\widehat{\mathcal{P}}}^\pi - V_{\mathcal{P}}^\pi \right\|_\infty \leq \frac{1}{1 - \gamma} \max_{\pi \in \Pi} \left\| \widehat{\mathcal{T}}_\gamma^\pi(V_{\mathcal{P}}^\pi) - \mathcal{T}_\gamma^\pi(V_{\mathcal{P}}^\pi) \right\|_\infty$$

$$\leq \frac{48 t_{\text{minorize}}}{1 - \gamma} \sqrt{\frac{2}{n \mathfrak{p}_\wedge} \log \frac{2|\mathbf{S}|^2|\mathbf{A}|}{\beta}}.$$

(G.3)

By Lemma D.1, as $\widehat{\pi}^* = \arg \max_{\pi \in \Pi} V_{\widehat{\mathcal{P}}}^\pi$, we have:

$$V_{\mathcal{P}}^* - V_{\mathcal{P}}^{\widehat{\pi}^*} \leq 2 \max_{\pi \in \Pi} \left\| V_{\widehat{\mathcal{P}}}^\pi - V_{\mathcal{P}}^\pi \right\|_\infty.$$

Thus, when:

$$n \geq \frac{32}{\mathfrak{p}_\wedge} \log \frac{2|\mathbf{S}|^2|\mathbf{A}|}{\beta}.$$

we have:

$$V_{\mathcal{P}}^* - V_{\mathcal{P}}^{\widehat{\pi}^*} \leq 96 t_{\text{minorize}} \sqrt{\frac{2}{n \mathfrak{p}_\wedge} \log \frac{2|\mathbf{S}|^2|\mathbf{A}|}{\beta}}$$

with probability $1 - \beta$.
At the same time, when $n \geq 32 \mathfrak{p}_\wedge^{-1} \log(2|\mathbf{S}|^2|\mathbf{A}|/\beta)$, we have:

$$\left\| V_{\widehat{\mathcal{P}}}^* - V_{\mathcal{P}}^* \right\|_\infty \leq \max_{\pi \in \Pi} \left\| V_{\widehat{\mathcal{P}}}^\pi - V_{\mathcal{P}}^\pi \right\|_\infty \leq \frac{48 t_{\text{minorize}}}{1 - \gamma} \sqrt{\frac{2}{n \mathfrak{p}_\wedge} \log \frac{2|\mathbf{S}|^2|\mathbf{A}|}{\beta}}$$

with probability $1 - \beta$ concurrently. $\qquad\square$

## G.2 Reduction to DR-DMDP Approach under $f_k$ Uncertainty Set

**Theorem G.2** (Restatement of Theorem 4.4). *Suppose Assumptions 1, 2 are in force. Then, for any*

$$n \geq \frac{32}{\mathfrak{p}_\wedge} \log \frac{2|\mathbf{S}|^2|\mathbf{A}|}{\beta}$$

*The output policy $\widehat{\pi}^*$ and $\frac{V_{\widehat{\mathcal{P}}}^*}{\sqrt{n}}$ returned by Algorithm 2 satisfies:*

$$0 \leq g_\mathcal{P}^* - g_\mathcal{P}^{\widehat{\pi}^*} \leq 120 t_{\mathrm{minorize}} \sqrt{\frac{2}{n\mathfrak{p}_\wedge} \log \frac{8|\mathbf{S}|^2|\mathbf{A}|}{\beta}}$$

$$\left\| \frac{V_{\widehat{\mathcal{P}}}^*}{\sqrt{n}} - g_\mathcal{P}^* \right\|_\infty \leq 60 t_{\mathrm{minorize}} \sqrt{\frac{2}{n\mathfrak{p}_\wedge} \log \frac{2|\mathbf{S}|^2|\mathbf{A}|}{\beta}} \tag{G.4}$$

*with probability $1 - \beta$, and the sample complexity to achieve $\varepsilon$-optimal policy and value function with probability $1 - \beta$ is:*

$$n = O\left( \frac{t_{\mathrm{minorize}}^2}{\mathfrak{p}_\wedge \varepsilon^2} \log \frac{2|\mathbf{S}|^2|\mathbf{A}|}{\beta} \right).$$

*Proof.* Similar with The proof of Theorem D.4 we have:

$$0 \leq g_\mathcal{P}^* - g_\mathcal{P}^{\widehat{\pi}^*} \leq 2 \max_{\pi \in \Pi} \left\| g_\mathcal{P}^\pi - \frac{V_\mathcal{P}^\pi}{\sqrt{n}} \right\|_\infty + 2 \max_{\pi \in \Pi} \frac{1}{\sqrt{n}} \left\| V_{\widehat{\mathcal{P}}}^\pi - V_\mathcal{P}^\pi \right\|_\infty$$

$$\left\| \frac{V_{\widehat{\mathcal{P}}}^*}{\sqrt{n}} - g_\mathcal{P}^* \right\|_\infty \leq \max_{\pi \in \Pi} \left\| g_\mathcal{P}^\pi - \frac{V_\mathcal{P}^\pi}{\sqrt{n}} \right\|_\infty + \max_{\pi \in \Pi} \frac{1}{\sqrt{n}} \left\| V_{\widehat{\mathcal{P}}}^\pi - V_\mathcal{P}^\pi \right\|_\infty. \tag{G.5}$$

As, with Proposition B.2, we have $\mathcal{P}$ is uniformly ergodic, and for all $Q \in \mathcal{P}$ and $\pi \in \Pi$:

$$t_{\mathrm{minorize}}(Q_\pi) \leq 2 t_{\mathrm{minorize}}$$

Thus:

$$\left\| g_\mathcal{P}^\pi - \frac{V_\mathcal{P}^\pi}{\sqrt{n}} \right\|_\infty \leq \frac{18 t_{\mathrm{minorize}}}{\sqrt{n}}$$

Since by Lemma F.7, we have, when $n \geq \frac{32}{\mathfrak{p}_\wedge} \log \frac{2|\mathbf{S}|^2|\mathbf{A}|}{\beta}$:

$$\max_{\pi \in \Pi} \frac{1}{\sqrt{n}} \left\| V_{\widehat{\mathcal{P}}}^\pi - V_\mathcal{P}^\pi \right\|_\infty \leq \sup_{Q \in \mathcal{P}} \max_{\pi \in \Pi} 24 t_{\mathrm{minorize}}(Q_\pi) \sqrt{\frac{8}{n\mathfrak{p}_\wedge} \log \frac{2|\mathbf{S}|^2|\mathbf{A}|}{\beta}}$$

$$\leq 48 t_{\mathrm{minorize}} \sqrt{\frac{8}{n\mathfrak{p}_\wedge} \log \frac{2|\mathbf{S}|^2|\mathbf{A}|}{\beta}} \tag{G.6}$$

with probability $1 - \beta$. Combine the intermediate results, we have:

$$0 \leq g_\mathcal{P}^* - g_\mathcal{P}^{\widehat{\pi}^*} \leq \frac{36 t_{\mathrm{minorize}}}{\sqrt{n}} + 48 t_{\mathrm{minorize}} \sqrt{\frac{8}{n\mathfrak{p}_\wedge} \log \frac{2|\mathbf{S}|^2|\mathbf{A}|}{\beta}}$$

$$= \frac{36 t_{\mathrm{minorize}}}{\sqrt{n}} \left( 1 + \sqrt{\frac{128}{9\mathfrak{p}_\wedge} \log \frac{2|\mathbf{S}|^2|\mathbf{A}|}{\beta}} \right)$$

$$\overset{(i)}{\leq} 120 t_{\mathrm{minorize}} \sqrt{\frac{2}{n\mathfrak{p}_\wedge} \log \frac{2|\mathbf{S}|^2|\mathbf{A}|}{\beta}} \tag{G.7}$$

$$\left\| \frac{V_{\widehat{\mathcal{P}}}^*}{\sqrt{n}} - g_\mathcal{P}^* \right\|_\infty \overset{(ii)}{\leq} 60 t_{\mathrm{minorize}} \sqrt{\frac{2}{n\mathfrak{p}_\wedge} \log \frac{2|\mathbf{S}|^2|\mathbf{A}|}{\beta}}$$

with probability $1 - \beta$, where $(i)$ and $(ii)$ are derived simply by the trival parameter bound $\mathfrak{p}_\wedge \leq \frac{1}{2}$, $\mathbf{S}, \mathbf{A} \geq 1$, and $\beta < 1$. The sample complexity required for policy evaluation on $\widehat{\pi}^*$ and value evaluation on $\frac{V_{\widehat{\mathcal{P}}}^*}{\sqrt{n}}$ in achieving $\varepsilon$-optimality are:

$$n = \frac{c \cdot t_{\text{minorize}}^2}{\mathfrak{p}_\wedge \varepsilon^2} \log \frac{2|\mathbf{S}|^2|\mathbf{A}|}{\beta},$$

where $c = 2 \cdot 120^2$ and $c = 2 \cdot 60^2$ respectively. $\qquad\qquad\square$

## G.3 Anchored DR-AMDP Approach under $f_k$ Uncertainty Set

**Theorem G.3** (Restatement of Theorem 4.5). *Suppose Assumption 1, and 2 are in force. Then for any*

$$n \geq \frac{32}{\mathfrak{p}_\wedge} \log \frac{2|\mathbf{S}|^2|\mathbf{A}|}{\beta}.$$

*The policy $\widehat{\pi}^*$ and $g_{\widehat{\mathcal{P}}}^*$ returned by Algorithm 3 satisifes:*

$$0 \leq g_{\mathcal{P}}^* - g_{\mathcal{P}}^{\widehat{\pi}^*} \leq 120 t_{\text{minorize}} \sqrt{\frac{2}{n\mathfrak{p}_\wedge} \log \frac{2|\mathbf{S}|^2|\mathbf{A}|}{\beta}}$$

$$\left\| g_{\widehat{\mathcal{P}}}^* - g_{\mathcal{P}}^* \right\|_\infty \leq 60 t_{\text{minorize}} \sqrt{\frac{2}{n\mathfrak{p}_\wedge} \log \frac{2|\mathbf{S}|^2|\mathbf{A}|}{\beta}}.$$

(G.8)

*with probability $1 - \beta$, and the sample complexity to achieve $\varepsilon$-optimal policy and value function with probability $1 - \beta$ is:*

$$n = O\left( \frac{t_{\text{minorize}}^2}{\mathfrak{p}_\wedge \varepsilon^2} \log \frac{2|\mathbf{S}|^2|\mathbf{A}|}{\beta} \right).$$

*Proof.* Similarily with what we have in Theorem D.8, $\widehat{\pi}^*$ is an optimal policy for the anchored empirical uncertainty set $\widehat{\mathcal{P}}$:

$$g_{\widehat{\mathcal{P}}}^* = g_{\widehat{\mathcal{P}}}^{\widehat{\pi}^*}$$

so:

$$g_{\mathcal{P}}^* - g_{\mathcal{P}}^{\widehat{\pi}^*} \leq 2 \max_{\pi \in \Pi} \left\| g_{\widehat{\mathcal{P}}}^\pi - g_{\mathcal{P}}^\pi \right\|_\infty.$$

In addition:

$$\left\| g_{\widehat{\mathcal{P}}}^* - g_{\mathcal{P}}^* \right\|_\infty \leq \max_{\pi \in \Pi} \left\| g_{\widehat{\mathcal{P}}}^\pi - g_{\mathcal{P}}^\pi \right\|_\infty \qquad (\text{G.9})$$

Since $g_{\widehat{\mathcal{P}}}^\pi = \xi V_{\widehat{\mathcal{P}}}^\pi(s_0)$, for any $\pi \in \Pi$

$$\left\| g_{\widehat{\mathcal{P}}}^\pi - g_{\mathcal{P}}^\pi \right\|_\infty = \left\| \xi V_{\widehat{\mathcal{P}}}^\pi(s_0) - \xi V_{\mathcal{P}}^\pi(s_0) + \xi V_{\mathcal{P}}^\pi(s_0) - g_{\mathcal{P}}^\pi \right\|_\infty$$

$$\overset{(i)}{\leq} \left\| \xi V_{\widehat{\mathcal{P}}}^\pi - \xi V_{\mathcal{P}}^\pi \right\|_\infty + \left\| \xi V_{\mathcal{P}}^\pi - g_{\mathcal{P}}^\pi \right\|_\infty \qquad (\text{G.10})$$

$$\leq \xi \left\| V_{\widehat{\mathcal{P}}}^\pi - V_{\mathcal{P}}^\pi \right\|_\infty + \left\| \xi V_{\mathcal{P}}^\pi - g_{\mathcal{P}}^\pi \right\|_\infty$$

where $(i)$ relies to $\mathcal{P}$ is uniformly ergodic, thus, $g_{\mathcal{P}}^\pi(s) = g_{\mathcal{P}}^\pi$ for all $s \in \mathbf{S}$, $g_{\mathcal{P}}^\pi \equiv c$ for some constant $c$.

Then, we have:

$$0 \leq g_{\mathcal{P}}^* - g_{\mathcal{P}}^{\widehat{\pi}^*} \leq 2\xi \max_{\pi \in \Pi} \left\| V_{\widehat{\mathcal{P}}}^\pi - V_{\mathcal{P}}^\pi \right\|_\infty + 2 \max_{\pi \in \Pi} \left\| \xi V_{\mathcal{P}}^\pi - g_{\mathcal{P}}^\pi \right\|_\infty$$

$$\left\| g_{\widehat{\mathcal{P}}}^* - g_{\mathcal{P}}^* \right\|_\infty \leq \xi \max_{\pi \in \Pi} \left\| V_{\widehat{\mathcal{P}}}^\pi - V_{\mathcal{P}}^\pi \right\|_\infty + \max_{\pi \in \Pi} \left\| \xi V_{\mathcal{P}}^\pi - g_{\mathcal{P}}^\pi \right\|_\infty$$

(G.11)

With the choice of $\xi = \frac{1}{\sqrt{n}}$, by Lemma C.3 and Lemma F.7, when $n \geq \frac{32}{\mathfrak{p}_\wedge} \log \frac{2|\mathbf{S}|^2|\mathbf{A}|}{\beta}$, then for all $\pi \in \Pi$:

$$
\begin{aligned}
\left\| g_{\widehat{\mathcal{P}}}^\pi - g_{\mathcal{P}}^\pi \right\|_\infty &\leq 18 \frac{t_{\text{minorize}}}{\sqrt{n}} + 24 t_{\text{minorize}} \sqrt{\frac{8}{n\mathfrak{p}_\wedge} \log \frac{2|\mathbf{S}|^2|\mathbf{A}|}{\beta}} \\
&\leq \frac{18 t_{\text{minorize}}}{\sqrt{n}} \left( 1 + \sqrt{\frac{128}{9\mathfrak{p}_\wedge} \log \frac{2|\mathbf{S}|^2|\mathbf{A}|}{\beta}} \right) \\
&\leq 60 t_{\text{minorize}} \sqrt{\frac{2}{n\mathfrak{p}_\wedge} \log \frac{2|\mathbf{S}|^2|\mathbf{A}|}{\beta}}.
\end{aligned}
\tag{G.12}
$$

with probability $1 - \beta$. We concludes, when:

$$
n \geq \frac{32}{\mathfrak{p}_\wedge} \log \frac{2|\mathbf{S}|^2|\mathbf{A}|}{\beta},
$$

then with probability $1 - \beta$

$$
\begin{aligned}
0 \leq g_{\mathcal{P}}^* - g_{\mathcal{P}}^{\widehat{\pi}^*} &\leq 120 t_{\text{minorize}} \sqrt{\frac{2}{n\mathfrak{p}_\wedge} \log \frac{2|\mathbf{S}|^2|\mathbf{A}|}{\beta}} \\
\left\| g_{\widehat{\mathcal{P}}}^* - g_{\mathcal{P}}^* \right\|_\infty &\leq 60 t_{\text{minorize}} \sqrt{\frac{2}{n\mathfrak{p}_\wedge} \log \frac{2|\mathbf{S}|^2|\mathbf{A}|}{\beta}}.
\end{aligned}
\tag{G.13}
$$

holds. And the sample complexity required to achieve $\varepsilon$-optimality for both policy and average value function is:

$$
n = O\left( \frac{t_{\text{minorize}}^2}{\mathfrak{p}_\wedge \varepsilon^2} \log \frac{|\mathbf{S}|^2|\mathbf{A}|}{\beta} \right) = O\left( \frac{t_{\text{mix}}^2}{\mathfrak{p}_\wedge \varepsilon^2} \log \frac{|\mathbf{S}|^2|\mathbf{A}|}{\beta} \right).
$$

where $t_{\text{mix}} := \max_{\pi \in \Pi} t_{\text{mix}}(P_\pi) < \infty$, since $t_{\text{mix}}$ is equivalent to $t_{\text{minorize}}$ up to a constant, proved. $\qquad\square$

## H    Additional Experiment

To further expand on our results in Section 4, we provide additional experiments on baseline comparison and large-scale MDPs. First, we include comparisons with DR RVI Q-learning [46] on Hard MDP Instance 5.1 as baseline. Table 2 and Figure 3 shows the error performance for DR RVI Q-learning and the two algorithms for $\mathfrak{p}_\wedge = 0.9$ and $\delta = 0.1$. They demonstrated that the error levels of our two algorithms are comparable and significantly outperform the previous baseline.

Table 2: Performance comparison with DR RVI Q-learning.

| Sample | 10 | 32 | 100 | 316 | 1000 | 3162 | 10000 | 31622 | 100000 |
|---|---|---|---|---|---|---|---|---|---|
| DR RVI Q-learning [46] | 1.84e-1 | 7.36e-2 | 7.47e-2 | 6.20e-2 | 5.52e-2 | 4.46e-2 | 3.60e-2 | 3.08e-2 | 2.61e-2 |
| Reduction to DMDP | 1.21e-1 | 5.95e-2 | 5.35e-2 | 2.74e-2 | 1.39e-2 | 6.65e-3 | 3.49e-3 | 3.21e-3 | 1.17e-3 |
| Anchored DR-AMDP | 1.67e-1 | 6.52e-2 | 6.26e-2 | 2.90e-2 | 1.16e-2 | 7.51e-3 | 3.27e-3 | 2.37e-3 | 1.23e-3 |

Further, to demonstrate the capability of our framework in solving large-scale MDPs, we consider a large-scale MDP with 20 states and 30 actions, as in Wang et al. [46]. The nominal transition distribution is specified by $p_{s,a} \sim \mathcal{N}(1, \sigma_{s,a})$, where $\sigma_{s,a} \sim \mathbf{Uniform}[0, 100]$, followed by normalization. We then choose the uncertainty size $\delta = 0.4$ to introduce stronger perturbations, following the setting in Wang et al. [46] under the KL-divergence. Note that although $\delta = 0.4$ violates Assumption 2, the slope of the linear regression of our results on the logarithmic scale remains very close to $-1/2$, further supporting our theoretical guarantees. This observation empirically validates the theoretical results established in our theorems.

Further, to demonstrate the capability of our framework in solving large-scale MDP instances, we consider a large-scale MDP with 20 states and 30 actions, as in Wang et al. [46]. Specifically, Algorithm 2 and Algorithm 3 are evaluated on two distinct large-scale instances, respectively, to verify their effectiveness. The nominal transition distribution is specified by $p_{s,a} \sim \mathcal{N}(1, \sigma_{s,a})$,

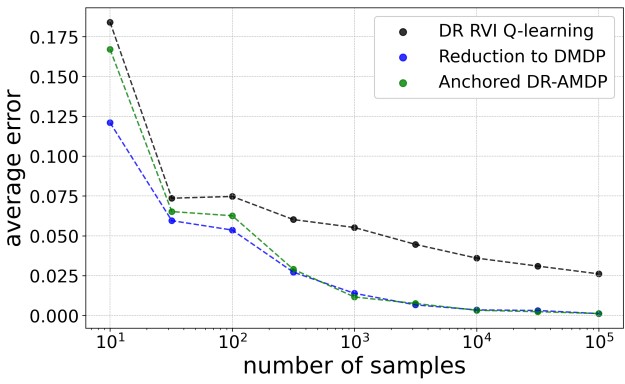

Figure 3: KL-divergence case for Algorithm 2 and 3

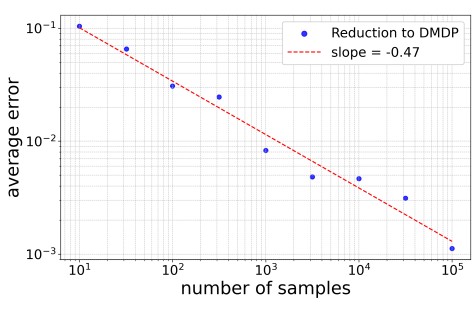

(a) Algorithm 2 for large-scale MDP.

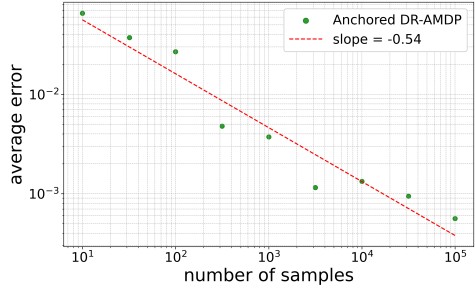

(b) Algorithm 3 for large-scale MDP.

where $\sigma_{s,a} \sim$ **Uniform**$[0, 100]$, followed by normalization. We then choose the uncertainty size $\delta = 0.4$ to introduce stronger perturbations, following the setting in Wang et al. [46] under the KL-divergence. Note that although $\delta = 0.4$ violates Assumption 2, the slope of the linear regression of our results on the logarithmic scale remains very close to $-1/2$, further supporting our theoretical guarantees. This observation empirically validates the theoretical results established in our theorems.

Table 3: Performance on large-scale MDP across different sample sizes.

| Sample | 10 | 32 | 100 | 316 | 1000 | 3162 | 10000 | 31622 | 100000 |
|---|---|---|---|---|---|---|---|---|---|
| Reduction to DMDP | 1.04e-1 | 6.54e-2 | 3.07e-2 | 2.46e-2 | 8.28e-3 | 4.82e-3 | 4.66e-3 | 3.12e-3 | 1.12e-3 |
| Anchored DR-AMDP | 6.55e-2 | 3.72e-2 | 2.69e-2 | 4.79e-3 | 3.73e-3 | 1.15e-3 | 1.33e-3 | 9.42e-4 | 5.61e-4 |

