# OpenReview forum: "Sample Complexity of Distributionally Robust Average-Reward Reinforcement Learning"
_NeurIPS.cc/2025/Conference — NeurIPS 2025 poster_

### Official Review · Reviewer_6nKR · 2025-06-30

**Clarity:** 4
**Significance:** 3
**Originality:** 3
**Rating:** 4
**Confidence:** 4

**Summary:**

The paper considers the nonasymptotic sample complexity of learning near-optimal policies for distributionally robust average-reward MDPs under $f_k$ and KL divergence uncertainty sets. The authors consider the uniformly ergodic setting. They provide two (closely related) algorithms based on discounted reduction, and based on an anchoring-based approximation of the average-reward MDP. They provide an intermediate result showing a refinement of the sample complexity of discounted distributionally robust MDPs of independent interest. The paper also provides some numerical experiments.

**Questions:**

1. What dependence on minimum support/minorization is exhibited by the hard MDP instances? Relatedly, what lower bounds exist in the literature for the sample complexity of robust MDPs, both for discounted and for average-reward settings?
2. More description of the technical obstacles encountered and overcome by the authors would be beneficial in appreciating the level of contribution of this paper. In particular, how do the obstacles faced by the authors differ from those encountered in other papers on (non-robust) average-reward MDPs? Also, it would be appreciated if the authors could discuss why they did not present results which hold for weakly-communicating MDPs.

**Ethical Concerns:**

["NO or VERY MINOR ethics concerns only"]

**Final Justification:**

The proof sketch details provided by the authors in the discussion substantially clarify my questions about the novelty of the paper's techniques relative to the non-robust setting, and will improve the contribution of the paper.

**Limitations:**

yes

**Quality:**

3

**Strengths And Weaknesses:**

Strengths:
I think the paper is written clearly. The algorithms are natural and the main results seem to have good significance, and therefore I believe that this paper will lead to future work on this problem. While I have some doubt about the level of technical obstacles faced (below), I believe this is mitigated by the fact that this natural problem has not been previously addressed from a sample complexity perspective.

Weaknesses:
The results depend on the minimum support parameter, and they also depend on the square of the minorization time (rather than linearly on the minorization time, which is the correct scaling for the non-robust case). It is unclear whether these are the correct dependencies, and they are not explored theoretically by the paper as no lower bounds are provided (except for the trivial application of the non-robust lower bound). This is mitigated somewhat by the experiments which vary the minorization time/minimum support parameter (which coincide for this instance), but the dependence is not explored clearly (there are no experiments where the minorization time is the x-axis). It would be nice to have more discussion of lower bounds within the related work section. The results are in terms of the uniform mixing time, while work on non-robust average-reward MDPs has worked on sharper parameters (such as the optimal bias span) which are finite for weakly communicating MDPs. Overall, I have some doubt of the level of technical obstacles faced in obtaining the results of the paper and how they differ from obstacles which have been already solved for non-robust average-reward MDPs.

Other minor comments:
1. It should be clarified in table 1 that only corollaries for the mixing time are presented, since (especially for standard MDPs) there are several other complexity parameters which are considered
2. I don’t understand the point made on line 168 that the minoration time offers sharper theoretical insights, it seems to be undermined by the subsequent claim of equivalence up to constants between the mixing and minoration times.
3. Maybe for clarity it should be emphasized in definition 3 that this is a maximum over all deterministic policies

---

> ### Author Rebuttal · Authors · 2025-07-31
>
> We greatly appreciate the reviewer’s time and effort in evaluating our work. We hope that the following response can address some of your concerns:
>
> **Question 1 (a): Dependence on the various parameters**
>
> Regarding the dependence on $p_\wedge$, several prior works [1, 2, 3]-show that even for DMDP under the KL-divergence, the sample complexity lower bound necessarily depends on $1/p_\wedge$. Therefore, we believe this dependence may be inherent and possibly unavoidable for the KL setting. Moreover, [4] has also shown that in the $f_k$ divergence setting, the statistical risk of DR functional estimation converges like $n^{(1-k)/k}$ when $k > 1$ is close to 1, without the minimum support probability assumption, suggesting the impossibility of obtaining a parametric rate without this $p_\wedge$ constraint. In fact, a careful observation would reveal an estimation error convergence rate for k = 1 in the setting of [4] behaving like $\widetilde \Theta(1/(p_\wedge n^{1/2}))$. This suggests that the $1/p_\wedge$ could be necessary in a lower bound. However, due to the highly intertwined relationship between the minorization time $t_{\mathrm{minorize}}$ and $p_\wedge$ in our setting, it is unclear what a lower bound would look like.
>
> **Question 1 (b): Lower bounds**
>
> We mention that robustness presents additional statistical challenges to optimal policy learning. For instance, in the standard MDP setting, the minimax sample complexity of policy learning in AMDP setting is $t_{\mathrm{mix}}/\epsilon^2$, which can be obtained by a reduction method from AMDP setting where the sample complexity scales like $t_{\mathrm{mix}}(1-\gamma)^{-2}\epsilon^{-2}$. However, as noted in prior work (e.g. [2], which studies the chi-square case as a special instance of our framework), the sample complexity lower bound for DR-RL under this setting includes a factor of $(1-\gamma)^{-4}\epsilon^{-2}$. Viewing $(1-\gamma)\epsilon$ as the relative error, the extra powers $(1-\gamma)^{-2}$ could be suggesting a $t_{\mathrm{mix}}^2$ dependence in the worst case, potentially matching our upper bound in this regard, reflecting a non-trivial deviation of the robust MDP setting from the non-robust counterpart.
>
> **Question 2 (a): Technical challenges**
>
> In summary, our work makes a novel contribution by providing the first non-asymptotic, prior knowledge free algorithms for DR-AMDP under weak assumptions and provable sample complexity guarantees, while addressing technical obstacles that prior works have not resolved.
>
> 1. **Weaker Assumption:** We only make the ergodicity assumption on the nominal transition kernel $P$, and demonstrate the first sample complexity bounds for DR-AMDP. In contrast, previous work under DR setting relies on the assumptions that the mixing time is uniformly bounded over the uncertainty set $\mathcal P$. However, in reality, this assumption is unrealistic and too strong because it is very difficult for the system to discriminate the properties of all kernels in an uncertainty set. Whereas our DR work does not rely on this stringent assumption, we simply assume the ergodicity of the nominal transition kernel $P$; which has never been considered in previous work.
> 2. **Non-trivial Extension to DR setting with optimal rate:** Recent studies [5] have attempted to employ the reduction to DMDP method to the DR-AMDP context. However, this reduction framework only obtains a suboptimal sample complexity bound $\widetilde O(\varepsilon^{-4})$. The fact that prior efforts were unable to achieve the optimal quadratic rate $\widetilde O(\varepsilon^{-2})$ Using these methods highlights the substantial technical barriers involved. This demonstrates that even extending existing techniques to the DR setting is itself a non-trivial and challenging task. The optimal rate achieved in the non-DR setting critically relies on the fact that the error term $(1-\gamma)V_P^\pi - g_P^\pi$ converges to zero at a rate proportional to $O((1-\gamma)t_{\mathrm{minorize}})$ as $\gamma \to 1$. This enables the reduction and anchoring to be feasible.
> However, the same theoretical vehicles implemented in the non-DR MDP setting cannot directly lead to a similar sample complexity reduction in the DR setting, due to the presence of the uncertainty set.
> In DR-RL, the adversary has the capacity to perturb the transition kernel, inducing a sharp increase in mixing time. This effect can be seen in the provided hard MDP instance [2], minorization time tends to infinity when the minimal support $p_\wedge$ tends to $0$ by perturbation, which prevents the reduction to DMDP in the non-DR setting from having a finite sample guarantee. In order to achieve the presented sample complexity result in our work, a substantial effort is required to connect the radius of the uncertainty set, the minorization time, the discount factor, and the minimum support probability. These considerations lead to an optimal $\widetilde O(1/(1-\gamma)^2)$ scaling, matching the optimal dependence on $1/(1-\gamma)$ in the sample complexity lower bound for classical MDPs.
> 3. **The first prior knowledge-free DR algorithms:** Prior work on reductions to DMDP and anchored AMDP often relies on prior knowledge of certain problem-specific parameters to achieve optimal sample complexity in the non-DR setting. For example, Wang [6] requires setting the discount factor $\gamma=1-\Theta (\varepsilon/t_{\mathrm{minorize}})$ that depends on the minorization time parameter. Such information-sensitive algorithms are impractical in real-world scenarios.  In contrast, our approach does not rely on such information, further demonstrating the generality and feasibility of our results. Specifically, we provide a novel theoretical analysis, allowing us to use the DMDP and Anchored AMDP with a much more flexible choice of $\gamma$ and $\eta$ without prior knowledge.
>
> **Question 2 (b): Weakly communicating**
>
> We appreciate your suggestion to include more discussion of sharper parameters, such as bias span, which is used in a weakly communicating setting of classical MDPs. Actually, our theory can be **directly applied** to the setting of weakly communicating robust MDPs under the bias span parameter, provided that the average robust Bellman equation has a constant gain solution. However, the existence of a solution is, in general, invalid, a fact that is known in the stochastic game literature under a multichain setup (note that, in contrast, multichain MDPs always have a solution to the non-robust multichain Bellman equation). It is currently unclear if we can show the existence of a solution to the constant gain robust Bellman equation under a weakly communicating assumption. Hence, we decide not to state our results using bias span, which arises naturally in the weakly communicating setting, for which we do not know if the Bellman equation has a solution.
>
> **Supplementary: Experiment for the dependence of minorization time**
>
> Minimum support probability and minorization time are entangled. Since for hard MDP instances, $t_{\mathrm{minorize}} = 1/(2p_\wedge)$. For this reason, we cannot disentangle the dependence of $t_{\mathrm{minorize}}$ and $p_\wedge$ in hard MDP settings. We present the results for the different $t_{\mathrm{minorize}}$ below. Note that it is hard for us to disentangle the minorization time and the minimum support $p_\wedge$; we cannot clearly show the dependence of the minimization time solely. Nonetheless, we have experimented with different minorization times for reference
>
> | $t_{\mathrm{minorize}}$ | 1      | 5      | 10     | 20     | 40     | 60     | 80     | 100    | 120    | 140    | 160    | 180    | 200    |
> |:-----------------------:|:------:|:------:|:------:|:------:|:------:|:------:|:------:|:------:|:------:|:------:|:------:|:------:|:------:|
> | **Error**               | 9.08e-3| 6.31e-2| 1.04e-1| 2.04e-1| 3.29e-1| 4.05e-1| 4.65e-1| 5.55e-1| 5.65e-1| 6.16e-1| 6.14e-1| 6.78e-1| 6.85e-1|
>
> Thank you for your insightful comments.
>
> [1] Shi, L., et al. The Curious Price of Distributional Robustness in Reinforcement Learning with a Generative Model.
>
> [2] Shi, L., & Chi, Y. (2024). Distributionally robust model-based offline reinforcement learning with near-optimal sample complexity.
>
> [3] Wang, S., Si, N., Blanchet, J., & Zhou, Z. (2024). Sample complexity of variance-reduced distributionally robust Q-learning.
>
> [4] Duchi, J. C., & Namkoong, H. (2021). Learning models with uniform performance via distributionally robust optimization.
>
> [5] Roch, Zachary, et al. "A finite-sample analysis of distributionally robust average-reward reinforcement learning."
>
> [6] Wang, S., Blanchet, J., & Glynn, P. (2023). Optimal sample complexity for average reward markov decision processes.

---

> ### Comment · Reviewer_6nKR · 2025-08-04
>
> I appreciate the responses 1a and 1b, and believe they may strengthen the paper. I am not completely convinced by the responses 2a and 2b:
>
> 2a: The authors cite the (non-peer-reviewed) paper [5] to justify their claim that there are substantial technical obstacles involved in using the discounted reduction approach. However, the paper [5] does not make use of recent developments in the analysis of discounted reductions for (non-robust) average-reward problems (ex. [38, 58] in the numbering of the main paper). Hence this is not sufficient justification. Also, the authors highlight that they develop algorithms for the robust setting which are prior-knowledge free, but as mentioned in their paper this problem of prior knowledge has already been solved in the non-robust setting (see line 95). Therefore it is not clear whether there are any significant new difficulties in the robust setting specifically related to removing prior knowledge. Overall, I still wonder about the difficulty in deriving the results of the present paper for researchers who are well-versed in prior work. As stated in my review, I do not necessarily view this as a fatal issue, but the authors should make clear whether the key techniques behind their results come primarily from prior developments or are new.
>
> 2b: Is this issue still present when using the minimum support lower bound assumption?

---

> > ### Author Response · Authors · 2025-08-04
> > **Clarifications on the Issues**
> >
> > We thank the reviewer for raising these important points, and we are happy to provide clarifications regarding the issues raised.
> >
> > **2a Reply**
> >
> > We would like to emphasize that generalizing results from the non-DR setting to the DR setting is far from straightforward and cannot simply be achieved by building directly on pre-existing non-DR results. There are several fundamental reasons for this:
> > 1. **Uncertainty size and mixing time challenges:** In the DR setting, the presence of the uncertainty size parameter $\delta$ introduces significant challenges. Specifically, the mixing time $t_{\mathrm{mix}}$ can become arbitrarily large on the uncertainty set. This violates key assumptions in non-DR works, which such quantities are assumed to be bounded merely on the nominal kernel $P$, and consequently prevents the direct extension of finite-sample guarantees from the Reduction to DMDP frame in the non-DR setting to the DR case.
> > 2. **Error bounds and optimal rates:** Even with stronger assumptions (e.g., uniformly bounded mixing times or span seminorms of relative value function over the uncertainty set), prior works in DR-RL fail to address the issue of achieving an **optimal rate**. The critical distinction lies in the error bounds:
> >    - For non-DR RL, the error between empirical value function $V_{\widehat P}^\pi$ and value function $V_{P}^\pi$ is $||V_{\widehat P}^\pi - V_P^\pi|| = O(\frac{1}{(1-\gamma)\sqrt n})$.
> >    - However, in DR-RL, prior to our work, the error bound for the empirical robust value function $V_{\widehat {\mathcal P}}^\pi$ compared to $V_{\mathcal P}^\pi$ is at least $||V_{\widehat P}^\pi - V_{\mathcal P}^\pi|| = O(\frac{1}{(1-\gamma)^2\sqrt{n}})$.
> >
> >    This fundamental difference between the non-DR and DR settings makes recent advancements in the analysis of discount reduction for non-DR average-reward problems inapplicable to the DR setting. To achieve the optimal dependence on $\varepsilon$ for the average-reward problem under the DR setting, one must improve the error rate of the robust value function to the error level $O(1/(1-\gamma)\sqrt n)$, which is precisely what our work accomplishes. This improvement is based on a novel optimization result specifically tailored to the DR case, independent of conclusions from non-DR work. We emphasize that this is a core contribution and a key innovation of our study.
> > 3. **Prior knowledge-free novelty:** Actually, [58] inspired us that knowledge-free algorithms for DR-AMDP could be designed. However, the specific techniques we developed to extend from the non-DR to the DR setting-particularly those highlighted in the previous two points-are both innovative and non-trivial. These techniques differ fundamentally from the Reduction to DMDP approach under the non-DR setting. Our knowledge-free algorithms are grounded in these novel analyses, which form the core contribution and key innovation of this work. Furthermore, the absence of prior knowledge is not merely a feature but an inherent property of our approach. Notably, we are the first to establish such algorithms within the DR average-reward setting.
> >
> >
> > **2b Reply**
> >
> > For this question, we would like to emphasize that this issue still present even under the minimum support lower bound assumption. The minimum support lower bound assumption states that the smallest **non-zero** transition probability is $p_\wedge > 0$, but it still allows for certain transition probabilities to be exactly zero. As such, this assumption is irrelevant to determining whether an MDP is weakly communicating.
> >
> > For example, consider the MDP with $\mathbf S = \set{s_1, s_2}$ and $\mathbf A = \set{a_1, a_2}$. The transition kernel is defined as $p_{s_1, a_1}(s_1) = 1$, $p_{s_1, a_2}(s_2) = 1$, $p_{s_2, a_1}(s_2) = 1$, and $p_{s_2, a_2}(s_1)=1$. Under this setup, the MDP is weakly communicating with $p_\wedge = 1$.
> > However, if action $a_1$ is taken at both $s_1$ and $s_2$, the MDP degenerates into a multichain. While the example satisfy the minimum support lower bound assumption with $p_\wedge = 1$, the underlying issue still present.
> >
> > Again, we thank the reviewer for your questions and comments, and hope that our responses answer your questions.

---

> > > ### Comment · Reviewer_6nKR · 2025-08-05
> > >
> > > Regarding 2b, my question was whether assuming a sufficiently small uncertainty size (relative to the minimum support parameter) would be sufficient to ensure a constant gain solution?
> > >
> > > Regarding 2a, the justification provided by the authors lists improvements which have been made in the non-DR setting (improved dependence on $1/(1-\gamma)$, removing prior knowledge), and then makes the claim that they had not yet been shown in the DR setting. While this is true, I feel as if insufficient detail/justification has been provided to support the claim that achieving these improvements in the DR setting is substantially different/more difficult. As the authors repeatedly claim these improvements as "key innovations" of their work, I believe they should make this much more clear. For example, the authors state "These techniques differ fundamentally from the Reduction to DMDP approach under the non-DR setting." Details supporting this claim (and other similar claims) would be greatly appreciated, and in my opinion go a long way in justifying the technical novelty claimed by the authors and hence improving the paper.

---

> ### Author Response · Authors · 2025-08-05
>
> Thank you for the questions and engaging the discussion!
>
> **2a Response**
>
> While our previous explanation may have caused some misunderstanding, we wish to clarify that the DR (distributionally robust) setting is technically distinct from the non-DR case. There are challenges unique to the DR framework that must be addressed before techniques such as Reduction to DMDP and Anchored methods can be effectively applied.
>
> We highlight the differences between the non-DR and DR settings through a proof sketch
>
> First, we need to ensure that all transition kernels within the uncertainty set are mixing, given that only the nominal transition kernel is mixing. We note that if there are kernels within the uncertainty set that induce multichain or even weakly communicating MDPs, the existing literature does not provide a positive answer regarding the existence of a solution to the robust Bellman equation and a stationary optimal policy. In contrast, the Bellman equation always has solutions in the non-DR setting. We will elaborate on this matter in our response **2b**.
>
> Our Proposition 4.2 directly addresses this concern:
>
> > $\sup_{Q, \pi}t_\mathrm{minorize}(Q_\pi)\leq 2t_\mathrm{minorize}$, when $\delta$ satisfies Assumption 2
>
> In Propositions B.2 and E.2 (which restate Proposition 4.2 under KL and $f_k$ divergences, respectively), we show that, under our assumption on $\delta$, any transition kernel $Q \in \mathcal{P}$ has its conditional distribution $q_{s,a}$ (given an $(s,a)$-pair) absolutely continuous with respect to the nominal distribution $p_{s,a}$, with a minimal support of $1-\frac{1}{2m_\vee}$. This ensures that any such $Q$ satisfies the $(m, \frac{p}{2})$-Doeblin condition and is thus uniformly ergodic; that is, $t_\mathrm{minorize}(Q_\pi)$ is uniformly bounded across all $Q \in \mathcal{P}$ and all policies.
>
>
>
> Second, to tackle the nonlinearity introduced by the DR operation in Lemma C.6,
> > Lemma C.6: $|\sup_{\alpha\geq 0}f(\hat p_{s,a}, V, \alpha) - \sup_{\alpha\geq 0}f(p_{s,a}, V, \alpha)| = O(\mathrm{Span}(V)||\frac{\hat p - p}{p}||_{L^\infty(p)} )$ w.p.1.
>
> Here, we use duality, the mean value theorem, and Lemma C.4
>
> > Lemma C.4: $\sup_{\alpha \geq 0}\alpha\frac{(\hat p - p)[e^{-V/\alpha}]}{p[e^{-V/\alpha}]} = O(\mathrm{Span}(V)||\frac{\hat p - p}{p}||_ {L^\infty(p)})$ w.p.1.
>
> to show that the difference between the empirical and nominal transition kernels under KL-divergence can be bounded by $\mathrm{Span}(V)$ and the relative empirical measure estimation error $\sup_{s,a}||\frac{\hat p_{s,a} - p_{s,a}}{p_{s,a}}||_ {L^\infty(p_{s,a})}$. This bound will reduce to a simple application of Hoeffding's inequality and union bound in the non-DR setting.
>
>
> Next, to connect these statistical analyses of DR Bellman operator to the sample complexity, we show in Theorem D.1 and Lemma C.3 that:
> > W.p.1. $V_{\mathcal P}^* - V_{\mathcal P}^{\hat\pi^*} \leq 2\max_{\pi\in \Pi} ||V_{\widehat {\mathcal P}}^\pi - V_{\mathcal P}^\pi|| \leq 2\sup_{\pi, Q}\frac{1}{1-\gamma}|| \widehat{\mathcal T}_ \gamma^\pi (V_Q^\pi) - \mathcal T_ \gamma^\pi(V_Q^\pi)||$
>
> where $Q\in\mathcal{P}$ and $\pi\in\Pi$.  This is an important step that reduces the analysis of policy error to the estimation error of DR Bellman operator evaluated at $V_Q^{\pi}$.
>
> Using equation C.27 and Lemma C.6 $||\widehat{\mathcal T}_ \gamma^\pi (V_Q^\pi) - \mathcal T_\gamma^\pi (V_Q^\pi)||$ can be bounded by $O(\mathrm{Span}(V_Q^\pi)||\frac{\hat p - p}{p}||_ {L^\infty(p)})$. Then, leveraging Proposition 4.2 (each $Q\in\mathcal{P}$ is uniformly ergodic) and the that $||\frac{\hat p - p}{p}||_ {L^\infty(p)} = \widetilde O(n^{-1/2})$, we obtain
>
> > Lemma C.7: $||\widehat{\mathcal T}_ \gamma^\pi (V_Q^\pi) - \mathcal T_ \gamma^\pi (V_Q^\pi)|| = \widetilde O(t_\mathrm{minorize}(Q_\pi)/\sqrt{n})$ with high probability.
>
> So, the estimation error is bounded in $1/(1-\gamma)$.  This leads to the conclusions in Theorems 4.4 and 4.5.
>
> For the $f_k$-divergence case, the overall proof structure is similar, but requires different techniques. In Lemma F.7, we show that the statistical error of estimating the robust Bellman operator can also be controlled by $t_\mathrm{minorize}$, while Lemma F.6 employs the envelope theorem to achieve a refined analysis that works for all $k > 1$. This is a challenging task: according to our best knowledge, no prior literature achieves an analysis that works for all $k>1$. In particular, it is known that for $k$ close to 1, the minimax statistical behavior of estimating the DR functional under $f_k$-divergence can diverge. This issue is addressed by a careful analysis with the introduction of the minimum support probability.

---

> > ### Author Response · Authors · 2025-08-05
> >
> > **2b Response**
> >
> > Whether an optimal policy achieves a constant gain under the assumption that all kernels in the uncertainty sets are weakly communicating remains an open problem. So, even if the center of the distribution ball is weakly communicating and the radius is small to ensure weak communication for all kernels in the uncertainty set, existing theory cannot guarantee the existence of constant gain solutions to the robust Bellman equation. We are currently working on this question, and it is beyond the scope of the present paper.
> >
> > Thank you again for your insightful questions and valuable feedback. We are happy to clarify further concerns.

---

> > > ### Comment · Reviewer_6nKR · 2025-08-06
> > >
> > > I thank the authors for providing these details. If there is space in the final paper then I believe that including even a brief proof sketch like the above, with pointers to key steps and technical differences to the non-DR setting, would greatly improve the paper. My questions have been resolved.

---

> > > > ### Author Response · Authors · 2025-08-06
> > > >
> > > > Thank you for your feedback and valuable suggestions. We are glad that your questions have been resolved. We will carefully include the brief proof sketch in the revised version to ensure clarity. If you have any further questions or suggestions, we would be happy to address them.

---

### Official Review · Reviewer_huvR · 2025-07-03

**Clarity:** 3
**Significance:** 2
**Originality:** 2
**Rating:** 4
**Confidence:** 3

**Summary:**

This paper studies offline RL under the assumption that the environment is modeled as a DR-AMDP. The transition dynamics are non-stationary and lie within a divergence-based uncertainty set, specified using either KL divergence or $f_k$​-divergences. The authors consider both average-reward and discounted-reward settings and propose two algorithms with theoretical sample complexity guarantees.

The first algorithm builds on Q-learning, using a modified distributionally robust Bellman update that incorporates pessimism by finding a transition in the uncertainty set to minimizing the value function. The second algorithm employs an anchoring technique, modifying the transition kernel with a fixed reference state to ensure robustness, and achieves the same sample complexity as the first.

Theoretical analysis shows a near-optimal sample complexity under a uniform ergodicity assumption. Empirical results on a synthetic two-state, two-action MDP validate the theoretical findings by demonstrating that the average error scales with the number of samples at a rate consistent with the theoretical bounds.

**Questions:**

* Does the theoretical guarantee require knowledge of $\delta$, the radius of the uncertainty set?

* Could the authors clarify the difference between DR-RL under the discounted setting and DR-RL under the discounted mixing setting?

* Is the update rule for g^* missing in Algorithm 3?

* A proof sketch should be provided to improve readability, and the novelty of the analysis should be clearly highlighted.

**Ethical Concerns:**

["NO or VERY MINOR ethics concerns only"]

**Final Justification:**

I accept the authors’ argument and will maintain my positive score for this paper.

**Limitations:**

The assumptions of uniform ergodicity and the availability of simulators make the proposed method less tractable for real-world applications.

**Quality:**

3

**Strengths And Weaknesses:**

**Strengths**
* This paper improves the sample complexity bound for offline DR-MDP with the assumption of uniform ergodicity.
*  Empirical results validate the theoretical result, where the average error scales with the number of samples at a rate consistent with the theoretical bounds.
* The proposed algorithms are designed without requiring knowledge of the mixing time.

**Weaknesses**
* The empirical results lack comparisons with other baseline approaches. It would be helpful if the authors could provide results for other algorithms, such as DRVI-LCB and DR Q-learning.

* The novelty of the paper should be clarified. In terms of the algorithm design, the two proposed methods are similar to previous works: Robust Q-Iteration [1] and the Plug-in approach [2].

* The proposed algorithms and theoretical guarantees require access to simulators for generating i.i.d. samples. This setting differs from the traditional offline RL setup, where the transition model must be estimated solely from offline data, making the approach less tractable in real-world scenarios.

* The environment used in the experiment is a simple MDP with only 2 states and 2 actions, which is not representative of most RL applications.

[1] Robust Reinforcement Learning using Offline Data

[2] The Plug-in Approach for Average-Reward and Discounted MDPs: Optimal Sample Complexity Analysis

---

> ### Author Rebuttal · Authors · 2025-07-31
>
> We thank the reviewer for this constructive feedback. We hope the following response can address some of your concerns:
>
> **Weakness 1: Baseline comparisons**
>
> Our experiments primarily validate the theoretical $O(n^{-1/2})$ statistical error rate of our algorithm, empirically demonstrating the first provable parametric non-asymptotic convergence in DR-AMDP. In response, we also include new comparisons with DR RVI Q-learning [5], further strengthening the empirical evaluation and demonstrating the efficiency of our approach.
>
> | Sample           | 10      | 32      | 100     | 316     | 1000    | 3162    | 10000   | 31622   | 100000  |
> |:----------------:|:-------:|:-------:|:-------:|:-------:|:-------:|:-------:|:-------:|:-------:|:-------:|
> |DR RVI Q-learning [5]   | 1.84e-1 | 7.36e-2 | 7.47e-2 | 6.20e-2 | 5.52e-2 | 4.46e-2 | 3.60e-2 | 3.08e-2 | 2.61e-2 |
> | Alg1 & Alg2      | 1.21e-1 | 5.95e-2 | 5.35e-2 | 2.74e-2 | 1.39e-2 | 6.65e-3 | 3.49e-3 | 3.21e-3 | 1.17e-3 |
>
> The error in Alg 1 & Alg 2 is the maximum over both algorithms.
>
> Since DRVI-LCB is an offline RL method, it does not apply to our setting.
>
> **Weakness 2: Clarify novelty; methods similar to prior works.**
>
> In summary, our work makes a novel contribution by providing the first non-asymptotic, prior knowledge-free algorithms for DR-AMDP under weak assumptions and provable sample complexity guarantees, while addressing technical obstacles that prior works have not resolved.
>
> 1. **Weaker Assumption:** We only assume ergodicity for the nominal transition kernel $P$, and provide the first sample complexity bound for DR-AMDP. In contrast, previous DR work requires the mixing time to be uniformly bounded over the uncertainty set, which is unrealistic and too strong in reality. Our approach avoids this stringent assumption by assuming only the ergodicity of $P$, which has never been addressed in previous work.
> 2. **Non-trivial Extension to DR setting with optimal rate:** Recent work [3] attempted to reduce DR-AMDP to DMDP, but only achieved a suboptimal sample complexity of $\widetilde O(\varepsilon^{-4})$, highlighting significant technical barriers to reaching the optimal $\widetilde O(\varepsilon^{-2})$ rate. In standard MDPs, optimal rates rely on fast convergence of the error term $\\|(1-\gamma)V_P^\pi - g_P^\pi\\| \\leq O((1-\gamma)t_{\mathrm{mix}})$ as $\gamma\to 1$, but this approach does not extend to the DR setting due to the added complexity of the uncertainty set. In DR-RL, adversarial perturbations can sharply increase mixing times, making finite-sample guarantees impossible. Achieving our sample complexity results thus requires new analysis to relate the uncertainty set, minorization time, discount factor, and minimum support. Our work overcomes these challenges and achieves the optimal $\widetilde O(1/(1-\gamma)^2)$ scaling, matching the best-known dependence for classical MDPs first, and then extends the reduction framework to DR-AMDP.
> 3. **The first prior knowledge-free DR algorithms:** Prior work on reductions to DMDP and anchored AMDP often relies on knowledge of problem-specific parameters (e.g., [2] needs $\gamma$ set based on $t_{\mathrm{minorize}}$) to achieve optimal sample complexity in the non-DR setting. Such information-sensitive algorithms are impractical in real-world. Our approach removes this requirement, allowing flexible choices of $\gamma$ and $\eta$ without prior knowledge, and offers a more general and practical solution with new theoretical analysis.
>
> Our contribution is mainly on the algorithmic design and theoretical guarantee for the DR-AMDP, which differs from Robust Q-Iteration [1]: designed for DR-DMDP and Plug-in Approach [2]: not robust and doesn't address DR setting, robust extension faces many technical difficulties and complications. Therefore, the results in [1,2] are not applicable to our setting; we need to devise many theoretical innovations to achieve our efficient prior-knowledge free algorithms and analyses.
>
> **Weakness 3: Simulator and offline RL**
>
> This is a great point. We focus on the generative model since it is important in its own right and plays a central role in many landmark achievements of RL (e.g., AlphaGo’s use of simulation-based planning). Moreover, the generative model serves as a crucial foundation for both sequential offline and online DR-RL tasks. Since there are no prior non-asymptotic sample complexity results for DR-AMDP, establishing results in the generative model is a necessary foundation for future, more applied studies. Our work aims to serve as a cornerstone for the field and to facilitate subsequent advances in both offline and online DR-RL.
>
> We agree that this offline RL setup would significantly broaden the scope and practical impact of our work. In the revised manuscript, we will explicitly discuss future work on offline settings.
>
> **Weakness 4: Experiment**
>
> Thank you for the question. We chose a simple MDP with 2 states and 2 actions mainly for theoretical reasons:
> 1. **Theoretical Focus:** As this is a theory-oriented paper, our primary goal is to empirically validate the theoretical relationship between sample size and accuracy. So simple and controlled MDPs enable clear comparison with theory.
> 2. **Choice “Hard” MDPs:** As shown in [4], this "Hard" MDP achieves the maximal lower bound $\widetilde\Omega(t_{\mathrm{minorize}}/\varepsilon^2)$ for the non-DR AMDP, making it highly representative for validating theoretical properties. For theory-focused investigations, we prioritized such “hardest” instances in our empirical analysis. This further justifies our experimental setup.
> 3. **Algorithmic Challenge:** Solving Algorithms 2 and 3 using value iteration in these hard MDPs represents one of the most challenging scenarios in RL, as the agent cannot trivially distinguish the optimal action, further confirming the robustness of our experimental validation.
>
> Additionally, we have conducted further experiments on Large MDP (20 states, 30 actions) as in [5] to support our theoretical result
> | Sample           | 10      | 32      | 100     | 316     | 1000    | 3162    | 10000   | 31622   | 100000  |
> |:----------------:|:-------:|:-------:|:-------:|:-------:|:-------:|:-------:|:-------:|:-------:|:-------:|
> | Large Scale MDP  | 1.04e-1 | 6.54e-2 | 3.07e-2 | 2.46e-2 | 8.28e-3 | 4.82e-3 | 4.66e-3 | 1.32e-3 | 1.12e-3 |
>
> In large instances, the results are consistent with our theoretical guarantee, with the fitted slope in log-scale being close to $-1/2$.
>
> **Question 1: Knowledge on uncertainty set radius**
>
> We would like to clarify that $\delta$ is not a system characteristic parameter, but rather a decision parameter specified by the user to represent how robust the user wants the learned policy to be. This parameter governs the tradeoff between the policy's resilience to environmental distributional shifts and its optimality under the nominal kernel $P$. Importantly, our theoretical guarantee holds for any user-specified $\delta$ within the range specified in assumption 2, which is the regime most relevant for practical applications of DR-AMDP.
>
> **Question 2: Differences: discounted vs. discounted mixing**
>
> The main difference between DR-RL under the discounted and discounted mixing settings lies in the additional mixing time assumption. In the general discounted setting, no assumptions are made about the mixing properties of the MDP, the mixing time may be unbounded under any policy.
>
> In contrast, discounted mixing MDP assumes an explicit mixing or stability property; e.g. the mixing time of the Markov chain induced by any policy is bounded. This additional assumption results in a sharper and more refined complexity bound.
>
> Specifically, in [6], under the mixing time assumption $t_{\mathrm{mix}} \leq (1-\gamma)^{-1}$, the sample complexity upper bound for standard MDP can be improved to $\widetilde O(t_\mathrm{mix}(1-\gamma)^{-2} \epsilon^{-2})$, which is less conservative than the worst case bound of $\widetilde O((1-\gamma)^{-3} \epsilon^{-2} )$ for discounted MDP.
>
> **Question 3: Update rule for $g^*$**
>
> No, the $g^*$ is obtained by solving for the unique solution of the robust Bellman equation. There are many ways to solve (or approximately solve) these equations, including robust relative value iteration (RVI), robust policy iteration (RPI), or approximate dynamic programming (ADP) methods.
>
> **Question 4: Proof sketch and Novelty**
>
> This is a great suggestion. Our proof relies on two key insights:
> 1. **Bounded Minorization Times:** Under the assumption of bounded uncertainty size, the minorization time on the full set of uncertainties can be effectively controlled.
> 2. **Improved Bellman Operator Error Bound:** The Bellman operator error between the empirical and nominal transition kernel can be improved from the usual $1/(1-\gamma)$ dependence to a dependence on $t_{\mathrm{minorize}}$
>
> These two ingredients are central to our analysis and distinguish our work from prior results. We will add a proof sketch and explicitly highlight the novelty of our analysis in the reviewed manuscript to improve its readability and clarity.
>
> **Limitation**
>
> Indeed, assuming uniform ergodicity on the nominal kernel $P$ could be a strong assumption in practice. However, previous work on DR-AMDP makes an even stronger assumption: all kernels in the uncertainty set are uniformly ergodic. Our work is the first to show that a parametric convergence rate is achievable under a weaker assumption that only the nominal kernel $P$ is uniformly ergodic.
>
> [3] Roch, Zachary, et al. A finite-sample analysis of distributionally robust average-reward reinforcement learning.
>
> [4] Wang, Shengbo, et al. Optimal sample complexity for average reward Markov decision processes.
>
> [5] Wang, Yue, et al. Model-free robust average-reward reinforcement learning.
>
> [6] Wang, Shengbo., et al. Optimal sample complexity of reinforcement learning for mixing discounted Markov decision processes.

---

> > ### Comment · Area_Chair_kuSJ · 2025-08-05
> > **Please respond to the rebuttal**
> >
> > Dear Reviewer huvR,
> >
> > The authors have provided a response to your questions. Could you please read their response, see if they have addressed your concerns, and ask other questions if you have any?  Please see the following message from Program Chairs:
> >
> > ******
> > Reviewers are expected to stay engaged in discussions, initiate them and respond to authors’ rebuttal, ask questions and listen to answers to help clarify remaining issues.
> > It is not OK to stay quiet.
> > It is not OK to leave discussions till the last moment.
> > If authors have resolved your (rebuttal) questions, do tell them so.
> > If authors have not resolved your (rebuttal) questions, do tell them so too.
> >
> > *******
> >
> > After that, please submit your Reviewer Acknowledgement also.
> >
> > Best,
> >
> > AC

---

> ### Author Response · Authors · 2025-08-05
>
> Dear Reviewer huvR,
>
> Thank you for reviewing our manuscript and providing feedback.
>
> As the discussion phase ends soon (by 8.6), we hope our responses have addressed your concerns. Please let us know if you need further clarification.
>
> Best regards,
>
> The Authors

---

> > ### Comment · Reviewer_huvR · 2025-08-05
> >
> > I appreciate the author's response and the additional experimental results. Most of my concerns have been addressed, and the large-scale MDP experiment seems promising.
> >
> > I do have a follow-up question regarding the assumption of bounded uncertainty size. Could you clarify where and how this assumption is used in the proof? Additionally, could you provide a justification for why this assumption is acceptable in your setting? Since the uncertainty set for transitions is typically constructed by UCB methods and requires the other theoretical results to ensure proper bounds.

---

> > > ### Author Response · Authors · 2025-08-05
> > >
> > > Thank you for your comments and questions.
> > >
> > > Regarding the size of the uncertainty set, we assume a bound on the radius **only** to ensure that all transition kernels within the uncertainty set are mixing. While the latter is a condition that is theoretically convenient, it is much harder to check in real settings.
> > >
> > > To address this, we provide an easy-to-check condition on $\delta$ under the KL and $f_k$ divergence models that is almost necessary (see, for example, the hard MDP instance). This assumption enables the following result,
> > > > Proposition 4.2: $\sup_{Q\in\mathcal P, \pi\in\Pi}t_\mathrm{minorize}(Q_\pi)\leq 2t_\mathrm{minorize}$, when $\delta$ satisfies the bound in Assumption 2,
> > >
> > > certifying the mixing property for all transition kernels within the uncertainty. Then, Proposition 4.2 becomes a cornerstone to achieve our $\widetilde O(n^{-1/2})$ convergence rate.
> > >
> > > We note that, to our best knowledge, all the existing literature assumes the easier assumption that all transition kernels within the uncertainty set are mixing, and our theorems still hold if we replace Assumption 2 by this latter condition.
> > >
> > >
> > > Moreover, the regime when $\delta$ is close to 0 is the most interesting for real applications. This is because, when $\delta$ is large, the adversary could be very powerful, leading to overly conservative policies, harming the deployment performance.
> > >
> > > Given this, we cannot recommend constructing the uncertainty set using the UCB and LCB in practice, because with a small sample size, the worst-case transition within the confidence set is rarely close to the real environment, leading to overly conservative policies and pessimistic values. Instead, the $\delta$ is usually calibrated using cross-validation. There is a rich literature on distributionally robust optimization (DRO) that addresses the selection of $\delta$.

---

> > > > ### Comment · Reviewer_huvR · 2025-08-05
> > > >
> > > > Thank you for the reply. However, regarding the choice of $\delta$ in the experiments. How did you verify that the selected value of $\delta$ satisfies Assumption 2?

---

> > > > > ### Author Response · Authors · 2025-08-06
> > > > >
> > > > > In our Hard-MDP Instance experiment, we set the uncertainty size to $\delta = 0.1$, matching the order specified in our assumptions. For the Large Scale MDP experiment, we follow the setting $\delta = 0.4$ as in [5], enabling stronger perturbations. Note that our algorithm is entirely prior-knowledge-free, meaning it can be implemented even if Assumption 2 is violated. Our numerical results demonstrate convergence at a $\widetilde O(n^{-1/2})$ rate in both examples, matching our theoretical guarantees. This confirms the validity of our theorems under the condition that all kernels within the uncertainty sets are mixing.
> > > > >
> > > > > [5] Wang, Yue, et al. Model-free robust average-reward reinforcement learning.

---

> > > > > > ### Comment · Reviewer_huvR · 2025-08-06
> > > > > >
> > > > > > I appreciate the authors' response. My concern has been addressed, and I hope the authors will consider including this discussion in the paper.

---

> > > > > > > ### Author Response · Authors · 2025-08-06
> > > > > > >
> > > > > > > Thank you for your feedback and valuable suggestions. We are glad that your concern has been addressed. We will carefully include these discussions in the revised version of the paper. If you have any further questions or suggestions, we would be happy to address them.

---

### Official Review · Reviewer_wDLr · 2025-07-03

**Clarity:** 2
**Significance:** 2
**Originality:** 2
**Rating:** 4
**Confidence:** 3

**Summary:**

This paper investigates distributionally robust (DR) average-reward reinforcement learning and first establishes a sample-complexity $t_{\text{mix}}^{2} \epsilon^{-2}$ for a model-based algorithm in this setting. Building on techniques from the non-robust RL literature, this paper proposes DR-DMDP with Reduction to DMDP, and then propose Anchored DR-AMDP to avoid the costly subproblem required by earlier methods. Both algorithms achieve near-optimal sample complexity and are empirically validated on hard MDP instance.

**Questions:**

What obstacles had to be overcome to adapt the discounted reduction and anchoring techniques to the distributionally robust average-reward RL setting?

Is there a known lower bound for distributionally robust average-reward RL?

**Ethical Concerns:**

["NO or VERY MINOR ethics concerns only"]

**Final Justification:**

I am satisfied with the response and have no unresolved issues other than the novelty of the contribution, as noted in the rebuttal. Consistent with my original review, since the strengths slightly outweigh the weaknesses, I will maintain my score.

**Limitations:**

yes

**Paper Formatting Concerns:**

I don't have any formatting concerns.

**Quality:**

3

**Strengths And Weaknesses:**

Strengths:
The paper introduces DR-DMDP with Reduction to DMDPs and Anchored DR-AMDP, establishing a sample-complexity $t_{\text{mix}}^{2}\,\epsilon^{-2}$ for the first time in the distributionally robust average-reward RL setting. Numerical experiments on a toy hard MDP instance demonstrates theoretical results


Weaknesses:
Both core ideas, the reduction to DMDPs and the anchoring technique, are adapted from earlier work. Consequently, the technical novelty appears to lie mainly in extending existing methods to the distributionally robust domain, and it is unclear how substantial that advance is.

---

> ### Author Rebuttal · Authors · 2025-07-31
>
> We thank the reviewer for the insightful comments. We hope that the following response will help to address some of the concerns.
>
> **Weakness: Limited technical novelty in robust domain extension & Question 1: Obstacles in adapting techniques to robust average-reward RL**
>
> We think your concern regarding the technical novelty of our methods (“Both core ideas, the reduction to DMDPs and the anchoring technique, are adapted from earlier work...”) and the question about the obstacles encountered when adapting these techniques to the distributionally robust (DR) average-reward RL setting are closely related. Therefore, we provide a reorganized response on the novelty and challenges as follows:
> 1. **Weaker Assumption:** We only make the ergodicity assumption on the nominal transition kernel $P$, and demonstrate the first sample complexity bounds for DR-AMDP. In contrast, previous work under the DR setting relies on the assumptions that the mixing time is uniformly bounded over the uncertainty set $\mathcal P$, or assumes an upper bound on the span seminorm of the optimal bias value function. However, this assumption is unrealistic in application and unsatisfactory from a theoretical point of view. In practice, it is difficult to require properties of all kernels within the uncertainty set, while in theory, there is no obvious way to check these properties. Our DR work does not rely on this type of uniform assumption; we simply assume the ergodicity of the nominal transition kernel $P$, which has never been studied or successfully analyzed in previous work.
> 2. **Non-trivial Extension to DR setting:** Although both ideas exist under non-DR, their extension to the DR context is far from straightforward. Recent studies [1] have attempted to employ the reduction to DMDP method to the DR-AMDP context. However, this reduction framework only obtains a suboptimal sample complexity bound $\widetilde O(\varepsilon^{-4})$. The fact that prior efforts were unable to achieve the optimal quadratic rate $\widetilde O(\varepsilon^{-2})$ highlights the technical barriers involved.
> The optimal rate achieved in non-DR setting critically relies on the fact that the error term $(1-\gamma)V_P^\pi - g_P^\pi$ converges to zero at a rate proportional to $O((1-\gamma)t_{\mathrm{minorize}})$ as $\gamma \to 1$. This enables the reduction and anchoring methods. However, the same theoretical vehicles implemented in the standard non-DR MDP setting cannot directly lead to a similar sample complexity reduction in the DR setting, due to the presence of the uncertainty set.
> In DR-RL, the adversary has the capacity to perturb the transition kernel, inducing a sharp increase in mixing time. This effect can be seen in the provided hard MDP instance [2], minorization time tends to infinity when the minimal support probability $p_\wedge$ tends to $0$ by perturbation, preventing the reduction to the DMDP framework in the non-DR setting from having a similar sample guarantee. In order to achieve the presented sample complexity result in our work, a substantial effort is required to connect the radius of the uncertainty set, the minorization time, the discount factor, and the minimum support probability. These considerations lead to an optimal $\widetilde O(1/(1-\gamma)^2)$ scaling, matching the optimal dependence on $1/(1-\gamma)$ in the sample complexity lower bound for classical MDPs.
> 3. **The first priori knowledge free DR algorithms:** Prior work on reductions to DMDP and anchored AMDP often relies on priori knowledge of certain problem-specific parameters to achieve optimal sample complexity in the non-DR setting. For example, Wang [2] requires setting the discount factor $\gamma=1-\Theta (\varepsilon/t_{\mathrm{minorize}})$ that depends on the minorization time parameter. Such information-sensitive algorithms are impractical in real-world scenarios.  In contrast, our approach does not rely on such information, further demonstrating the generality and feasibility of our results. Specifically, we provide a novel theoretical analysis, allowing us to use the DMDP and Anchored AMDP with a much more flexible choice of $\gamma$ and $\eta$ without prior knowledge.
>
> **Question 2: Lower bound for DR average-reward RL**
>
> The landscape of lower bounds in DR-RL setting is not yet fully understood, even for the discounted case.
> We think the average-reward setting could entail more intricate issues, such as the dependence of the mixing parameter, the radius of the uncertainty set, and the minimum support probability. This makes it challenging to establish tight or fine-grained lower bounds.
> Currently, the only known uniform lower bound for DR average-reward RL is the lower bound for non-DR $(\delta=0)$ average-reward RL: $\widetilde O(t_{\mathrm{mix}}/\varepsilon^{2})$. However, it is currently unclear if this is tight for all regimes, and we have some reasons to believe otherwise. Specifically, [3] established a lower for the statistical risks of $\widetilde \Omega(1/(1-\gamma)^{4}\varepsilon^{2})$ for DR-DMDPs with $\delta = O(1)$ under $\chi^2$-divergence uncertainty sets (which corresponds to the case $k=2$ in our $f_k$-divergence setting). In particular, this bound represents a deviation from the standard $\widetilde\Omega(1/(1-\gamma)^3\varepsilon^2)$ lower bound for non-robust MDPs. Therefore, using a reduction argument that views $\varepsilon(1-\gamma)$ as the relative error, from this result in [3], we anticipate a $\Omega(t_{\mathrm{minorize}}^2/{\varepsilon^{2}})$ dependence for the DR average-reward case. However, proving sharp lower bounds that encapsulate this intuition remains an open and intriguing direction for future research.
>
> [1] Roch, Zachary, et al. "A finite-sample analysis of distributionally robust average-reward reinforcement learning."
>
> [2] Wang, S., Blanchet, J., & Glynn, P. (2023). Optimal sample complexity for average reward markov decision processes.
>
> [3] Shi, L., Li, G., Wei, Y., Chen, Y., Geist, M., & Chi, Y. (2023). The Curious Price of Distributional Robustness in Reinforcement Learning with a Generative Model.

---

> > ### Comment · Reviewer_wDLr · 2025-08-04
> >
> > I thank the authors for response. I don't have further questions and maintain my score. But I recommend the authors add a clarification of the improvement compared to prior works in the revised version, even if the authors of prior works could be convinced of the novelty of this work after reading.

---

> > > ### Author Response · Authors · 2025-08-04
> > >
> > > Thank you for your feedback and suggestion. We will add a clarification of the improvements and novelty in the revised version. We appreciate your time and effort.

---

### Official Review · Reviewer_Umdh · 2025-07-04

**Clarity:** 4
**Significance:** 3
**Originality:** 2
**Rating:** 4
**Confidence:** 2

**Summary:**

This work proposes new algorithms with improved sample complexity for DRO RL problems for discounted and average reward settings. The improvements are possible due to recent advances in non-DRO settings, significantly tightening the dependence on the discount factor compared to prior work in the DRO setting and nearly matching the rates in the standard non-DRO setup.

While there are some objective weaknesses concerning the technical novelty compared to a non-DR setting, and limited empirical evaluation, my overall evaluation is positive. The main positive points are the practical importance of the problem, excellent presentation, and the technical depth of the derived results.

**Questions:**

This is a standard thing in DRO, but I could not find the definition of $(s, a)$ rectangular sets mentioned in line 114 of the paper.

Typo on line 145, should be "constant".

Something is wrong in Definition 3.3. It should be either for all policies or maximum over all policies.

What are $Q_{\pi}$ and $Q$ in Proposition 4.2. I assume this is another typo.

**Ethical Concerns:**

["NO or VERY MINOR ethics concerns only"]

**Final Justification:**

I thank the authors for addressing my concerns and answering questions. I appreciate the additional experiments provided and consequently maintain my score.

**Quality:**

3

**Strengths And Weaknesses:**

**Strengths:**
1. The sample complexity in the DR setting is improved for the discounted setting in terms of $1-\gamma$ dependence from $O((1-\gamma)^{-4})$ to $O((1-\gamma)^{-2})$. This is a significant improvement and in addition this allowed them to extend the result to average reward settings via black box reduction.

2. The sample complexity matches the lower bound in non-DR setting, making this extension essentially almost "for free". However, the dependence on the mixing time becomes quadratic instead of being linear; it is unclear to me if this is improvable.

3. The presentation of preliminaries is very useful to understand the key theoretical components.

4. Experimental results are well done, although limited to a toy (hard instance) task. Figure 2 verifies theoretical convergence rates. There are no error bars, but the variability among different numbers of samples is relatively small; this is likely because the task is very simple.

**Weaknesses:**

1. It is difficult for me to judge the technical difficulty of extending the Standard (non-DR) convergence result to this DR-RL setting, but I have the impression that the improvements are largely dependent on the very recent improvement by Want et al [38].

2. Some results should be better discussed in the next revision. For example, how to interpret the results in Table 1. The prior work in this table is mostly discussed in the text, but the positioning of the derived result in the literature is not complete.

3. Why do we need two algorithms in the average reward setting? Is there any advantage of Algorithm 3 over Algorithm 2? The differences should be discussed. What is the difference between them in a large-scale experimental setup?

---

> ### Author Rebuttal · Authors · 2025-07-31
>
> We sincerely thank the reviewer for the thoughtful and constructive feedback, as well as for recognizing the contributions of our work. Here we address the specific weaknesses and questions raised:
>
> **Weakness 1: Technical difficulty of extending non-DR result to DR-RL setting.**
>
> We would like to explicitly clarify several crucial points that distinguish our results from Wang [38], particularly regarding the challenges of extending their approach to the DR-RL setting:
>
> 1. **Non-trivial Extension to DR-RL**: While Wang [38] employs a reduction to DMDP for their analysis, this technique does not straightforwardly extend to the DR-RL setting. The primary reason is that their reduction critically relies on the fact that the error term $(1-\gamma)V_P^\pi - g_P^\pi$ converges to zero at a rate proportional to $O((1-\gamma)t_{\mathrm{minorize}})$ as $\gamma \to 1$. In the non-DR setting, this enables a sample complexity of order $t_{\mathrm{minorize}}/(1-\gamma)^2$, making the reduction feasible.
> However, the same theoretical techniques used by Wang [38] in the standard non-DR MDP setting cannot directly yield a similar sample complexity reduction in the robust setting, due to the presence of the uncertainty set.
> In DR-RL, the adversary has the capacity to perturb the transition kernel, inducing a sharp increase in mixing time. This effect can be seen in the provided hard MDP instance [38], minorization time tends to infinity when the minimal support $p_\wedge$ tends to $0$ by perturbation. In order to achieve the presented sample complexity result in this work, a substantial effort is required to connect the radius of the uncertainty set, the minorization time, the discount factor, and the minimum support probability. These considerations leads to not only a correct $\widetilde O(1/(1-\gamma)^2)$ scaling, but also matching the optimal dependence on $1/(1-\gamma)$ in the sample complexity lower bound for classical MDPs, but also an algorithm that need neither prior knowledge of $t_{\mathrm{minorize}}$ or $t_{\mathrm{mix}}$, nor the condition that all the kernels in the uncertainty set induces mixing. We will elaborate next.
>
> 2. **Priori Knowledge Free**: In [38], the work leverages a restrictive parameter selection $\gamma = 1-\Theta(\varepsilon/ t_{\mathrm{minorize}})$, which requires prior knowledge of minorization time even in the non-DR setting. In contrast, our approach does not rely on such information, leading to a feasible algorithm that can run without such knowledge. This is achieved by a novel theoretical analysis that generalizes the previous approach, allowing us to use the discounted MDP with a much more flexible choice of $\gamma$ (or $\eta$ in the anchored setting). In particular, this allows us to choose $\gamma = 1-\Theta(1/\sqrt{n})$ and $\eta=1/\sqrt{n}$, without prior knowledge, to achieve the claimed rates for DR-AMDP.
>
> **Weakness 2: Table 1 results interpretation is incomplete.**
>
> We thank the reviewer's feedback of our results as presented in Table 1. In the next revision, we will explicitly enhance the discussion of Table 1, both in the introduction and in a dedicated section, With respect to Table 1, our results represent a significant improvement over previous works in terms of both weaker assumptions that we only makes the ergodicity assumption on the nominal transition kernel $P$, and demonstrates the first sample complexity bounds for DR-AMDP. We will make these points more explicit in the revised manuscript and ensure that readers can readily interpret the advantages of our approach.
>
> **Weakness 3: Algorithm 2 vs. Algorithm 3: advantages, differences, scalability.**
>
> We thank the reviewers for their questions and suggestions for improving our experimentation. The primary motivation for including both algorithms is to systematically investigate whether well-established approaches from the non-DR setting—namely, reduction to DMDP and Anchored AMDP—can be effectively extended to the distributionally robust context. Both algorithms are widely recognized for their simplicity and sample efficiency under the non-DR setting, and it is valuable to understand their respective strengths and limitations when adapted to DR-RL.
>
> **The differences and potential advantages between two algorithms are:**
> 1. **Algorithm 2 (Reduction to DMDP)** leverages the connection between average-reward and discounted-reward problems, which can simplify analysis and benefit from existing results in the discounted setting. It can be implemented flexibly using a variety of established techniques, such as Q-learning and value iteration.
> 2. **Algorithm 3 (Anchored AMDP)**, on the other hand, leverages span contraction, which enables fast convergence under the span seminorm. It directly tackles the average-reward problem and may exhibit better empirical performance or stability, especially in environments where the reduction may introduce additional approximation errors.
> We will expand the discussion in the revised version to clearly articulate these differences, including their theoretical guarantees and practical trade-offs.
>
> **On large-scale experimental setups**
>
> To empirically compare the two algorithms under large-scale settings, we conducted example used in [1], which involves 20 states and 30 actions. Our results provide practical insight into their scalability and performance differences in more complex environments. We will include a detailed discussion of these experimental findings in the revised manuscript to further clarify the relative advantages of each algorithm. Actually, even in the large–scale experiments with large state-action space, the slope of the linear regression of our results on the logarithmic scale is very close to -1/2, further validating our theoretical guarantee:
>
> | Sample           | 10      | 32      | 100     | 316     | 1000    | 3162    | 10000   | 31622   | 100000  |
> |:----------------:|:-------:|:-------:|:-------:|:-------:|:-------:|:-------:|:-------:|:-------:|:-------:|
> | Large Scale MDP  | 1.04e-1 | 6.54e-2 | 3.07e-2 | 2.46e-2 | 8.28e-3 | 4.82e-3 | 4.66e-3 | 1.32e-3 | 1.12e-3 |
>
> **Question 1: Definition of $(s,a)$ rectangular sets**
>
> The definition of $(s,a)$-rectangular sets is provided in line 194 of the paper. We will add a clear forward reference in line 114 to enhance the clarity and ensure readers can easily locate the definition.
>
> **Question 2: Typo “constant”**
>
> Thank you for catching this typo. We have corrected the issue in the revised manuscript.
>
> **Question 3: Definition 3.3**
>
> We have revised this definition as follows to improve mathematical rigor.
> > Definition 3.3: And MDP (or its controlled transition kernel $P$ ) is said to be uniformly ergodic if for all policies $\pi \in \Pi$, $t_{\mathrm{minorize}}(P_\pi) < \infty$, denote $t_{\mathrm{minorize}} := \max_{\pi\in\Pi}t_{\mathrm{minorize}}(P_\pi)$
>
> **Question 4: $Q$ and $Q_\pi$**
>
> In this context, $Q:=\set{q_{s,a}\in \Delta(\mathbf S): (s,a)\in \mathbf S\times \mathbf A}$ is an element from the uncertainty set $\mathcal{P}$ consisting of transition kernels. On the other hand, $Q_\pi(s, s’):= \sum_{a\in\mathbf A}\pi(a|s)q_{s,a}(s’)$ is the transition matrix induced by policy $\pi$ and kernel $Q$. We have updated our manuscript to improve clarity.
>
> [1] Wang, Y., Velasquez, A., Atia, G. K., Prater-Bennette, A., & Zou, S. (2023, July). Model-free robust average-reward reinforcement learning.

---

> > ### Comment · Reviewer_Umdh · 2025-08-03
> >
> > I thank the authors for addressing my concerns and answering questions. I appreciate the additional experiments provided and consequently maintain my score.

---

> > > ### Author Response · Authors · 2025-08-04
> > >
> > > Thank you for your feedback and for appreciating our responses and additional experiments. We truly value your time and effort in reviewing our work.

---

### Note · Authors · 2025-08-12

# Final Remarks

We thank the reviewers for their thoughtful discussions and constructive comments. We are pleased to have reached consensus with the reviewers on the conclusions and contributions of our work:

**Algorithmic Contributions:** We propose two concise and effective algorithms for solving DR-AMDP with an $\widetilde O(\epsilon^{-2})$ dependence, establishing the first non-asymptotic sample complexity for learning near-optimal policies in DR-DMDP under both hard KL and $f_k$-divergence uncertainty sets. Importantly, our algorithms do not require any prior knowledge to initialize algorithm parameters.

**Theoretical Advances:** Based on our analytical framework, we address the technical challenge of generalizing from non-DR to DR settings.
1. In the DR context and under the weak assumption of a mixing property on the nominal transition kernel $P$, we prove that the entire uncertainty set $\mathcal{P}$ also ensures uniform mixing. This is the first work to only assume mixing for the nominal kernel while obtaining stronger conclusions, in contrast to previous work that required uniform mixing over the entire $\mathcal{P}$.
2.  We improve the worst case DR-DMDP sample complexity dependence on $(1-\gamma)^{-4}$ to $(1-\gamma)^{-2}$ in the mixing setting, enabling our the reduction and anchoring approach to achieve the $\widetilde O(\epsilon^{-2})$ sample complexity dependence.

**Empirical Validation:** Our experimental results provide strong validation of our theoretical conclusions. On a log-log scale, the error vs. sample size plot shows a slope of $-1/2$, confirming the our theoretical convergence rate upper bound.

Additionally, as suggested by the reviewers, we included comparisons with DR RVI Q-learning as a baseline and conducted large-scale experiments. Empirically, our algorithms outperforms DR RVI Q-learning in these additional experiments, and the error on large-scale MDPs aligns well with our theoretical predictions as well. This further demonstrates that our algorithm is concise, easy to deploy, and highly effective.

Thank you again to the reviewers and the Area Chair for your time, careful consideration, and constructive feedback.

---

### Decision · Program_Chairs · 2025-09-17

**Decision:**

Accept (poster)

**Comment:**

Summary: This paper investigates the sample complexity of distributionally robust reinforcement learning (DR-RL) under both discounted and average-reward settings. The authors extend recent advances in non-robust MDPs to the robust domain, introducing reductions and anchoring techniques to design algorithms with improved sample complexity guarantees. In particular, they establish the first nontrivial sample complexity bound for distributionally robust average-reward MDPs, matching known lower bounds in the non-robust setting up to mixing-time dependencies. Theoretical results are supported by experiments on a simple hard-instance MDP, which validate the predicted convergence rates.

Comments: This paper has received 4 expert reviews with scores 4, 4, 4, 4, and the average score is 4.0

The reviewers acknowledge that this makes meaningful technical progress by improving sample complexity guarantees in the discounted setting and extending them to the average-reward setting. Results are theoretically significant, with complexity bounds that closely match non-robust lower bounds. The proofs and preliminaries are presented clearly, and the experiments (though limited) confirm the theoretical convergence rates. The algorithms are natural and well-motivated, and the paper may stimulate further research in robust average-reward RL.

The reviewers, however, have raised many concerns. For example, the dependence of the results on parameters such as mixing time and minimum support is not fully clarified, and no lower bounds are provided to confirm optimality. The assumptions of uniform ergodicity and access to simulators reduce the practical relevance of the results.

Based on my reading of the paper, reviews, and the rebuttal discussion, I believe that this paper presents solid technical contributions to the theory of robust RL, particularly in establishing sample complexity guarantees for average-reward settings. While the novelty relative to prior work is not completely original, the results are natural and important, and the analysis is carefully executed. The main weakness lies in the lack of convincing empirical validation and the reliance on strong assumptions, which limit immediate practical impact. Nevertheless, the theoretical advances are substantial enough to warrant acceptance.